# Hyperparameter Optimization via Interacting with Probabilistic Circuits

## Abstract

Despite the growing interest in designing truly interactive hyperparameter optimization (HPO) methods, to date, only a few allow to include human feedback. However, these methods add friction to the interactive process, rigidly requiring to fully specify the user input as prior distribution ex ante and often imposing additional constraints on the optimization framework. This hinders the flexible incorporation of expertise and valuable knowledge of domain experts, which might provide partial feedback at any time during optimization. To overcome these limitations, we introduce a novel Bayesian optimization approach leveraging tractable probabilistic models named probabilistic circuits (PCs) as surrogate model. PCs encode a tractable joint distribution over the hybrid hyperparameter space and evaluation scores, and enable exact conditional inference and sampling, allowing users to provide valuable insights interactively and generate configurations adhering to their feedback. We demonstrate the benefits of the resulting interactive HPO through an extensive empirical evaluation of diverse benchmarks, including the challenging setting of neural architecture search.

## 1 Introduction

Hyperparameters crucially influence the performance of machine learning (ML) algorithms and must be set carefully to fully unleash the algorithm's potential (Bergstra & Bengio, 2012; Hutter et al., 2013; Probst et al., 2019). Manually finding good hyperparameters is a tedious and costly task. Thus, various approaches have been proposed to automatize this process which is referred to as *hyperparameter optimization* (HPO) (Bischl et al., 2023). Generally, HPO is framed as optimizing an expensive blackbox function since the true functional form of the objective is commonly unknown, and the evaluation of hyperparameter configurations is costly, as it requires training ML models several times. Given the rise of deep learning, there is also a growing interest in optimizing the hyperparameters defining the architecture of neural models, commonly referred to as neural architecture search (NAS). The main goal in NAS and HPO is to efficiently traverse the search space to find good configurations quickly and avoid unpromising regions. Bayesian optimization (BO) methods have proven to be sample efficient and converge on good configurations quickly. They exploit knowledge about previously evaluated configurations by learning a surrogate model at each iteration to approximate the optimized unknown objective function (Hutter et al., 2011; Falkner et al., 2018). A selection policy employs the learned surrogate to determine the next configuration to be evaluated (Garnett, 2023; Wang et al., 2022). Selection policies aim to balance the exploitation of promising regions of the search space with the exploration of undiscovered ones. Prominent policies optimize an acquisition function assessing the utility of each configuration in terms of the mentioned trade-off (Hutter et al., 2011; Hvarfner et al., 2022) or employ sampling strategies that frame the selection of the next configuration as a multi-armed bandit problem and maximize a reward provided by the surrogate (Shahriari et al., 2016; Wang et al., 2022).

Although the recent advancements in HPO and NAS could facilitate the design and optimization of ML models for non-experts, in most of the cases, hyperparameters are still tuned manually (Bouthillier & Varoquaux, 2020) and the majority of cutting-edge neural architectures, e.g., transformers (Vaswani et al., 2017), are derived by hand. Given that many ML practitioners perform hyperparameter tuning purely based on their knowledge, experience, and intuition, integrating this valuable knowledge to guide HPO algorithms *during* optimization is of high value since it can substantially foster the search and mitigate its cost. For example, in Fig. 1 (Left), three hyperparameters of the

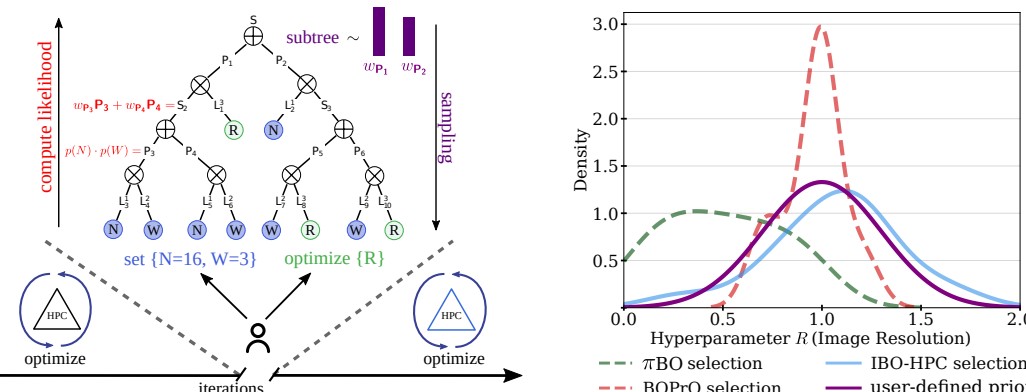

Figure 1: **Interactive Bayesian Hyperparameter Optimization.** (Left) We devise interactive Bayesian HPO by employing PCs as surrogate models encoding a joint distribution over hyperparameters and evaluation scores. They allow users to directly condition the surrogate on their beliefs while new candidates are generated via tractable sampling. Thus, user knowledge (priors or point-wise values) is reflected accurately. (Right) Accurately reflecting user beliefs is crucial for interactive HPO to fully leverage user knowledge. IBO-HPC (our method) precisely reflects the user prior provided over the hyperparameter $R$ (image resolution), while $\pi$BO and BOPrO fail to do so.

JAHS benchmark (Bansal et al., 2022) are optimized (depth multiplier $N$, width multiplier $W$, and resolution $R$ of a deep CNN). During an HPO run, a user might realize that values around $N = 16$ and $W = 3$ yield high-performing models. Hence, a user can guide the HPO algorithm with the obtained knowledge (here, $N = 16$ and $W = 3$ without restarting the optimization from scratch. This can considerably increase the convergence speed and quality of the final solution by focusing the optimization on remaining hyperparameters (here $R$, details in App. A). Recent works by Souza et al. (2021) and Hvarfner et al. (2022) allow users to infuse knowledge into a BO framework via user-defined prior distributions. To integrate user feedback, both approaches employ weighting schemes that alter the behavior of the acquisition function to follow the user-defined priors. Given a configuration, Souza et al. (2021) weight the surrogate's prediction with the prior, while, Hvarfner et al. (2022) directly reshape the acquisition function by weighting it with the prior. Although these approaches are valid and principled ways to guide an HPO task, their weighting schemes limit the set of compatible acquisition functions and, therefore, the selection policies. Furthermore, they assume user knowledge to be available only *ex ante*, i.e., before the optimization, hindering users to adjust their beliefs flexibly, at *any* iteration during the optimization. Moreover, these approaches might not reflect user knowledge precisely as defined in the prior due to the non-linear integration of the priors in the acquisition function. For example, in Fig. 1 (Right), the configurations selected by both BOPrO (Souza et al., 2021) and $\pi$BO (Hvarfner et al., 2022) during the first 20 iterations of optimization remarkably deviate from the given user prior.

To overcome the above limitations and to integrate user feedback in HPO and NAS more flexibly, making the optimization truly interactive, we introduce INTERACTIVE BAYESIAN OPTIMIZATION VIA HYPERPARAMETER PROBABILISTIC CIRCUITS (IBO-HPC).[1] This novel BO method provides an elegant and flexible mechanism to incorporate user knowledge at *any time* during optimization without relying on weighting schemes reshaping an acquisition function. Instead, we derive a selection policy that accurately reflects user beliefs by allowing users to directly *condition* the surrogate on those beliefs. Conditions can be specific values or distributions representing users' uncertainties over values of a subset of hyperparameters. Besides this natural incorporation of user knowledge, we introduce a novel, purely data-driven selection policy which suggests new configuration candidates leveraging conditional sampling and avoiding an additional inner loop optimization of an acquisition function. We achieve these benefits by employing as a surrogate a fairly recent tractable probabilistic model called probabilistic circuits (PCs) (Choi et al., 2020). In contrast with other common choices in HPO and NAS such as Gaussian processes (GPs) (Rasmussen & Williams, 2006) and random forests (RFs) (Breiman, 2001), PCs compactly model a joint distribution. PCs allow the tractable

---

[1]We make our code available at `https://anonymous.4open.science/r/jahs_sand-40B3/` and provide all logs and data at `https://hessenbox.tu-darmstadt.de/getlink/fiL7ifX7hq1TfA7MnxndkELS/logs_v2.zip`.

and exact computation of arbitrary marginal and conditional distributions, and provide tractable (conditional) sampling, rendering them a natural choice for interactive HPO. To ensure robustness against potentially misleading user input, we devise a decay mechanism to decrease the influence of user knowledge over time. In the following, we refer to these PCs that encode joint distributions over hyperparameters and evaluations as *hyperparameter probabilistic circuits* (HPC).

We make the following contributions: **(1)** We introduce a novel HPO method named IBO-HPC that enables direct incorporation of user knowledge into the selection policy at any time s.t. the user knowledge is precisely reflected in the selection policy as specified by the user. **(2)** We formally define a notion of interactive policy in HPO, and show that IBO-HPC conforms to this notion and is guaranteed to reflect user knowledge as provided in the optimization process. **(3)** We provide an extensive empirical evaluation of IBO-HPC showing that it is competitive with strong HPO and NAS baselines without user interaction and outperforms them when leveraging user knowledge.

## 2 RELATED WORK

Bayesian optimization (BO) is an approach to optimize an unknown and expensive black-box function and thus is a natural choice for tackling HPO tasks. BO methods sequentially update, based on observations, a surrogate model that approximates the unknown objective function and captures dependencies between configurations from the search space and the evaluation function's values. In a selection policy, the surrogate is used to choose only promising configurations evaluated next (Hutter et al., 2011; Snoek et al., 2012; Mockus, 1975; Shahriari et al., 2016). Common choices for surrogate models are Gaussian processes (GPs) (Rasmussen & Williams, 2006) or random forests (RFs) (Breiman, 2001).

Due to the rise of deep learning, neural architecture search (NAS) has become increasingly relevant. Besides BO, different approaches based on local search (Den Ottelander et al., 2021), reinforcement learning (Pham et al., 2018; Zoph & Le, 2017), and gradient descent (Liu et al., 2019) have been proposed. See White et al. (2023) for a comprehensive survey on NAS. To foster the development and fair comparison of HPO and NAS algorithms, different benchmarks have been proposed. These benchmarks define a search space over architectures and hyperparameters and train all candidates to provide several quantities such as validation and test accuracy. Ying et al. (2019) and Dong & Yang (2020) introduce such benchmarks for NAS (NAS-Bench-101/201) while Bansal et al. (2022) introduces a benchmark for joint optimization of hyperparameters and neural architectures (JAHS).

Leveraging previous HPO runs to increase the efficiency of HPO on related problems has been considered in several works under the umbrella of hyperparameter transfer learning (HTL). Prominent approaches perform HTL by projecting objective responses of all runs to a common response surface, via learning a common representation of HPO tasks or by pruning the search space based on previous tasks (Yogatama & Mann, 2014; Wistuba et al., 2015; Perrone et al., 2018; Vanschoren, 2018; Salinas et al., 2020; Horváth et al., 2021).

With the goal of making HPO more trustworthy, several works have introduced methods to provide explanations in HPO (Hutter et al., 2014; Moosbauer et al., 2021; Watanabe et al., 2023; Segel et al., 2023), while involving feedback during search has received little attention. Hvarfner et al. (2022) allow users to provide prior beliefs via prior distributions over the search space. The provided user prior is used to reshape the acquisition function when selecting new configurations, thus, favoring configurations that received a high likelihood in the prior. Mallik et al. (2023) propose a similar mechanism to incorporate user knowledge in multi-fidelity optimization. However, such approaches assume an *ex ante* full specification of the priors, often with additional constraints such as requiring invertible priors (Ramachandran et al., 2020) or a specific acquisition function (Souza et al., 2021). As illustrated in Sec. 1, the mentioned approaches have several drawbacks in reflecting given user knowledge well when selecting new configurations. We believe that truly interactive HPO methods should not assume ex ante specification of user knowledge and should reflect user knowledge properly.

## 3 INTERACTIVE HYPERPARAMETER OPTIMIZATION

We now present IBO-HPC, which provides a flexible way to interact with the optimization process via probabilistic queries. This way, user knowledge is decoupled from shaping acquisition functions

while reflecting given knowledge in the selection policy as specified by the user. In this section, we briefly revise BO as a general framework for HPO before introducing our definition of a *feedback adhering interactive policy*. Next, we introduce HPCs and present the concrete instantiation of our interactive optimization method. For completeness, we start with a general formal definition of the HPO problem (Kohavi & John, 1995; Hutter et al., 2019).

**Definition 1** (Hyperparameter optimization). *Given hyperparameters $\mathcal{H} = \{H_1, \dots, H_n\}$ with associated domains $\mathbf{H}_1, \dots, \mathbf{H}_n$, and a set of problem instances $\mathcal{X}$, we define a search space $\Theta = \mathbf{H}_1 \times \cdots \times \mathbf{H}_n$. For a given problem instance $\mathbf{x} \in \mathcal{X}$ and evaluation function $f : \Theta \times \mathcal{X} \to \mathbb{R}$, hyperparameter optimization aims to solve $\boldsymbol{\theta}^* = \arg\min_{\boldsymbol{\theta} \in \Theta} f(\boldsymbol{\theta}; \mathbf{x})$.*

### 3.1 INTERACTIVITY IN BAYESIAN HPO

**Bayesian Optimization.** BO aims to optimize a black-box objective function $f : \Theta \to \mathbb{R}$ which is costly to evaluate, i.e., to find the input $\boldsymbol{\theta}^* \in \arg\min_{\boldsymbol{\theta} \in \Theta} f(\boldsymbol{\theta})$ (Shahriari et al., 2016). BO typically tackles such problem in sequential steps, leveraging two key ingredients: a probabilistic surrogate model and a selection policy determining the next $\boldsymbol{\theta}'$ to be evaluated. Given a set $\mathcal{D}_n$ of observations that correspond to the configurations with associated evaluations $(\boldsymbol{\theta}_j, f(\boldsymbol{\theta}_j))_{j=1\dots n}$, the surrogate $s \in \mathcal{S}$ aims to induce a distribution over functions $p(f|\mathcal{D}_n)$. The selection policy uses $s$ to select the next $\boldsymbol{\theta}' \in \Theta$ s.t. it achieves a good exploration-exploitation trade-off. Prominent selection policies optimize an acquisition function $a : \Theta \times \mathcal{S} \to \mathbb{R}$, such as expected improvement (Jones et al., 1998), that estimates the utility of an evaluation at an arbitrary point $\boldsymbol{\theta} \in \Theta$ under a surrogate $s \in \mathcal{S}$. The configuration $\boldsymbol{\theta}'$, evaluated next, is selected by optimizing $a$. A popular alternative to obtain the next configuration $\boldsymbol{\theta}'$ is Thompson sampling (Wang et al., 2022). The obtained tuple $(\boldsymbol{\theta}', f(\boldsymbol{\theta}'))$ is added to $\mathcal{D}_n$ and used to update the surrogate model for the next iteration. This process is repeated until convergence.

An interactive BO method should be capable of incorporating, at any time, the knowledge provided by users; also, the selection policy should reflect the provided knowledge as specified by the user. Consequently, we formalize the concept of an interactive selection policy, or interactive policy for short, that adheres to these requirements.

**Definition 2** (Interactive Policy). *An interactive policy is a function $p : \mathcal{S} \times \Theta \times \mathcal{K} \to \mathcal{P}(\Theta)$ mapping from the set of surrogates $\mathcal{S}$ and search space $\Theta$ to the set of all distributions $\mathcal{P}(\Theta)$ over the search space $\Theta$ while accepting user knowledge $\mathcal{K} \in \mathcal{K}$.*

Note that the collection of user knowledge $\mathcal{K}$ and the set of surrogates $\mathcal{S}$ are left unspecified to keep Def. 2 rather general. Since Def. 2 can be trivially achieved, e.g. by ignoring user knowledge, we introduce *feedback adhering interactive policies* that guarantee that (i) user knowledge affects the policy's outcome and (ii) user knowledge is reflected in the interactive policy as specified.

**Definition 3** (Feedback Adhering Interactive Policy). *Given user knowledge $\mathcal{K} \in \mathcal{K}$ and surrogate $s_t \in \mathcal{S}$ at iteration $t$, an interactive policy $p$ is called efficacious if $p(\Theta, s_t, \mathcal{K}) \neq p(\Theta, s_t, \emptyset)$ where $\emptyset$ indicates that $p$ is applied without user knowledge. If further $\mathcal{K}$ is provided as a distribution $q(\hat{\mathcal{H}})$ over $\hat{\mathcal{H}} \subset \mathcal{H}$, we call $p$ feedback adhering if it is efficacious and $\int_{\mathcal{H} \setminus \hat{\mathcal{H}}} p(\Theta, s_t, \mathcal{K}) = q(\hat{\mathcal{H}})$ holds, i.e., the distribution over $\hat{\mathcal{H}}$ induced by the selection policy equals the prior $q(\hat{\mathcal{H}})$ in the next iteration.*

In Def. 3, the first condition ensures that the user knowledge has an effect on the sampling policy. The second condition ensures that in the first iteration, after that a user provides a distribution over a subset of hyperparameters, the values sampled for the specified hyperparameters follow exactly the distribution $q$ given by the user. Note that user knowledge could also be misleading; thus, Def. 3 does not require user knowledge to have exclusively positive effects. Equipped with Def. 2 and 3, we now introduce IBO-HPC that adheres to both definitions.

### 3.2 INTERACTIVE BAYESIAN OPTIMIZATION WITH HYPERPARAMETER PROBABILISTIC CIRCUITS

In this section, we introduce an interactive Bayesian optimization method that fulfills Def. 3. It employs hyperparameter probabilistic circuits (HPCs) as a surrogate model and a selection policy leveraging the flexible inference and sampling of HPCs, avoiding an additional inner loop optimization

of an acquisition function. Before delving into our method, we first provide preliminaries on probabilistic circuits.

**Hyperparameter Probabilistic Circuits (HPCs).** Motivated by the lack of truly interactive Bayesian HPO methods, we seek a policy that enables flexible interactions with the optimization procedure by providing an arbitrary amount of knowledge about hyperparameters at any time *during* the optimization while reflecting user beliefs as specified. Probabilistic circuits (Choi et al., 2020) are computation graphs that compactly represent multivariate distributions. PCs can answer a wide range of probabilistic queries in a tractable fashion and (conditionally) generate new samples. These features make them a good candidate for our purpose of building a policy adhering to Def. 3.

More formally, a PC is a computational graph encoding a distribution over a set of random variables $\mathbf{X}$. It is defined as a tuple $(\mathcal{G}, \phi)$ where $\mathcal{G} = (V, E)$ is a rooted, directed acyclic graph and $\phi : V \to 2^{\mathbf{X}}$ is the *scope* function assigning a subset of random variables to each node in $\mathcal{G}$. For each internal node $\mathsf{N}$ of $\mathcal{G}$, the scope is defined as the union of scopes of its children, i.e. $\phi(\mathsf{N}) = \cup_{\mathsf{N}' \in \mathrm{ch}(\mathsf{N})} \phi(\mathsf{N}')$. Each leaf node $\mathsf{L}$ computes a distribution/density over its scope $\phi(\mathsf{L})$. All internal nodes of $\mathcal{G}$ are either a sum node $\mathsf{S}$ or a product node $\mathsf{P}$ where each sum node computes a convex combination of its children, i.e., $\mathsf{S} = \sum_{\mathsf{N} \in \mathrm{ch}(\mathsf{S})} w_{\mathsf{S},\mathsf{N}} \mathsf{N}$, and each product computes a product of its children, i.e., $\mathsf{P} = \prod_{\mathsf{N} \in \mathrm{ch}(\mathsf{P})} \mathsf{N}$. We assume *smooth* and *decomposable* PCs (see App. C for details); thus, our method can exploit tractable inference, sampling, and conditioning of PCs. For a more detailed description of PCs, refer to App. C; for an overview, see Fig. 1 (Left).

**Algorithm 1: Interactive BO with HPCs (IBO-HPC).** Our interactive BO method allows for flexible incorporation of user knowledge at any iteration via conditional sampling enabling true interaction with users.

---

**Data:** Search space $\boldsymbol{\Theta}$ over $\mathcal{H} = \{H_1, \ldots, H_n\}$, problem instance $\mathbf{x} \in \mathcal{X}$, prior distribution $p(\boldsymbol{\Theta})$, objective $f : \boldsymbol{\Theta} \times \boldsymbol{\mathcal{X}} \to \mathbb{R}$, user prior $q(\hat{\mathcal{H}})$ is optional and can be provided at any time, decay $\gamma$

1   $\mathcal{D} = \emptyset$;
2   **for** $i \in \{1, \ldots, J\}$ **do**
3     $\boldsymbol{\theta} \sim p(\boldsymbol{\Theta})$;
4     $\mathcal{D} = \mathcal{D} \cup \{(\boldsymbol{\theta}, f(\boldsymbol{\theta}, \mathbf{x}))\}$;
5   **while** *not converged* **do**
6     every $L$-th iteration, fit HPC $s$ on $\mathcal{D}$;
7     $f^* = \max_f \mathcal{D}$;
8     $b \sim \mathrm{Ber}(\rho)$;
9     **if** *prior $q(\hat{\mathcal{H}})$ given and $b = 1$* **then**
10       sample $N$ conditions $\mathbf{h} \sim q(\hat{\mathcal{H}})$;
11       $\mathbf{C} = \emptyset$;
12       **for** *condition $\mathbf{h}_i$ in $\mathbf{h}$* **do**
13         sample $B$ configurations $\boldsymbol{\theta}'_{1,\ldots,B} \sim s(\mathcal{H} \setminus \hat{\mathcal{H}} | \hat{\mathcal{H}}, f^*)$;
14         $\boldsymbol{\theta}_i^* = \arg\max_{\boldsymbol{\theta}' \in \boldsymbol{\theta}'_{1,\ldots,B}} s(\boldsymbol{\theta}' | f^*)$;
15         $\mathbf{C} = \mathbf{C} \cup \boldsymbol{\theta}_i^*$;
16       $\boldsymbol{\theta}^* \sim \mathcal{U}(\mathbf{C})$;
17     **else**
18       $\boldsymbol{\theta}^* \sim s(\mathcal{H} | f^*)$;
19     $\mathcal{D} = \mathcal{D} \cup (\boldsymbol{\theta}', f(\boldsymbol{\theta}', \mathbf{x}))$;
20     $\rho = \gamma \cdot \rho$;
21     present evaluations $\mathcal{D}$;

---

We jointly model the hyperparameters and evaluation scores with PCs; thus, we refer to these surrogates as hyperparameter probabilistic circuits (HPC). Given the hybrid (discrete and continuous) nature of hyperparameter search spaces, in this work, we focus on a type of PCs tailored for hybrid domains named mixed sum-product networks (MSPNs). An MSPN is a decomposable and smooth PC with piecewise polynomial leaves. These properties types allow MSPNs to represent valid distributions over hybrid domains (i.e. discrete and continuous variables) (Molina et al., 2018).

**Method.** We now describe our method shown in Algorithm 1. Since our surrogate is a density estimator, we start off by sampling $J$ hyperparameter configurations from a prior distribution $p$, e.g., a uniform distribution, and evaluate them by querying the objective function $f$ (**Line 1-5**). The function $f$ yields a performance score of a model trained to solve a given problem instance $\mathbf{x}$ with the sampled configuration $\boldsymbol{\theta}$. After evaluating each sampled $\boldsymbol{\theta}$ we obtain a set $\mathcal{D}$ of pairs $(\boldsymbol{\theta}, f_{\boldsymbol{\theta}}(\mathbf{x}))$ and fit a HPC $s$ estimating the joint distribution $p(\mathcal{H}, F)$, where $\mathcal{H}$ is the set of hyperparameters and $F$ is a random variable representing the evaluation score (**Line 7**). IBO-HPC proceeds by selecting a configuration $\boldsymbol{\theta}$ that gets evaluated next, i.e., our *feedback adhering interactive policy* is applied. Our policy exploits the flexible and exact inference of HPCs to derive arbitrary conditional distributions according to the partial evidence at hand (Peharz et al., 2015). We target the configurations that are likely to achieve a better evaluation score. Thus, a posterior distribution over the hyperparameter space is derived by conditioning on the best score $f^* = \max_f \mathcal{D}$ observed so far alongside with

(optional) user knowledge $\mathcal{K}$. For now, $\mathcal{K}$ is assumed to be given in the form of conditions such as $\hat{\mathcal{H}} = \hat{\mathbf{h}}$ where $\hat{\mathcal{H}} \subset \mathcal{H}$ is a subset of hyperparameters being set to $\hat{\mathbf{h}}$. With Bayes rule and tractable marginal inference and sampling of HPCs, we obtain the conditional distribution and use it to sample a new configuration from promising regions in the search space.

$$p(\mathcal{H} \setminus \hat{\mathcal{H}} | \hat{\mathcal{H}}, F = f^*) = s(\mathcal{H} \setminus \hat{\mathcal{H}} | \hat{\mathcal{H}}, F = f^*)$$
$$\boldsymbol{\theta} \sim p(\mathcal{H} \setminus \hat{\mathcal{H}} | \hat{\mathcal{H}}, F = f^*) \tag{1}$$

Since users might be uncertain about hyperparameter values, defining a prior $q(\hat{\mathcal{H}})$ over $\hat{\mathcal{H}}$ might be more reasonable than setting a fixed value for certain hyperparameters. The prior $q(\hat{\mathcal{H}})$ is interpreted as a distribution over conditions of the form $\hat{\mathcal{H}} = \hat{\mathbf{h}}$ where $\hat{\mathbf{h}} \sim q(\hat{\mathcal{H}})$. This induces a different weighted version of the distribution given in Eq. 1.

$$p(\mathcal{H} \setminus \hat{\mathcal{H}} | \hat{\mathcal{H}}, F = f^*) \cdot q(\hat{\mathcal{H}}) = s(\mathcal{H} \setminus \hat{\mathcal{H}} | \hat{\mathcal{H}}, F = f^*) \cdot q(\hat{\mathcal{H}}) \tag{2}$$

Since user intuition can be wrong, we allow IBO-HPC to recover from sub-optimal user knowledge by deciding whether or not to use the provided $\mathcal{K}$ based on a Bernoulli distribution with success probability $\rho$. To ensure that IBO-HPC gradually recovers when misleading $\mathcal{K}$ is provided, we decrease the likelihood of using $\mathcal{K}$ in each iteration after $\mathcal{K}$ was supplied via a decay factor $\gamma$. When user knowledge is provided at iteration $T$, the distribution over configurations after $T + t$ iterations reads:

$$\gamma^t \rho \cdot s(\mathcal{H} \setminus \hat{\mathcal{H}} | \hat{\mathcal{H}}, F = f^*) \cdot q(\hat{\mathcal{H}}) + (1 - \gamma^t \rho) \cdot s(\mathcal{H} | F = f^*) \tag{3}$$

Note that fusing the distribution in Eq. 2 with the HPC to allow exact inference and conditioning is non-trivial since the prior $q$ is defined over an arbitrary subset and no further assumptions about $q$ are made. Thus we approximate Eq. 2 by sampling $N$ times from $q(\hat{\mathcal{H}})$ and use Eq. 1 to obtain $N$ conditional distributions respecting the user prior $q(\hat{\mathcal{H}})$. To select a promising configuration for evaluation from the approximated distribution in Eq. 2 while still achieving exploration, we sample $B$ configurations from all $N$ conditionals. For each conditional, the configuration maximizing the likelihood $s(\mathcal{H} | F = f^*)$ is selected to reduce the candidate set to configurations likely to achieve a high evaluation score. This leaves us with $N$ configurations from which we sample uniformly to select the configuration evaluated next **(Line 10-16)**. We found that setting $B = 1$ works surprisingly well. A discussion about the quality of our approximation is given in App. B.4. The surrogate is kept fixed for $L$ optimization rounds before retraining it. This fosters exploration by leveraging uncertainty encoded in the (conditional) distribution of the surrogate. An iteration is concluded by updating the set of evaluations $\mathcal{D}$ that can be presented to the users **(Line 20-21)**. The algorithm runs until convergence or another condition for termination, e.g., a time budget limit is encountered.

**Remark 1.** *Although different, our sampling policy shares similarities with Thompson Sampling (TS): TS samples function values from the posterior and selects the next configuration based on the obtained maximum of the function. Instead of sampling the function value from the posterior, we use the maximum obtained so far and use it to sample the next configuration.*

**Theoretical Properties.** After presenting IBO-HPC as an instance of an IBO algorithm, we now show that the policy applied to select the next configuration is a feedback adhering interactive policy according to Def. 3. Since modeling dependencies among hyperparameters is non-trivial for users, we only require users to provide a product distribution as prior knowledge.

**Proposition 1** (IBO-HPC Policy is feedback adhering interactive). *Given a search space $\Theta$ over hyperparameters $\mathcal{H}$, an HPC $s \in \mathcal{S}$, user knowledge $\mathcal{K} \in \mathcal{K}$ in form of a prior $q$ over $\hat{\mathcal{H}} \subset \mathcal{H}$ s.t. the marginal distribution over $\hat{\mathcal{H}}$ of $s$ conditioned on $f^*$ is different than $q(\hat{\mathcal{H}})$, i.e., $\int_{\mathcal{H} \setminus \hat{\mathcal{H}}} s(\hat{\mathcal{H}} | F = f^*) \neq q(\hat{\mathcal{H}})$, the selection policy of IBO-HPC is feedback adhering interactive. The proof is provided in the App. B.1.*

Besides being feedback adhering, IBO-HPC is a global optimizer for black-box optimization.

**Proposition 2** (IBO-HPC is a global optimizer). *IBO-HPC minimizes simple regret, which is defined as $r = f(\mathbf{h}) - f(\mathbf{h}^*)$ for a hyperparameter configuration $\mathbf{h} \in \Theta$ and global optimum $\mathbf{h}^*$. A proof is given in B.2.*

To analyze the convergence rate of IBO-HPC, we involve the expected improvement (EI) with a surrogate PC $s$. This gives us the following proposition.

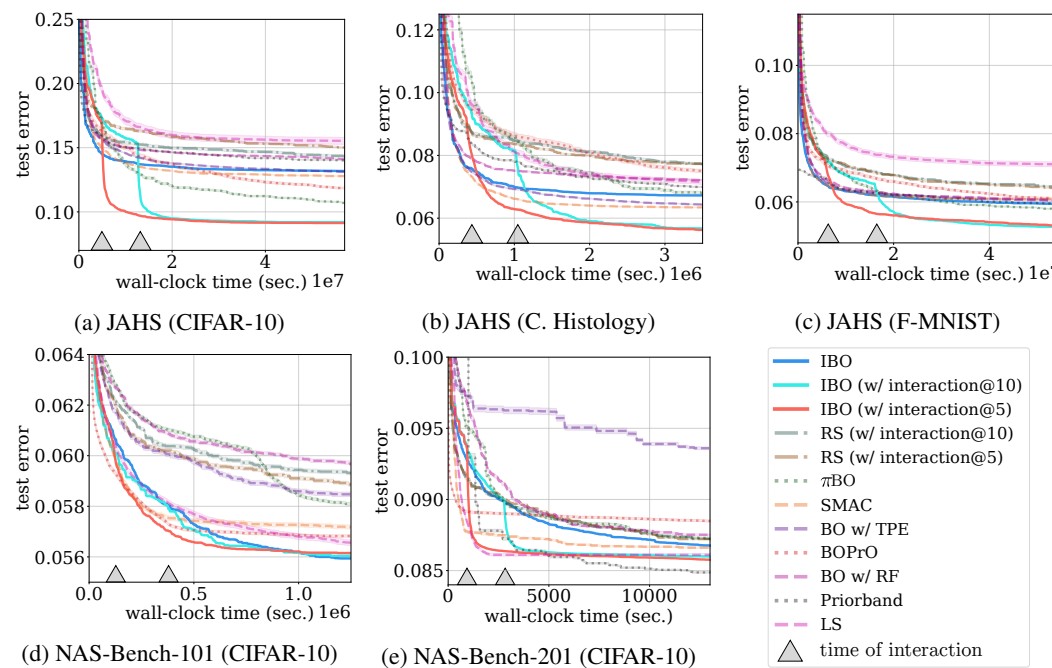

(a) JAHS (CIFAR-10)  (b) JAHS (C. Histology)  (c) JAHS (F-MNIST)

(d) NAS-Bench-101 (CIFAR-10)  (e) NAS-Bench-201 (CIFAR-10)

Figure 2: **IBO-HPC outperforms state of the art.** For 5/5 tasks across three challenging benchmarks, IBO-HPC is competitive with strong baselines when no user knowledge is provided. When beneficial user beliefs (△) are provided, either after 5 iterations (—) or after 15 iterations (—), it outperforms all competitors w.r.t. convergence and solution quality on 4/5 tasks.

**Proposition 3** (Convergence of IBO-HPC). *Assume a non-noisy differentiable $L$-Lipschitz continuous function $f : \mathbb{R}^d \to \mathbb{R}$ with global optimum $\mathbf{h}^* \in \mathbb{R}^d$ that is convex within a ball $B_r(\mathbf{h}^*) = \{\mathbf{h} \in \mathbb{R}^d : ||\mathbf{h} - \mathbf{h}^*|| < r\}$. Further, assume we have given a dataset $\mathcal{D} = \{(\mathbf{h}_1, y_1), \ldots, (\mathbf{h}_n, y_n)\}$ where all $\mathbf{h}_i \in B_r$ and $y_i = f(\mathbf{h}_i)$ and a decomposable, complete PC $s$ over $\mathcal{H} \cup \{F\}$ where the support of $\mathcal{H} = B_r$ and the support of $F = \mathbb{R}$. Assume $s$ locally maximizes the likelihood over $\mathcal{D}$ and that all leaves are Gaussians. Then, the convergence rate of IBO-HPC is lower bounded by the expected improvement (EI) in each iteration, that is*

$$\sum_{i=1}^{\tau_s} w_i \cdot \left( \prod_{j=1}^{d} erf\left(\frac{\mathbf{h}_{tj}^* - \boldsymbol{\mu}_{ij}}{\Sigma_{i_{jj}}\sqrt{2}}\right) - \prod_{j=1}^{d} erf\left(\frac{\mathbf{h}_j^* - \boldsymbol{\mu}_{ij}}{\Sigma_{i_{jj}}\sqrt{2}}\right) + L\epsilon_i \right). \tag{4}$$

*Here, $\tau_s$ is the number of induced trees of $s$ (see Def. 4 in App. B.3), $\epsilon_i = ||\boldsymbol{\mu}_i + \alpha_i \cdot diag(\Sigma_i) - \mathbf{h}^*||$, each $\boldsymbol{\mu}_i$ is the mean vector of a $d$-dimensional multivariate Gaussian defined by the $i$-th induced tree, $\Sigma_i$ is the corresponding correlation matrix and $\mathbf{h}_t^*$ is the best performing configuration until iteration $t$. A proof is given in App. B.3.*

Intuitively, the EI (and thus the convergence rate) is determined by (1) the probability of sampling a configuration in a region bringing us closer to $\mathbf{h}^*$ and (2) how much we expect to move towards the optimum $\mathbf{h}^*$ if (1) occurs. While (1) is lower bounded by considering how close the mixture means are to the best obtained configuration $\mathbf{h}_t^*$ at iteration $t$, and how far off the mixture means are from the optimum $\mathbf{h}^*$, (2) is lower bounded for each mixture component by $\epsilon_i$ and the Lipschitz constant $L$.

## 4 EXPERIMENTAL EVALUATION

We now provide an extensive empirical evaluation of IBO-HPC and aim to answer the following research questions: **(Q1)** Can IBO-PC compete with prominent baseline HPO algorithms? **(Q2)** How does the performance of IBO-HPC, provided with user knowledge at various points during optimization, compare to existing approaches incorporating user knowledge ex ante? **(Q3)** Is IBO-HPC capable of reliably recovering from misleading user interactions?

**Experimental Setup.** We compare IBO-HPC against seven diverse competitors: local search (LS) (White et al., 2020), BO with random forest (RF) surrogate (Head et al.), and SMAC (Hutter et al., 2011) as standard HPO methods. Furthermore, we used random search (RS) (Bergstra & Bengio, 2012) with user priors, BOPrO (Souza et al., 2021), $\pi$BO (Hvarfner et al., 2022) and Priorband (Mallik et al., 2023) which allow ex ante incorporation of user knowledge. Since Priorband is a multi-fidelity method, we reserved the number of epochs as the fidelity in each benchmark. For all non-multi-fidelity methods, set it to the highest possible value. For our evaluation, we employ four real-world benchmarks, i.e., NAS-Bench-101 (Ying et al., 2019) and NAS-Bench-201 (Dong & Yang, 2020) as tabular NAS benchmarks, JAHS (Bansal et al., 2022) as a surrogate benchmark for joint architecture and hyperparameter search, as well as HPO-B (Arango et al., 2021) containing a large collection of challenging OpenML tasks. We evaluate IBO-HPC on seven diverse tasks and five different search spaces covering continuous, discrete, and mixed spaces. Our evaluation considers the optimization of neural architectures and tree-based models on image and tabular data. For NAS (and NAS+HPO), we consider CIFAR-10 (JAHS, NAS-Bench-101, NAS-Bench-201), Fashion-MNIST (JAHS), and Colorectal Histology (JAHS). To make the optimization problem more challenging, we extended the search space definition of JAHS (see App. D). This is possible since JAHS is a surrogate benchmark. Additionally, we consider optimizing an XGBoost and random forest classifier on the credit-g dataset, taken from the HPO-B benchmark. As usual, we optimize the validation accuracy as our objective. All algorithms were repeated on 500 different seeds for a maximum of 2000 iterations (100 iterations for HPO-B). We report the mean test error against computational cost and provide standard error to quantify uncertainty. The computational costs are reported as the accumulated wall-clock time of training and evaluation of each sampled configuration, where the benchmarks provide wall-clock times for each configuration. All experiments were run on DGX-A100 machines.

**User Interactions.** For the experiments, beneficial and misleading user interactions have been defined as user priors for each benchmark. To define priors, we randomly sampled $10k$ configurations and kept the best/worst performing ones, denoted as $\mathbf{h}^+$ and $\mathbf{h}^-$, respectively. To demonstrate that user priors over a few hyperparameters are enough to improve the performance of IBO-HPC significantly, we defined beneficial interactions by selecting a small subset of hyperparameters $\hat{\mathcal{H}} \subset \mathcal{H}$. Then, we defined a prior over each $H \in \hat{\mathcal{H}}$ s.t. the probability of sampling the value of $H$ given in $\mathbf{h}^+$, denoted by $\mathbf{h}^+[H]$, is 1000 times higher than sampling a different value than $\mathbf{h}^+[H]$. For misleading interaction, $\hat{\mathcal{H}}$ was chosen to be large to demonstrate that IBO-HPC recovers even if a large amount of misleading information is provided. We then defined priors over $\hat{\mathcal{H}}$ as for beneficial interactions; however, this time the probability to sample $\mathbf{h}^-[H]$ is 1000 times higher than for other values for each $H \in \hat{\mathcal{H}}$. The priors were chosen to be rather strong since, as emphasized in Sec. 1, the stronger the prior, the better $\pi$BO and BOPrO reflect user knowledge in their selection policy. Striving for a fair comparison, we opted for such strong priors. Further, we aimed to show that IBO-HPC reliably recovers from receiving large amounts of strongly misleading knowledge. Sometimes, it is easier for users to specify a concrete value for certain hyperparameters instead of defining a distribution. Thus, we also conducted experiments with priors defined as a point mass. See App. D for further details.

## 4.1 (Q1) IBO-HPC is Competitive in HPO & NAS

To demonstrate the effectiveness of IBO-HPC, we ran IBO-HPC on all tasks without user interaction. We compared its performance against two strong BO baselines and LS. Fig. 2 shows that the performance of IBO-HPC without user interaction is competitive to or outperforms BO baselines in 4/5 tasks across the NAS-101/201 and JAHS benchmarks. We obtained similar results on HPO-B (see App. D.3). These results show that IBO-HPC performs well in complex and realistic settings. Also, it underlines that HPCs accurately capture characteristics of the objective function and that our sampling-based selection policy reliably identifies good configurations. Besides the quality of the final solution, we also observe that IBO-HPC converges at rates similar to those of the baselines. The only exception, where SMAC and LS achieve better results than IBO-HPC, is NAS-101. While LS is known to be a strong baseline for NAS (White et al., 2020; Den Ottelander et al., 2021), we posit that SMAC's more complex selection policy is particularly effective in handling sparse search spaces like NAS-101, which comes with a much sparser representation than NAS-201 and JAHS. To summarize, we state that IBO-HPC is competitive with existing strong BO baselines when no interaction takes place, and answer (Q1) affirmatively.

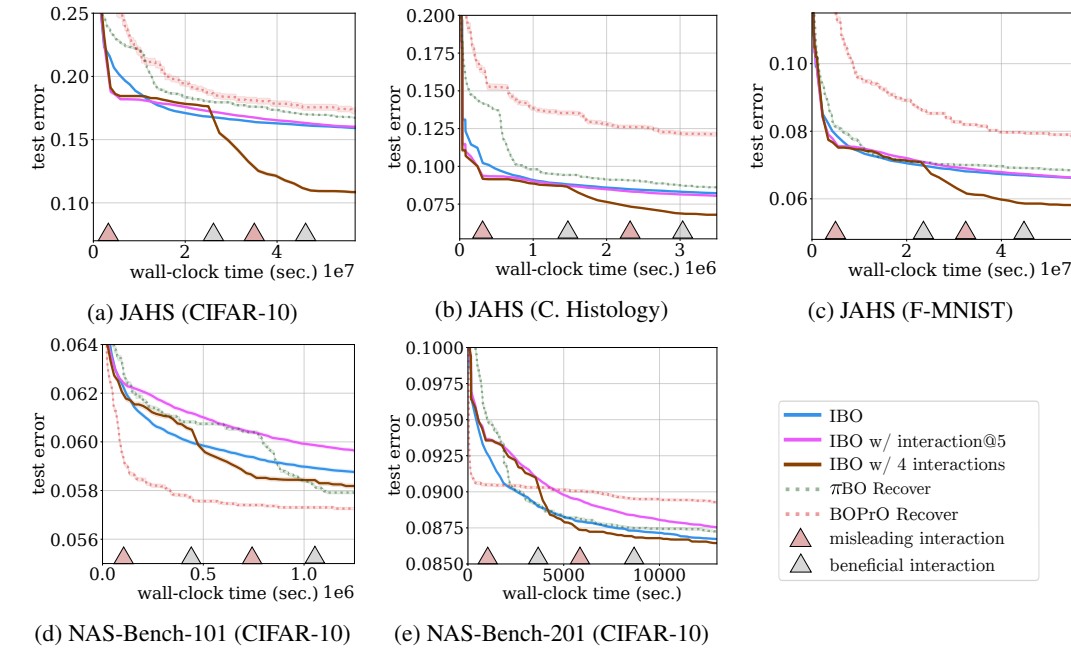

(a) JAHS (CIFAR-10)  (b) JAHS (C. Histology)  (c) JAHS (F-MNIST)

(d) NAS-Bench-101 (CIFAR-10)  (e) NAS-Bench-201 (CIFAR-10)

Figure 3: **IBO-HPC recovers from misleading interactions.** IBO-HPC automatically recovers (—) from misleading feedback provided as point values at the 5th iteration of the search (1st △). Also, when providing harmful and beneficial beliefs alternatively (△/△), IBO-HPC (—) catches up with or outperforms $\pi$BO (····) and BOPrO (····) in 4/5 cases.

### 4.2 (Q2, Q3) INTERACTIVE AND RESILIENT HPO & NAS WITH IBO-HPC

We now demonstrate that IBO-HPC successfully handles various kinds of user knowledge (point values and distributions), analyze the benefits of user knowledge w.r.t. convergence speed, and demonstrate IBO-HPC's recovery from misleading beliefs. Therefore, different beneficial and misleading user beliefs about hyperparameters for all benchmarks were defined (details in App. D).

**Beneficial Interactions.** Fig. 2 shows a clear positive effect of providing beneficial user beliefs (i.e. distribution or fixing hyperparameters) to IBO-HPC across all tasks. This holds for very early interactions (after 5 iterations; — and —) and later interactions (after 15 iterations; —). Remarkably, we observed a clear benefit in terms of convergence speed *and* improvement in solution quality, especially for more complex search spaces. Besides outperforming strong HPO baselines incapable of incorporating user knowledge, IBO-HPC is competitive to or outperforms $\pi$BO, Priorband and BOPrO, which assume user priors to given ex ante. These results show that IBO-HPC's policy reflects user beliefs accurately and leverages provided knowledge effectively by sampling configurations that are promising according to both, user belief and the surrogate HPC. App. D provides additional experiments on HPO-B showing similar results.

**Recovery and Multiple Interactions.** User beliefs could also be misleading for the optimization process; thus, an interactive HPO algorithm should recover from such misleading interactions and allow users to correct their initial beliefs. We demonstrate that IBO-HPC recovers from misleading user knowledge by deliberately providing IBO-HPC with known sub-optimal values for a large number of hyperparameters to ensure a significant negative effect on the optimization process (see App. D for details). Fig. 3 shows that IBO-HPC (—) recovers similarly well or better as $\pi$BO and BOPrO from misleading interactions. In most cases, IBO-HPC catches up with standard HPO competitor methods. This confirms that IBO-HPC's recovery mechanism works reliably and that misleading user beliefs do not deteriorate IBO-HPC's performance in the long run. Again, on NAS-101, IBO-HPC is less effective, which we attribute to the sparse nature of NAS-101 (see Sec. 4.1). Since users might revise their beliefs when no improvement is obtained, we demonstrate that IBO-HPC successfully handles multiple, contradictory interactions. Therefore, we first provided IBO-HPC with the same misleading beliefs as before at an early stage (after 5 iterations), followed by an alternation of beneficial and

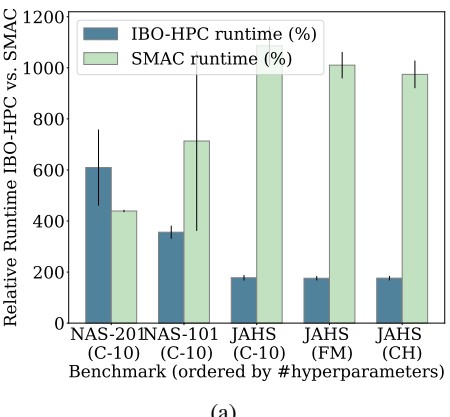
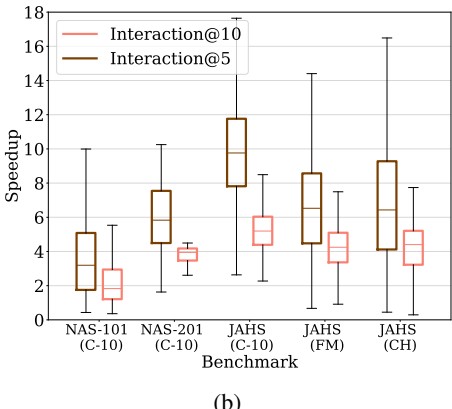

(a)                     (b)

Figure 4: **IBO-HPC achieves considerable runtime improvements.** (a) IBO-HPC is more efficient than SMAC in 4/5 cases (averaged over 20 runs). With the number of hyperparameters increasing, the gap between IBO-HPC and SMAC in terms of computational efficiency is larger. Runtimes of 2000 iterations normalized between [0, 1] are reported per benchmark (highest obtained runtime for a given benchmark is 1). (b) Beneficial interactions lead to significant speed-ups, from 2 to $10\times$.

harmful beliefs every 10 iterations. As expected, the misleading interactions decelerate IBO-HPC, and the recovery mechanism is triggered. In contrast, with beneficial interactions, IBO-HPC quickly catches up with the competitors or even outperforms them, confirming that IBO-HPC leverages valuable feedback in critical conditions (see Fig. 3 (—) and App. D).

**Speed-up.** Both, the runtime of the optimization loop (fitting surrogate and suggesting the next configuration) and convergence speed are crucial for efficient HPO. We, therefore, analyze the average runtime of IBO-HPC's optimization loop and the increase in convergence speed when valuable user knowledge is provided to IBO-HPC as a distribution. In Fig. 4 (a) we compare the runtime of the optimization loop of SMAC and IBO-HPC, averaged over 20 runs. IBO-HPC is considerably faster than SMAC in 4/5 cases, especially in larger search spaces. We attribute this to the efficiency of PCs and of our selection policy (i.e., conditional sampling). In Fig. 4 (b), we ran IBO-HPC without user interaction and obtained the wall-clock time needed for the best evaluation result (denoted as $t_w$). Then, we ran IBO-HPC with beneficial user knowledge and measured the estimated wall-clock time until IBO-HPC found an equally well or better-performing configuration (denoted as $t_i$). Fig. 4 (b) reports the relative performance speedup $\frac{t_w}{t_i}$ for all 500 runs. A median speed-up of 2 to $10\times$ with useful user interactions, clearly demonstrates IBO-HPC's increase in convergence speed while saving resources. Since IBO-HPC effectively incorporates various user interactions, leading to remarkable speedups, and provides a reliable recovery mechanism, we answer **(Q2)** and **(Q3)** positively.

## 5 CONCLUSION

We introduced a novel definition of interactive BO policies and an interactive BO method named IBO-HPC that leverages the flexible inference of probabilistic circuits to flexibly incorporate user beliefs. With no user knowledge, IBO-HPC is competitive with strong baselines and it outperforms competitors when knowledge is available. Also, it reliably recovers from misleading user beliefs and converges significantly faster when provided with valuable user knowledge, thus, saving resources.

**Limitations & Future Work.** Whereas IBO-HPC enables flexible interactive BO, the necessity to retrain the surrogate model at each iteration remains. Thus, prospective directions could explore methods of continual learning (Mundt et al., 2023) to increase the overall efficiency and knowledge reuse over different HPO and NAS settings. Moreover, to model hybrid domains, we relied on HPCs employing piecewise polynomials that might not be sufficient to model complex distributions. Therefore, more sophisticated alternatives could further improve performance.

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

# A    MOTIVATION & REAL-WORLD EXAMPLE

Reflecting user knowledge accurately is crucial for interactive HPO methods to fully benefit from human knowledge and improve trustworthiness. Existing weighting scheme based methods like $\pi$BO and BOPrO fail to reflect user priors accurately in their selection policy as it can be seen in Fig. 5 (a). Here, we show a 1d-example of a Branin function with an optimum around $x = 0.5$. The user prior (in red) is placed at $x = 0.3$. Both $\pi$BO and BOPrO fail to select the next configuration from the high-density region of the prior; thus, the user prior is not incorporated in the selection process as a user would expect. We followed the recommendation of (Hvarfner et al., 2022) and set $\beta = \frac{T}{10}$ where we ran $\pi$BO for $T = 10$ iterations. Our method, IBO-HPC, solves this issue, which we now demonstrate based on a real-world example.

To this end, we ran $\pi$BO, BOPrO – both of which leverage a weighting scheme to incorporate user priors –, and IBO-HPC for $T = 100$ iterations on the CIFAR-10 task of the JAHS benchmark (Bansal et al., 2022). Thus, we set the decay parameter of $\pi$BO to 10. We specified a Gaussian prior distribution with $\mu = 1$ and $\sigma = 0.3$ (Fig. 5 (b), purple) over the hyperparameter RESOLUTION ($R$) that controls the down-/up-sampling rate of an image fed into a neural network. The rest of the hyperparameters for this specific task (i.e. the network architecture and all other hyperparameters; see App. D for details) were optimized by $\pi$BO, BOPrO and IBO-HPC without any user knowledge. All methods received the same user prior ($\pi$BO and BOPrO from the beginning of the optimization; IBO-HPC after 5 iterations). From the iteration the user prior was provided on, we then considered the values chosen for RESOLUTION by $\pi$BO, BOPrO, and IBO-HPC for the next 20 iterations and estimated a density of selected values for $R$ (see Fig. 5 (b)). We chose 20 as the horizon under consideration because for higher $\beta$, the prior is weighted down later in $\pi$BO (see (Hvarfner et al., 2022), Alg. 1) and BOPrO (see (Souza et al., 2021) Eq. 4). In the JAHS setup with $T = 100$ and $\beta = 10$, the prior is weighted down after the 10th iteration in $\pi$BO and BOPrO. In the 20th iteration, $\pi$BO and BOPrO exponentially weigh down the prior with exponent 0.5. The density value of the mode of our prior is then $1.26^{0.5} \approx 1.12$. For IBO-HPC, we chose the decay $\gamma = 0.995$; hence, after 20 iterations, we get $1.26 \cdot \gamma^{20} \approx 1.14$ for the mode of the prior. Thus, we weigh down the prior by approximately the same factor in $\pi$BO, BOPro, and IBO-HPC, ensuring a fair comparison. We obtained that neither the choices for $R$ by $\pi$BO (green dashed line) nor the choices of BOPrO (red dashed line) reflect the user prior as specified. While $\pi$BO's choices of RESOLUTION are biased towards smaller values, BOPrO does not reflect the user's uncertainty well in its choices of RESOLUTION. In contrast, IBO-HPC (blue solid line) precisely reflects the user prior as specified (up to random variations due to sampling).

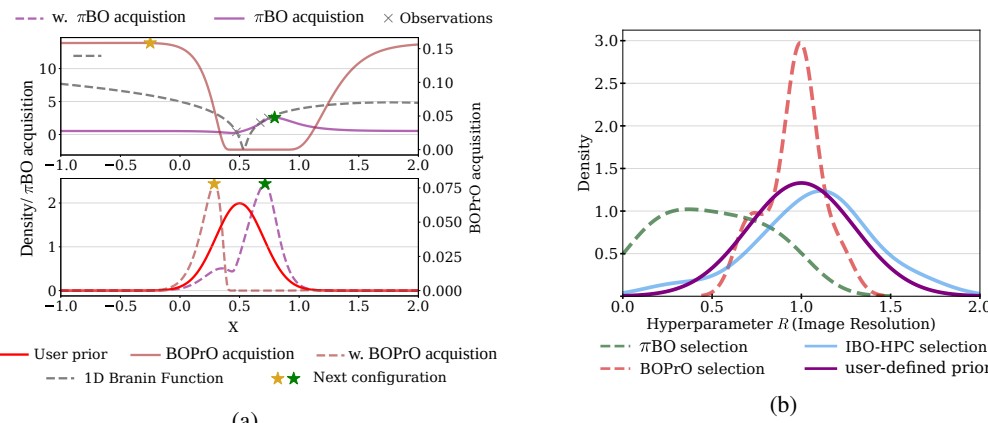

(a)                                                                                  (b)

Figure 5: **IBO-HPC reflects user priors as specified.** In contrast to other weighting scheme based methods like $\pi$BO and BOPrO, IBO-HPC reflects the user prior as specified in its selection policy.

# B    PROOFS

In this section we provide the proof of Proposition 1 of the main paper.

## B.1 IBO-HPC's Policy is Feedback Adhering Interactive

**Proposition 1 (IBO-HPC Policy is feedback adhering interactive).** Given a search space $\Theta$ over hyperparameters $\mathcal{H}$, an HPC $s \in \mathcal{S}$, user knowledge $\mathcal{K} \in \mathcal{K}$ in form of a prior $q$ over $\hat{\mathcal{H}} \subset \mathcal{H}$ s.t. $\int_{\mathcal{H} \setminus \hat{\mathcal{H}}} s(\mathcal{H}|F = f^*) \neq q(\hat{\mathcal{H}})$, the selection policy of IBO-HPC is feedback adhering interactive.

*Proof.* We have to show that the policy of IBO-HPC is feedback adhering, i.e. it conforms with Def. 3: The distribution over the configuration space used to obtain new configurations is different if user knowledge is provided from the distribution used if no user knowledge is provided (policy is efficacious) and the provided user knowledge is represented during configuration selection as specified (feedback adhering).

We first show that the selection policy of IBO-HPC is efficacious.

**IBO-HPC selection policy is efficacious.** Since the decay mechanism allowing IBO-HPC to recover from misleading knowledge can be treated as a constant in each iteration, it is enough if $s(\mathcal{H} \setminus \hat{\mathcal{H}}|\hat{\mathcal{H}} = \hat{\mathbf{h}}, F = f^*) \cdot q(\hat{\mathcal{H}} = \hat{\mathbf{h}}) \neq s(\mathcal{H} \setminus \hat{\mathcal{H}}|\hat{\mathcal{H}} = \emptyset, F = f^*) \cdot q(\hat{\mathcal{H}} = \emptyset)$ holds for any surrogate $s$ representing a joint distribution over search space $\mathcal{H}$ and prior $q$ over $\hat{\mathcal{H}} \subset \mathcal{H}$ to make the policy efficacious. Note that we assume that $\mathcal{K}$ is given in form of a prior $q(\hat{\mathcal{H}})$ over $\hat{\mathcal{H}}$ as before. Since $\emptyset \notin \hat{\mathcal{H}}$ is assumed, our policy ignores any prior if no user knowledge is provided. Thus, in this case, the policy samples from the distribution

$$s(\mathcal{H}|F = f^*) = s(\mathcal{H} \setminus \hat{\mathcal{H}}|\hat{\mathcal{H}}, F = f^*) \cdot \int_{\mathcal{H} \setminus \hat{\mathcal{H}}} s(\mathcal{H}|F = f^*) \tag{5}$$

Since $s(\mathcal{H} \setminus \hat{\mathcal{H}}|\hat{\mathcal{H}}, F = f^*)$ is the same, regardless of whether user knowledge is given or not, user knowledge will lead to a different distribution if $\int_{\mathcal{H} \setminus \hat{\mathcal{H}}} s(\mathcal{H}|F = f^*) \neq q(\hat{\mathcal{H}})$ holds. Since Prop. 1 demands that this is the case, our policy is efficacious according to Def. 2.

We can now proceed and show feedback adherence of the IBO-HPC selection policy.

**IBO-HPC selection policy is feedback adhering.** The proof that our policy is feedback adhering directly follows by design: If a user prior $q(\hat{\mathcal{H}})$ is given, Eq. 3 is approximated by sampling $N$ conditions $\mathbf{h}'_{1,...,N} \sim q(\hat{\mathcal{H}})$ and computing $N$ conditionals $s(\mathcal{H} \setminus \hat{\mathcal{H}}|\hat{\mathcal{H}} = h'_1, F = f^*), \ldots, s(\mathcal{H} \setminus \hat{\mathcal{H}}|\hat{\mathcal{H}} = h'_N, F = f^*)$. We can approximate $q(\hat{\mathcal{H}})$ arbitrarily close with $N \to \infty$. To select the next configuration, we sample $B$ configurations from each of the $N$ conditionals and select the configuration maximizing $s(\mathcal{H}|F = f^*)$ for each conditional. This leaves us with $N$ candidates. Note that at this point, the hyperparameters $\hat{\mathcal{H}}$ still follow $q(\hat{\mathcal{H}})$ with $N \to \infty$ as the conditions of $s(\mathcal{H} \setminus \hat{\mathcal{H}}|\hat{\mathcal{H}} = h'_1, F = f^*), \ldots, s(\mathcal{H} \setminus \hat{\mathcal{H}}|\hat{\mathcal{H}} = h'_N, F = f^*)$ remain fixed and only hyperparameters of the set $\mathcal{H} \setminus \hat{\mathcal{H}}$ can vary/are sampled. Thus, maximizing the likelihood $s(\mathcal{H}|F = f^*)$ is only done w.r.t. hyperparameters in $\mathcal{H} \setminus \hat{\mathcal{H}}$. This implies that sampling hyperparameters $\mathcal{H} \setminus \hat{\mathcal{H}}$ can be biased while sampling from $q(\hat{\mathcal{H}})$ is unaffected because the conditions $\mathbf{h}'_{1,...,N}$ are sampled first in i.i.d. fashion. Our policy selects the configuration evaluated next by uniformly sampling from the remaining $N$ candidates. Since uniformly sampling $L$ times from a set of $N$ samples from a distribution $q$ results in approximating $q$ arbitrarily close for $N \to \infty$ and $L \to \infty$, we conclude that user priors are exactly reflected as specified in our selection policy. This concludes our proof that the selection policy of IBO-HPC is efficacious and feedback adhering. $\square$

## B.2 IBO-HPC Minimizes Simple Regret

We introduce the following proposition:

**Proposition 4** (IBO-HPC minimizes Simple Regret). *IBO-HPC minimizes simple regret, which is defined as $r = f(\mathbf{h}) - f(\mathbf{h}^*)$ for a hyperparameter configuration $\mathbf{h} \in \Theta$ and global optimum $\mathbf{h}^*$.*

*Proof.* Assume that $w > 0$ holds for each weight $w$ of a PC $s$, that each leaf node of $s$ is a distribution $p$ s.t. $p(x) > 0$ for some $x$ and assume $f$ is not noisy. Then, the PC fulfills the positivity assumption, i.e. $s(\mathcal{H} = \mathbf{h}, F = f(\mathbf{h})) > 0$. It follows that $s(\mathcal{H} = \mathbf{h}|F = f^*) > 0$ for any $f^*$ and any $\mathbf{h} \in \Theta$.

Thus, with iterations $T \to \infty$, the probability of sampling the global optimum $\mathbf{h}^*$ in one of the iterations gets 1, and thus $r = f(\mathbf{h}^*) - f(\mathbf{h}^*) = 0$. □

### B.3 Convergence Speed of IBO-HPC

In this section, we analyze the convergence speed of IBO-HPC at each iteration. Therefore, let us state a well-known result of the PC literature on which our analysis is based.

**Definition 4.** *Induced Trees (Zhao et al., 2016). Given a complete and decomposable PC $s$ over $\mathcal{H} = \{H_1, \ldots, H_n\}$, $\mathcal{T} = (\mathcal{T}_V, \mathcal{T}_E)$ is called an induced tree PC from $s$ if*

   1. $\mathsf{N} \in \mathcal{T}_V$ where $\mathsf{N}$ is the root of $s$.

   2. *for all sum nodes $\mathsf{S} \in \mathcal{T}_V$, exactly one child of $\mathsf{S}$ in $s$ is in $\mathcal{T}_V$, and the corresponding edge is in $\mathcal{T}_E$.*

   3. *for all product node $\mathsf{P} \in \mathcal{T}_V$, all children of $\mathsf{P}$ in $s$ are in $\mathcal{T}_V$, and the corresponding edges in $\mathcal{T}_E$.*

We can use Def. 4 to represent decomposable and complete PCs as mixtures (Zhao et al., 2016).

**Proposition 5** (Induced Tree Representation). *Let $\tau_s$ be the total number of induced trees in $s$. Then the output at the root of $s$ can be written as $\sum_{t=1}^{\tau_s} \prod_{(k,j) \in \mathcal{T}_{tE}} w_{kj} \prod_{i=1}^n p_t(H_i = h_i)$, where $\mathcal{T}_t$ is the $t$-th unique induced tree of $s$ and $p_t(H_i)$ is a univariate distribution over $H_i$ in $\mathcal{T}_t$ as a leaf node.*

With this, we are ready to analyze the convergence speed of IBO-HPC in each iteration. Assume a non-noisy differentiable $L$-Lipschitz continuous function $f : \mathbb{R}^d \to \mathbb{R}$ with global optimum $\mathbf{h}^* \in \mathbb{R}^d$ that is convex within a ball $B_r(\mathbf{h}^*) = \{\mathbf{h} \in \mathbb{R}^d : ||\mathbf{h} - \mathbf{h}^*|| < r\}$. Further, assume we have given a dataset $\mathcal{D} = \{(\mathbf{h}_1, y_1), \ldots, (\mathbf{h}_n, y_n)\}$ where all $\mathbf{h}_i \in B_r$ and $y_i = f(\mathbf{h}_i)$ and a decomposable, complete PC $s$ over $\mathcal{H} \cup \{F\}$ where the support of $\mathcal{H} = B_r$ and the support of $F = \mathbb{R}$. Assume $s$ locally maximizes the likelihood over $\mathcal{D}$ and that all leaves are Gaussians. Note that LearnSPN yields decomposable and complete PCs that locally maximize the likelihood of the given data (Gens & Domingos, 2013).

We analyze the convergence properties of our algorithm by examining the expected improvement (EI) in each iteration. Therefore, denote the best score obtained until iteration $t$ as $y_t^*$ and its corresponding configuration as $\mathbf{h}_t^*$. For better readability, we write $s(\mathcal{H} = \mathbf{h}|F = y_t^*)$ as $s(\mathbf{h}|y_t^*)$ from now on. Then, the expected improvement of IBO-HPC within $B_r$ is given by

$$\int_{\mathbf{h} \in B_r} s(\mathbf{h}|y_t^*) \cdot \mathbb{I}[f(\mathbf{h}) < y_t^*] \cdot f(\mathbf{h}) \tag{6}$$

$$= \int_{\mathbf{h}^*}^{\mathbf{h}_t^*} s(\mathbf{h}|y_t^*) \cdot f(\mathbf{h}). \tag{7}$$

Here, w.l.o.g. we assume that $\mathbf{h}_k^* < \mathbf{h}_{tk}^*$ for all dimensions $k \in \{1, \ldots, d\}$ and call $\mathbb{I}$ the indicator function. Using Prop. 5, the fact that the first product of the induced tree representation of a PC $s$ acts as an edge selector, the fact that the conditional of a PC is a PC again, and the Gaussian leaf parameterization of $s$, we can write $s$ as a Gaussian Mixture, i.e., $s(\mathbf{h}|y_t^*) = \sum_{i=1}^{\tau_s} w_i \phi(\mathbf{h}; \boldsymbol{\mu}_i, \Sigma_i)$. Here, $\phi$ is the density of the Gaussian distribution parameterized by mean $\boldsymbol{\mu}$ and covariance matrix $\Sigma$ and corresponds to the second product in the induced tree representation of $s$. Thus, Eq. 6 can be rewritten as

$$\sum_{i=1}^{\tau_s} w_i \int_{\mathbf{h}^*}^{\mathbf{h}_t^*} \phi(\mathbf{h}; \boldsymbol{\mu}_i, \Sigma_i) \cdot f(\mathbf{h}). \tag{8}$$

Due to the $L$-Lipschitz assumption, $||f(\mathbf{h}) - f(\mathbf{h}')|| \le L \cdot ||\mathbf{h} - \mathbf{h}'||$ holds for all $\mathbf{h}, \mathbf{h}' \in B_r$. Hence, we can use a Taylor approximation and write $f(\mathbf{h}) \approx f(\mathbf{h}^*) + \nabla f(\mathbf{h}^*) \cdot ||\mathbf{h} - \mathbf{h}^*||$ which is upper bounded by $f(\mathbf{h}^*) + L||\mathbf{h} - \mathbf{h}^*||$. Then, we can write an upper bound of EI as

$$\sum_{i=1}^{\tau_s} w_i \int_{\mathbf{h}^*}^{\mathbf{h}_t^*} \phi(\mathbf{h}; \boldsymbol{\mu}_i, \Sigma_i) \cdot (f(\mathbf{h}^*) + L||\mathbf{h} - \mathbf{h}^*||) \tag{9}$$

$$= \sum_{i=1}^{\tau_s} w_i \left( \int_{\mathbf{h}^*}^{\mathbf{h}_t^*} \phi(\mathbf{h}; \boldsymbol{\mu}_i, \Sigma_i) \cdot f(\mathbf{h}^*) + \int_{\mathbf{h}^*}^{\mathbf{h}_t^*} \phi(\mathbf{h}; \boldsymbol{\mu}_i, \Sigma_i) \cdot L||\mathbf{h} - \mathbf{h}^*|| \right) \tag{10}$$

$$= \sum_{i=1}^{\tau_s} w_i \cdot \left( f(\mathbf{h}^*) \cdot \int_{\mathbf{h}^*}^{\mathbf{h}_t^*} \phi(\mathbf{h}; \boldsymbol{\mu}_i, \Sigma_i) + \mathbb{E}_{\phi_i}[g_{\mathbf{h}^*}(\mathbf{h})] \right). \tag{11}$$

In the last step, we defined $g_{\mathbf{h}^*}(\mathbf{h}) := L||\mathbf{h} - \mathbf{h}^*||$. Note that we take the expectation w.r.t. the truncated normal distribution because we consider the interval $[\mathbf{h}^*, \mathbf{h}_t^*]$. Also note that $f(\mathbf{h}^*)$ is constant. Thus, we can omit it for the sake of convergence analysis. Since $g_{\mathbf{h}^*}$ is linear, we can use the linearity of the expectation and write

$$\sum_{i=1}^{\tau_s} w_i \cdot \left( \int_{\mathbf{h}^*}^{\mathbf{h}_t^*} \phi(\mathbf{h}; \boldsymbol{\mu}_i, \Sigma_i) + g_{\mathbf{h}^*}(\mathbb{E}_{\phi_i}[\mathbf{h}]) \right) \tag{12}$$

$$= \sum_{i=1}^{\tau_s} w_i \cdot \left( (\Phi(\mathbf{h}_t^*; \boldsymbol{\mu}_i, \Sigma_i) - \Phi(\mathbf{h}^*; \boldsymbol{\mu}_i, \Sigma_i)) + L||\mathbb{E}_{\phi_i}[\mathbf{h}] - \mathbf{h}^*|| \right), \tag{13}$$

where $\Phi(\mathbf{h}; \boldsymbol{\mu}, \Sigma)$ is the cumulative distribution function of multivariate Gaussian. Since the expectations $\mathbb{E}_{\phi_i}[\mathbf{h}]$ are taken over the truncated normal, they can be lower bounded by $\boldsymbol{\mu} + \alpha \cdot \text{diag}(\Sigma)$. Thus, we have to set a series of $\alpha_i$ where each $\alpha_i = \min(\mathbf{h}_t^* - \boldsymbol{\mu}_i, \mathbf{h}^* - \boldsymbol{\mu}_i)$. Then, we can write

$$\sum_{i=1}^{\tau_s} w_i \cdot \left( (\Phi(\mathbf{h}_t^*; \boldsymbol{\mu}_i, \Sigma_i) - \Phi(\mathbf{h}^*; \boldsymbol{\mu}_i, \Sigma_i)) + L||(\boldsymbol{\mu}_i + \alpha_i \cdot \text{diag}(\Sigma_i)) - \mathbf{h}^*|| \right). \tag{14}$$

Setting $\epsilon_i = ||\boldsymbol{\mu}_i + \alpha_i \cdot \text{diag}(\Sigma_i) - \mathbf{h}^*||$ and splitting the sum yields

$$\sum_{i=1}^{\tau_s} w_i \cdot \Phi(\mathbf{h}_t^*; \boldsymbol{\mu}_i, \Sigma_i) - \sum_{i=1}^{\tau_s} w_i \cdot \Phi(\mathbf{h}^*; \boldsymbol{\mu}_i, \Sigma_i) + \sum_{i=1}^{\tau_s} w_i L \epsilon_i. \tag{15}$$

Using that the cumulative multivariate Gaussian $\Phi(\mathbf{h}_t^*; \boldsymbol{\mu}_i, \Sigma_i)$ can be lower bounded by $\prod_{j=1}^{d} \Phi(\mathbf{h}_{ti}^*; \boldsymbol{\mu}_{ij}, \Sigma_{i_{jj}})$, we can lower-bound the entire equation, giving us

$$\sum_{i=1}^{\tau_s} w_i \cdot \prod_{j=1}^{d} \Phi(\mathbf{h}_{tj}^*; \boldsymbol{\mu}_{ij}, \Sigma_{i_{jj}}) - \sum_{i=1}^{\tau_s} w_i \cdot \prod_{j=1}^{d} \Phi(\mathbf{h}_j^*; \boldsymbol{\mu}_{ij}, \Sigma_{i_{jj}}) + \sum_{i=1}^{\tau_s} w_i L \epsilon_i. \tag{16}$$

Since $\Phi(\frac{x-\mu}{\sigma}) = \frac{1}{2}\left(1 + \text{erf}(\frac{x-\mu}{\sigma\sqrt{2}})\right)$ holds, we rewrite

$$\sum_{i=1}^{\tau_s} w_i \cdot \prod_{j=1}^{d} \text{erf}\left(\frac{\mathbf{h}_{tj}^* - \boldsymbol{\mu}_{ij}}{\Sigma_{i_{jj}}\sqrt{2}}\right) - \sum_{i=1}^{\tau_s} w_i \cdot \prod_{j=1}^{d} \text{erf}\left(\frac{\mathbf{h}_j^* - \boldsymbol{\mu}_{ij}}{\Sigma_{i_{jj}}\sqrt{2}}\right) + \sum_{i=1}^{\tau_s} w_i L \epsilon_i \tag{17}$$

$$= \sum_{i=1}^{\tau_s} w_i \cdot \left( \prod_{j=1}^{d} \text{erf}\left(\frac{\mathbf{h}_{tj}^* - \boldsymbol{\mu}_{ij}}{\Sigma_{i_{jj}}\sqrt{2}}\right) - \prod_{j=1}^{d} \text{erf}\left(\frac{\mathbf{h}_j^* - \boldsymbol{\mu}_{ij}}{\Sigma_{i_{jj}}\sqrt{2}}\right) + L \epsilon_i \right). \tag{18}$$

Note that we dropped constants and scaling by $\frac{1}{2}$ of the error function as it does not affect the overall result.

Intuitively spoken, the EI is lower bounded by the cumulative probability mass (given by error function erf) within the region defined by the largest discrepancy between minimal error w.r.t. to the observed data (i.e., bad convergence when $s$ overfits) and the maximal error w.r.t. $\mathbf{h}^*$ (i.e., $\mathcal{D}$ does not contain points close to the optimum), multiplied by a linear approximation of the objective $f$ between the best observed configuration $\mathbf{h}_t^*$ and $\mathbf{h}^*$.

Note that this result does not incorporate user knowledge. The analysis of the effect of user knowledge is straightforward. If helpful user knowledge is given, this can be seen as shifting at least one dimension $j$ of at least one mean vector $\boldsymbol{\mu}_k$ by some $\delta$ towards $\mathbf{h}^*$, i.e., $\boldsymbol{\mu}_{*k} = \boldsymbol{\mu}_k + (0, \dots, \delta, \dots, 0)$. Then, assuming all $\Sigma_i$ stay as above,

$$
\sum_{i=1}^{\tau_s} w_i \cdot \Big( \prod_{j=1}^{d} \mathrm{erf}\Big( \frac{\mathbf{h}_{tj}^* - \boldsymbol{\mu}_{ij}}{\Sigma_{i_{jj}} \sqrt{2}} \Big) - \prod_{j=1}^{d} \mathrm{erf}\Big( \frac{\mathbf{h}_{j}^* - \boldsymbol{\mu}_{ij}}{\Sigma_{i_{jj}} \sqrt{2}} \Big) + L\epsilon_i \Big)
$$

$$
\leq \sum_{i=1, i \neq k}^{\tau_s} w_i \cdot \Big( \prod_{j=1}^{d} \mathrm{erf}\Big( \frac{\mathbf{h}_{tj}^* - \boldsymbol{\mu}_{ij}}{\Sigma_{i_{jj}} \sqrt{2}} \Big) - \prod_{j=1}^{d} \mathrm{erf}\Big( \frac{\mathbf{h}_{j}^* - \boldsymbol{\mu}_{ij}}{\Sigma_{i_{jj}} \sqrt{2}} \Big) + L\epsilon_i \Big)
$$

$$
+ w_k \cdot \Big( \prod_{j=1}^{d} \mathrm{erf}\Big( \frac{\mathbf{h}_{tj}^* - \boldsymbol{\mu}_{kj}}{\Sigma_{k_{jj}} \sqrt{2}} \Big) - \prod_{j=1}^{d} \mathrm{erf}\Big( \frac{\mathbf{h}_{j}^* - \boldsymbol{\mu}_{kj}}{\Sigma_{k_{jj}} \sqrt{2}} \Big) + L\epsilon_k \Big).
$$

This is easy to see since the distribution we sample configurations from is shifted towards the global optimum $\mathbf{h}^*$, thus increasing the probability of sampling a configuration closer to $\mathbf{h}^*$, ultimately leading to faster convergence.

### B.4 Accuracy of IBO-HPC's Selection Policy

Here, we briefly discuss the accuracy of IBO-HPC's policy in selecting new configurations for evaluation based on the obtained data (see Eq. 2). Note that the sampling from the distribution provided in Eq. 2 is accurate if (1) the $s$ represents the data $\mathcal{D}$ accurately and (2) sampling from $s$ and the prior $q$ is unbiased (i.e., samples are drawn according to the underlying distribution). Let us start with (1). Since we employ LearnSPN (Gens & Domingos, 2013) to obtain $s$ (a PC in form of SPN), $s$ will locally maximize the log-likelihood of the training data (i.e., the configuration-evaluation pairs obtained). This means that there is no other SPN in the space of the learnable SPNs via LearnSPN that achieves a better log-likelihood given the data.[2] Hence, as long as the ground truth distribution $p$ (or a good approximation of it) is representable by an SPN, we can recover $p$ with arbitrarily small error with iterations $T \to \infty$.

Looking at (2), we sample from two distributions when selecting a new configuration. First, we sample from the prior $q$, then from the conditional $s(\mathcal{H} \setminus \hat{\mathcal{H}} | \hat{\mathcal{H}} = \mathbf{h}, F = f^*)$ where $\mathbf{h} \sim q$. Assuming $q$ is a tractable distribution (e.g., a parametric one such as an isotropic Gaussian), sampling is immediate and not biased (i.e., performed via simple transformations such as the Box-Muller transform). Note that the assumption on $q$ being a tractable (and relatively simple) distribution can be made safely since providing highly complex distributions as user knowledge is hard to do for most users. When considering sampling from the conditional $s(\mathcal{H} \setminus \hat{\mathcal{H}} | \hat{\mathcal{H}} = \mathbf{h}, F = f^*)$, it should be noted that this conditional is a valid PC again (specifically, a PC in the form of an SPN when obtained with LearnSPN). The model is unchanged and only evaluated differently, i.e., by providing the partial evidence at leaves and evaluating the model bottom-up first (see Choi et al. (2020)). Then, PC sampling is performed top-down by sampling from the simple categorical variables represented by the sum nodes and then from the selected univariate leaves. Thus, the process is tractable (linear in the circuit size) and not biased by further operations or assumptions (Choi et al., 2020). Thus, we conclude that the approximation in Eq. 2 is accurate in the limit $N, T \to \infty$.

## C Probabilistic Circuits

Since probabilistic circuits (PCs) are a key component of our method, we provide more details on these models in the following. Let us first start with a rigorous definition of PCs.

---

[2]Assuming an oracle for the variable splitting. See Proposition 1 in Gens & Domingos (2013).

**Definition 5.** *A probabilistic cricuit (PC) is a computational graph encoding a distribution over a set of random variables $\mathbf{X}$. It is defined as a tuple $(\mathcal{G}, \phi)$ where $\mathcal{G} = (V, E)$ is a rooted, directed acyclic graph and $\phi : V \to 2^{\mathbf{X}}$ is the scope function assigning a subset of random variables to each node in $\mathcal{G}$. For each internal node $\mathsf{N}$ of $\mathcal{G}$, the scope is defined as the union of scopes of its children, i.e. $\phi(\mathsf{N}) = \cup_{\mathsf{N}' \in \mathrm{ch}(\mathsf{N})}$. Each leaf node $\mathsf{L}$ computes a distribution/density over its scope $\phi(\mathsf{L})$. All internal nodes of $\mathcal{G}$ are either a sum node $\mathsf{S}$ or a product node $\mathsf{P}$ where each sum node computes a convex combination of its children, i.e., $\mathsf{S} = \sum_{\mathsf{N} \in \mathrm{ch}(\mathsf{S})} w_{\mathsf{S},\mathsf{N}} \mathsf{N}$, and each product computes a product of its children, i.e. $\mathsf{P} = \prod_{\mathsf{N} \in \mathrm{ch}(\mathsf{P})} \mathsf{N}$.*

With this definition at hand, we describe the tractable key operations of PCs relevant to our method in more detail.

**Inference.** Inference in PCs is a bottom-up procedure. To compute the probability of given evidence $\mathbf{X} = \mathbf{x}$, the densities of the leaf nodes are evaluated first. This yields a density value for each leaf. The leaf densities are then propagated bottom-up by computing all product/sum nodes. Eventually, the root node holds the probability/density of $\mathbf{x}$. Note that typically, multiple leaf nodes correspond to the same random variable. Thus, if the children of a sum node have the same scope, we can interpret sum nodes as mixture models. Conversely, if the children of a product node have *non-overlapping* scopes, a product node can be interpreted as a product distribution of two (independent) random variables. We call these two properties smoothness and decomposability. More formally, *smoothness* means that for each sum node $\mathsf{S} \in V$ it holds that $\phi(\mathsf{N}) = \phi(\mathsf{N}')$ for $\mathsf{N}, \mathsf{N}' \in \mathrm{ch}(\mathsf{S})$. *Decomposability* means that for each product node $\mathsf{P} \in V$ it holds that $\phi(\mathsf{N}) \cap \phi(\mathsf{N}') = \emptyset$ for $\mathsf{N}, \mathsf{N}' \in \mathrm{ch}(\mathsf{P})$, $\mathsf{N} \neq \mathsf{N}'$. Hence, PCs can be interpreted as hierarchical mixture models.

**Marginalization.** Decomposability implies that marginalization is tractable in PCs and can be done in linear time of the circuit size. This is because integrals that can be rewritten by nesting single-dimensional integrals can be computed only in terms of leaf integrals, which are assumed to be tractable as they follow certain distributions (e.g., Gaussian). Computing such nested integrals only in terms of leaf integrals is possible because single-dimensional integrals commute with the sum operation and affect only a single child of product nodes. For more details on the computational implications of decomposability, refer to (Peharz et al., 2015).

Practically, there are two ways to marginalize certain variables from the scope of a PC. One approach is structure-preserving, and marginalization is achieved by setting all leaves corresponding to the set of random variables that are supposed to be marginalized to 1. The second approach constructs a new PC representing the marginal distribution, i.e. the structure of the PC is changed. The second approach is beneficial if samples should be drawn from the marginalized PC because the sampling procedure remains the same, i.e. the PC is adopted to obtain the marginal distribution, not vice versa.

**Conditioning.** Computing a conditional distribution $p(\mathbf{X}_1 | \mathbf{X}_2) = \frac{p(\mathbf{X})}{\int_{\mathbf{x}_2} p(\mathbf{X})}$ where $\mathbf{X}_1 \cup \mathbf{X}_2 = \mathbf{X}$ and $\mathbf{X}_1 \cap \mathbf{X}_2 = \emptyset$ is achieved by combining marginalization (denominator) and inference (numerator). Since inference is tractable for PCs in general and marginalization is tractable for decomposable PCs, conditioning is also tractable.

**Sampling.** Sampling in PCs is a top-down procedure and recursively samples a sub-tree, starting at the root. Each sum node $\mathsf{S}$ holds a parameter vector $\mathbf{w}$ s.t. $\sum_{i=0}^{|\mathrm{ch}(\mathsf{S})|} \mathbf{w}_i = 1$. Based on the distribution induced by $\mathbf{w}$, one of the children of $\mathsf{S}$ is sampled as a sub-tree. By decomposability, the scope of the children of a product node is non-overlapping; thus, sampling from a product node corresponds to sampling from all its child nodes. If a leaf node is reached, a sample is obtained from the distribution at that leaf.

**Learning.** Learning PCs consists of two steps: Identify the structure of the PC and learn the parameters of the PC. A common approach to learning both the structure and parameters is LearnSPN (Gens & Domingos, 2013). We employ LearnSPN to learn the PC after obtaining new data. The basic idea of LearnSPN is to split the data by alternating clustering (i.e., split the data along the sample dimension) and independence tests (i.e., split the data along the features dimension). In other words, the data matrix is split by rows (samples) and columns (features). Usually, rows are clustered when the independence test fails in splitting the features. Clusters correspond to sum nodes in the learned PC, while product nodes correspond to successfully passed independence tests (assessing that two subsets of features are statistically independent). The parameters (i.e., weights of sum nodes) are

set proportional to the cluster sizes of clusters represented by the child nodes of a sum node. Leaf parameters are commonly defined via maximum likelihood estimation.

## D    EXPERIMENTAL DETAILS

Here we present additional details of our empirical evaluation.

### D.1    SEARCH SPACE EXTENSION OF JAHS

To make the HPO problem on JAHS more challenging, we decided to extend the search space slightly as JAHS – as a surrogate benchmark – allows us to query hyperparameter values which were not tested explicitly in the benchmark. We defined three search spaces for JAHS which are presented in the following table.

| | S1 | S2 | S3 |
|---|---|---|---|
| Activation | [Mish, ReLU, Hardswish] | [Mish, ReLU, Hardswish] | [Mish, ReLU, Hardswish] |
| Learning Rate | [1e-3, 1e0] | [1e-3, 1e0] | [1e-3, 1e0] |
| Weight Decay | [1e-5, 1e-2] | [1e-5, 1e-2] | [1e-5, 1e-2] |
| Trivial Argument | [True, False] | [True, False] | [True, False] |
| Op1 | 0-6 | 0-6 | 0-6 |
| Op2 | 0-6 | 0-6 | 0-6 |
| Op3 | 0-6 | 0-6 | 0-6 |
| Op4 | 0-6 | 0-6 | 0-6 |
| Op5 | 0-6 | 0-6 | 0-6 |
| Op6 | 0-6 | 0-6 | 0-6 |
| N | 1-15 | 1-11 | 1-5 |
| W | 1-31 | 1-23 | 1-16 |
| Epoch | 1-200 | 1-200 | 1-200 |
| Resolution | 0-1 | 0-1 | 0-1 |

Table 1: **JAHS Search Space.** We define three versions of the JAHS search space, ranging from simpler to harder spaces.

## D.2 INTERACTIONS

Here we provide the interactions used for our experiments.

**JAHS** The following JSON code shows the interactions performed in our JAHS experiments. The first interaction is a misleading interaction, followed by a beneficial interaction and a no interaction (for recovery).

```
[
    {
        "type": "bad",
        "intervention": {"Activation": 1, "LearningRate":
0.8201676371308472, "N": 15,
        "Op1": 3, "Op2": 4, "Op3": 1, "Op4": 2, "Resolution":
0.5096959403985494,
        "TrivialAugment": 0, "W": 14,
         "WeightDecay": 0.002697686639935806, "epoch": 10},
        "iteration": 5
    },
    {
        "type": "good",
        "intervention": {"N": 3, "W": 16, "Resolution": 1},
        "iteration": 15
    },
    {
        "type": "good",
        "intervention": null,
        "iteration": 20
    },
    {
        "type": "good",
        "kind": "dist",
        "intervention": {"N": {"dist": "cat", "parameters":
        [1, 1, 1, 1e4, 1, 1, 1, 1, 1, 1, 1, 1, 1, 1, 1, 1]},
        "W": {"dist": "cat", "parameters":
        [1, 1, 1, 1, 1, 1, 1, 1, 1, 1, 1, 1, 1, 1, 1, 1, 1e4]},
        "Resolution": {"dist": "uniform", "parameters": [0.98,
1.02]}},
        "iteration": 5
    }
]
```

**NAS-Bench-101** The following JSON code shows the interactions performed in our experiments on NAS-Bench-101. The first interaction is a misleading interaction, followed by a beneficial interaction and a no interaction (for recovery).

```
[
    {
        "type": "bad",
        "kind": "point",
        "intervention": [0, 1, 1, 0, 0, 0, 0, 1, 0, 0, 0, 1, 0, 1,
0, 0, 1, 1, 1, 0, 1],
        "iteration": 5
    },
    {
        "type": "good",
        "kind": "point",
        "intervention": [1, 0, 1, 0, 1, 1, 1, 0, 0, 0, 0, 0, 1, 0,
0, 0, 1, 0, 1, 0, 1],
        "iteration": 12
```

```json
        },
        {
            "type": "good",
            "kind": "point",
            "intervention": null,
            "iteration": 20
        },
        {
            "type": "good",
            "kind": "dist",
            "intervention": {
                "e_0_1": {"dist": "cat", "parameters": [1, 1e4]},
                "e_0_2": {"dist": "cat", "parameters": [1e4, 1]},
                "e_0_3": {"dist": "cat", "parameters": [1, 1e4]},
                "e_0_4": {"dist": "cat", "parameters": [1e4, 1]},
                "e_0_5": {"dist": "cat", "parameters": [1, 1e4]},
                "e_0_6": {"dist": "cat", "parameters": [1, 1e4]},
                "e_1_2": {"dist": "cat", "parameters": [1, 1e4]},
                "e_1_3": {"dist": "cat", "parameters": [1e4, 1]},
                "e_1_4": {"dist": "cat", "parameters": [1e4, 1]},
                "e_1_5": {"dist": "cat", "parameters": [1e4, 1]},
                "e_1_6": {"dist": "cat", "parameters": [1e4, 1]},
                "e_2_3": {"dist": "cat", "parameters": [1e4, 1]},
                "e_2_4": {"dist": "cat", "parameters": [1, 1e4]},
                "e_2_5": {"dist": "cat", "parameters": [1e4, 1]},
                "e_2_6": {"dist": "cat", "parameters": [1e4, 1]},
                "e_3_4": {"dist": "cat", "parameters": [1e4, 1]},
                "e_3_5": {"dist": "cat", "parameters": [1, 1e4]},
                "e_3_6": {"dist": "cat", "parameters": [1e4, 1]},
                "e_4_5": {"dist": "cat", "parameters": [1, 1e4]},
                "e_4_6": {"dist": "cat", "parameters": [1e4, 1]},
                "e_5_6": {"dist": "cat", "parameters": [1, 1e4]}
            },
            "iteration": 5
        }
]
```

**NAS-Bench-201** The following JSON code shows the interactions performed in our experiments on NAS-Bench-201. The first interaction is a misleading interaction, followed by a beneficial interaction and a no interaction (for recovery).

```json
[
    {
        "type": "good",
        "kind": "point",
        "intervention": {"Op_0": 2, "Op_1": 2, "Op_2": 0},
        "iteration": 5
    },
    {
        "type": "bad",
        "kind": "point",
        "intervention": {"Op_0": 1, "Op_1": 2, "Op_2": 1},
        "iteration": 5
    },
    {

        "type": "good",
        "kind": "point",
        "intervention": null,
```

```
        "iteration": 20
    },
    {
        "type": "good",
        "kind": "dist",
        "intervention": {"Op_0": {"dist": "cat", "parameters": [1,
1, 1e4, 1, 1]},
                         "Op_1": {"dist": "cat", "parameters": [1,
1, 1e4, 1, 1]},
                         "Op_2": {"dist": "cat", "parameters":
[1e4, 1, 1, 1, 1]}},
        "iteration": 5
    }
]
```

## D.3 FURTHER RESULTS & ABLATIONS

In this section, we provide further results and ablations. Fig. 6 provides additional results on two challenging tasks of the HPO-B benchmark (search space IDs 6767, 6794; dataset ID 31). Both search spaces have more than 10 hyperparameters and the goal is to solve a classification task. IBO-HPC outperforms the baselines or is competitive with the baselines in both cases, i.e., where feedback is given, and no user feedback is given. Fig 7 shows results of IBO-HPC on JAHS, NAS201, and NAS101 where the given user feedback was either a fixed value or a distribution over configurations. Both cases are handled well by IBO-HPC, demonstrating its flexibility. Fig. 8 provides a more detailed view of the effectiveness of IBO-HPC and its recovery mechanism. It can be seen that IBO-HPC successfully recovers from harmful user feedback in JAHS and NAS-201 (—). Also, it can be seen that IBO-HPC handles alternating and contradictory user feedback well by leveraging information from beneficial feedback and ignoring harmful feedback (—). In NAS-101, however, IBO-HPC is less effective in general, which can be explained by the extreme sparsity of the NAS-101 benchmark. While NAS-101 and NAS-201 are highly similar, NAS-101 uses a binary encoding of architectures, while NAS-201 uses a much denser dictionary-like representation. Although both benchmarks are highly similar, IBO-HPC performs well on NAS-201 but is not as effective on NAS-101, underlining our explanation.

Fig. 9 shows the CDF of test accuracy across the baselines and IBO-HPC. It can be seen that IBO-HPC invests more computational resources in good-performing configurations than other methods while achieving state-of-the-art or better results. In other words, IBO-HPC avoids exploration in unpromising regions of the search space. This is because IBO-HPC samples configurations from a conditional distribution where the condition is the best evaluation score obtained. Thus, exploration is purely data-driven and focuses on regions that perform similarly to the incumbent at a particular iteration.

Fig. 10 shows the influence of the decay parameter $\gamma$ in cases where harmful or misleading user knowledge was provided to IBO-HPC at an early iteration (10 in this case). It can be seen that for higher $\gamma$, IBO-HPC requires more time to recover than for smaller $\gamma$. This aligns with our expectations since a larger $\gamma$ corresponds to a high likelihood of the user knowledge being used for many iterations. In contrast, if $\gamma$ is small, likely, the user knowledge is only considered for a certain number of iterations with high likelihood. Thus, for smaller $\gamma$ IBO-HPC can recover faster.

Fig. 11 shows the effect of conditioning on the $\{0.25, 0.5, 0.75\}$-quantile of the obtained evaluation scores instead of the maximum evaluation score. As expected, the higher the quantile, the better the performance of IBO-HPC as we aim to maximize the objective function. Thus, conditioning on higher values guides the optimization algorithm to configurations that yield better evaluation scores.

Lastly, Fig. 12 depicts the effect of changing $L$, i.e. the number of samples drawn from the surrogate before the surrogate is updated. We found that the sample size has no effect on the overall performance of IBO-HPC. However, for some tasks (JAHS CIFAR-10 and CO), a significant variation of convergence speed in early iterations – depending on the choice of $L$ – was obtained. Choosing $L = 20$ seems to lead to fast and stable convergence behavior.

We followed the same experimental protocol as for all other experiments in Fig. 10-12, except that each algorithm was run only 100 times instead of 500 times on each task.

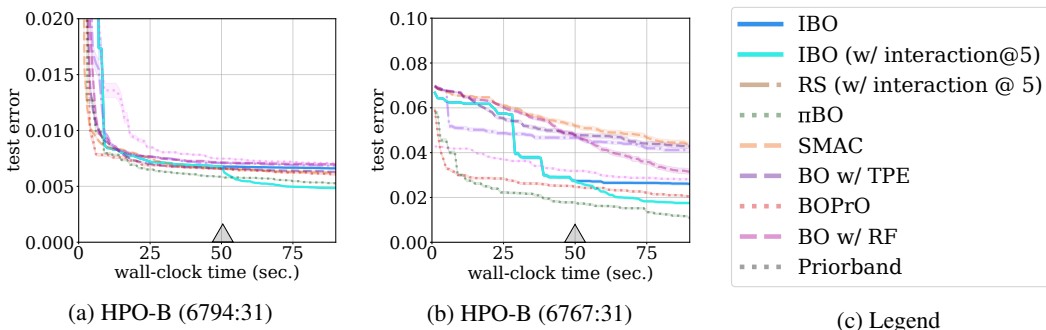

(a) HPO-B (6794:31)    (b) HPO-B (6767:31)    (c) Legend

Figure 6: **IBO-HPC is competitive or outperforms strong baselines on HPO-B.** (a) IBO-HPC outperforms all BO baselines that allow users to provide a prior before optimization when feedback is provided at the 5th iteration. Moreover, IBO-HPC is competitive to other BO methods without any user knowledge given. Reults were obtained on HPO-B with search space ID 6794 and dataset ID 31. (b) IBO-HPC outperforms all BO baselines when no user feedback is provided and beats all interactive BO baselines, except for $\pi$BO, when feedback is provided at the 5th iteration. Results are obtained on HPO-B with search space ID 6767 and dataset ID 31.

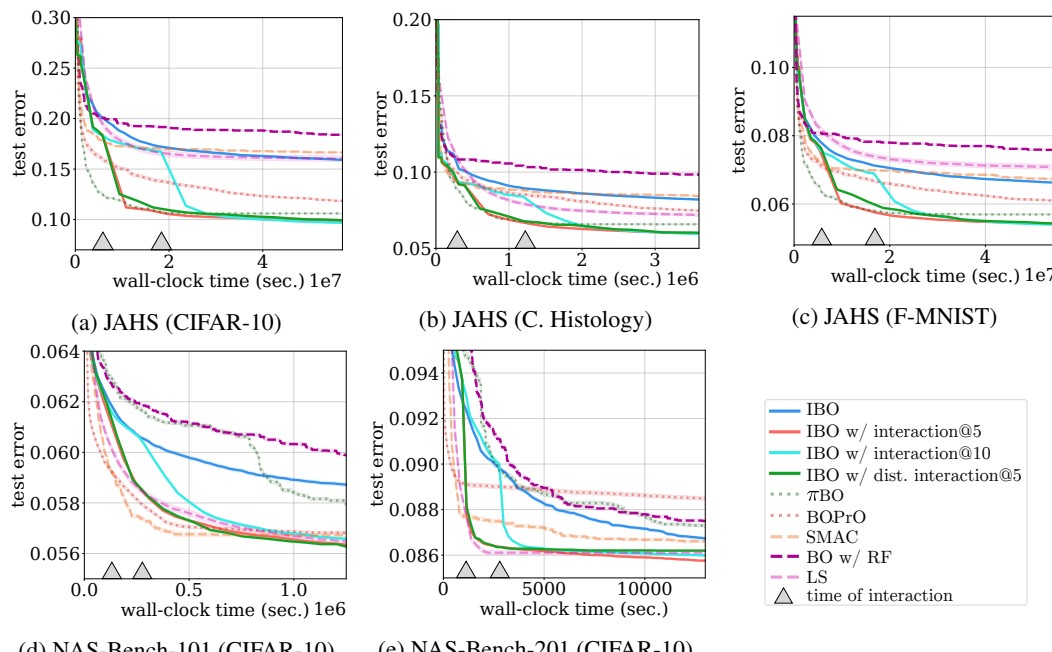

(a) JAHS (CIFAR-10)    (b) JAHS (C. Histology)    (c) JAHS (F-MNIST)

(d) NAS-Bench-101 (CIFAR-10)    (e) NAS-Bench-201 (CIFAR-10)

Figure 7: **IBO-HPC outperforms state of the art.** For 4/5 tasks across three challenging benchmarks, IBO-HPC is competitive with strong baselines when no user knowledge is provided. When beneficial user beliefs (△) are provided, either as distributions (—) or point values (—, —), it outperforms all competitors w.r.t. convergence and solution quality on most tasks. Early interactions (—/— at 5th iteration, — at 10th iteration) speed convergence up.

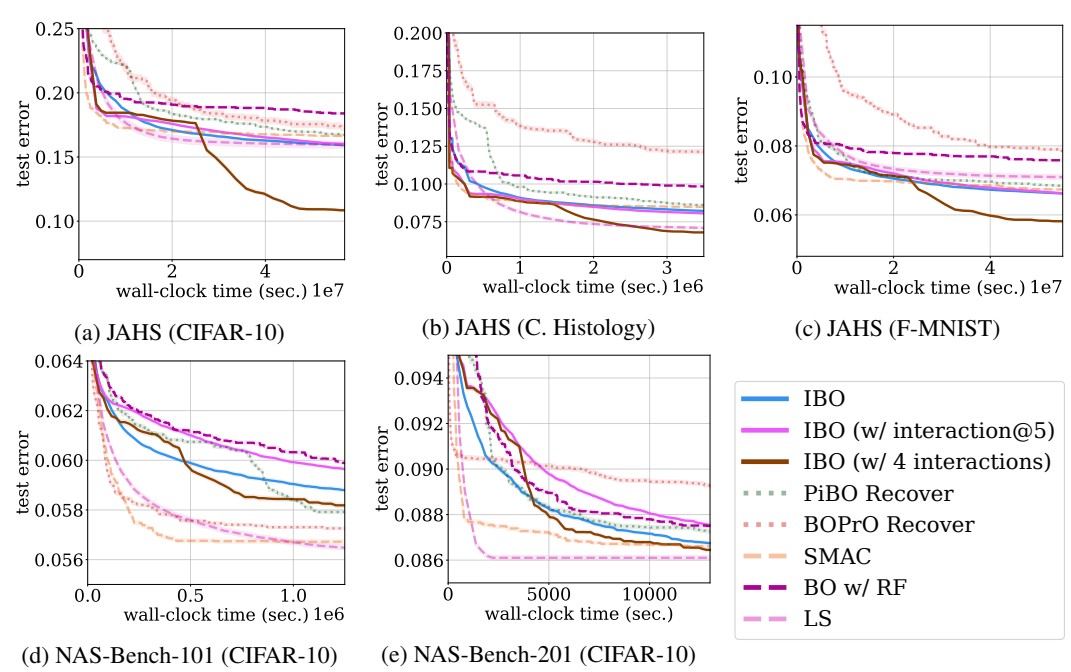

Figure 8: **IBO-HPC recovers from misleading user feedback.** IBO-HPC successfully and consistently recovers from misleading user feedback and performs equally well as if no feedback was given. Also, IBO-HPC handles alternating, contradictory feedback well and is able to leverage beneficial feedback while ignoring misleading feedback.

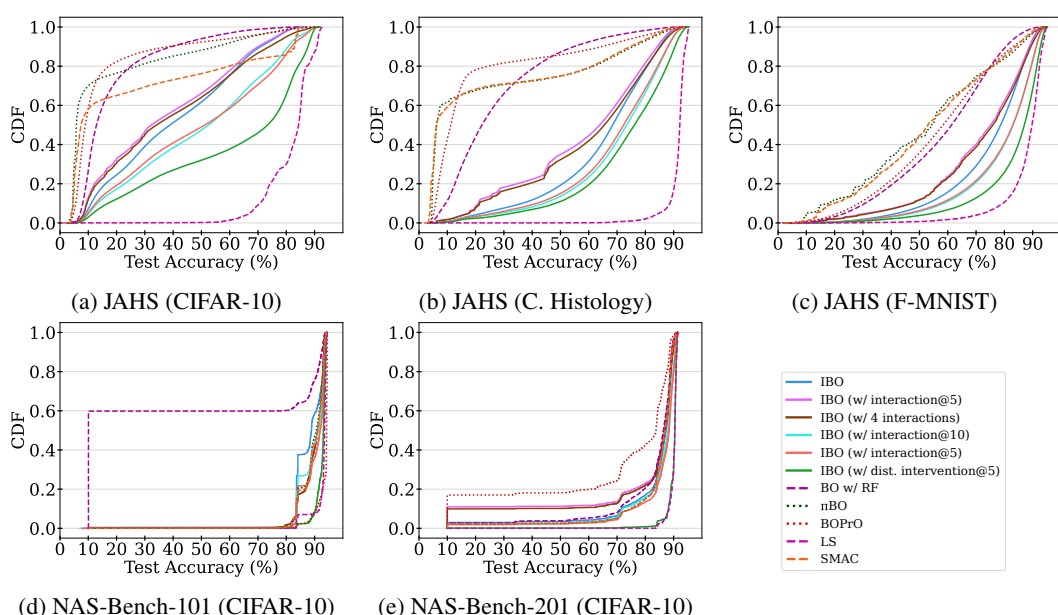

Figure 9: **CDF of Test Accuracy.** IBO-HPC samples more good-performing configurations than most other BO baselines on most tasks. Thus, IBO-HPC invests more computational resources in good configurations than other methods. We conjecture that this is because IBO-HPC selects configurations s.t. they are likely to perform similarly to the incumbent in each iteration. Interestingly, RS also samples many well-performing configurations on the JAHS benchmark.

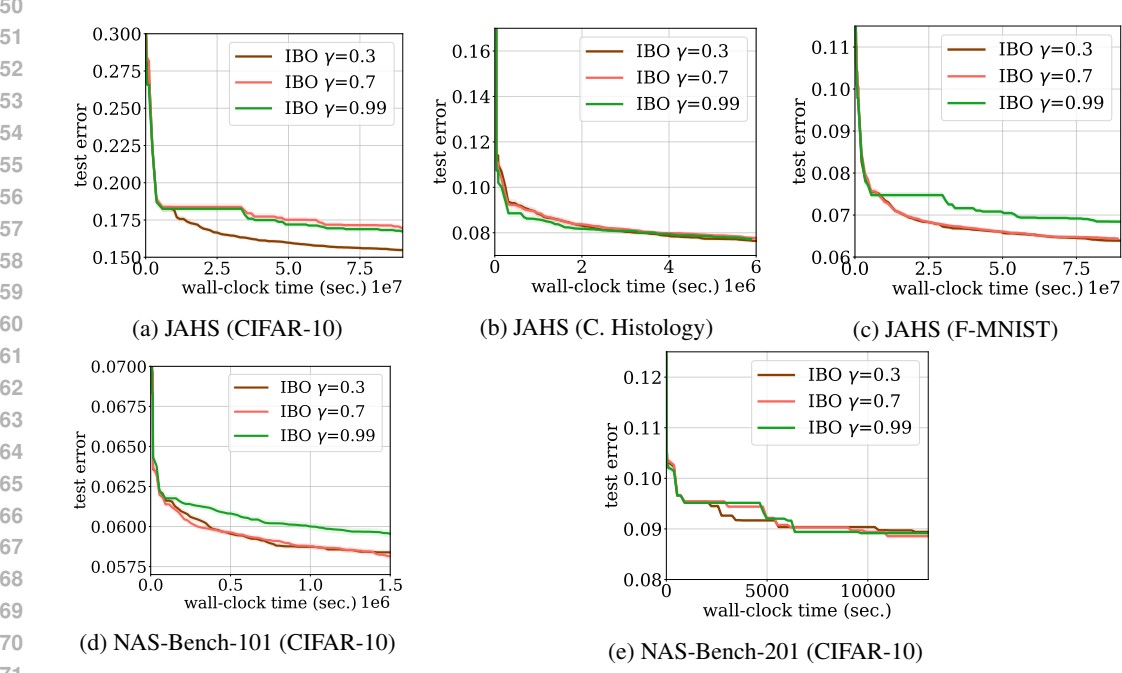

(a) JAHS (CIFAR-10)     (b) JAHS (C. Histology)     (c) JAHS (F-MNIST)

(d) NAS-Bench-101 (CIFAR-10)     (e) NAS-Bench-201 (CIFAR-10)

Figure 10: **Ablation: Effect of $\gamma$ on recovery of IBO-HPC.** As expected, we found that IBO-HPC recovers faster for smaller values of $\gamma$. This is because smaller $\gamma$ values lead to a higher decay of the probability of conditioning on the provided user knowledge. Thus, with faster decay, IBO-HPC recovers faster from harmful or misleading user knowledge (provided at iteration 10).

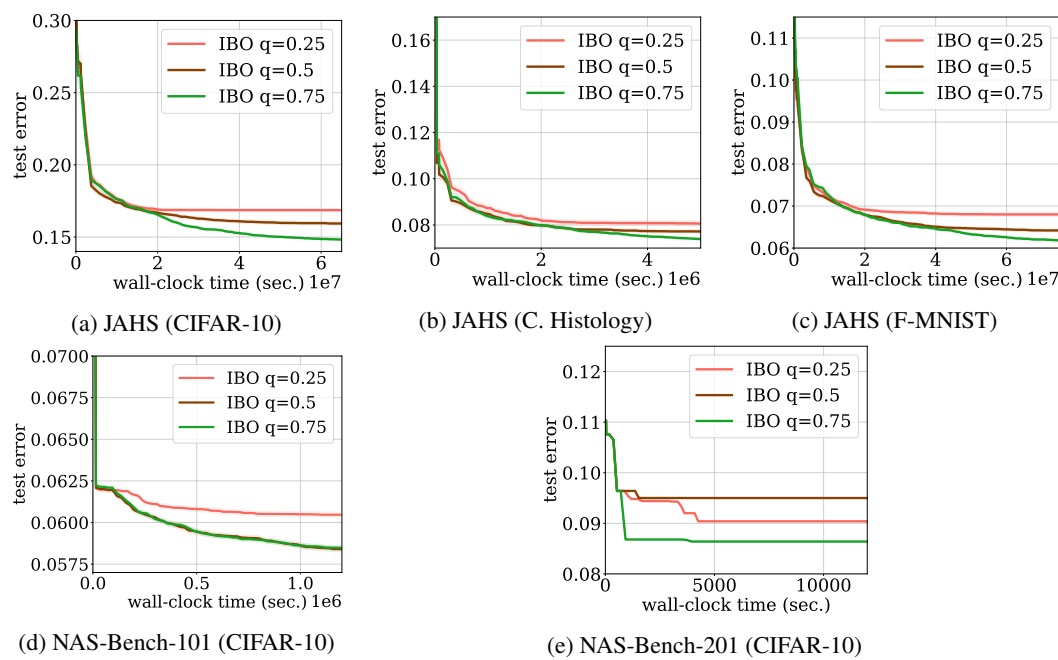

(a) JAHS (CIFAR-10)     (b) JAHS (C. Histology)     (c) JAHS (F-MNIST)

(d) NAS-Bench-101 (CIFAR-10)     (e) NAS-Bench-201 (CIFAR-10)

Figure 11: **Conditioning on sub-optimal evaluation scores slow down IBO-HPC-** Conditioning on the evaluation score of high-performing configurations is crucial for the performance of IBO-HPC. To analyze the effect of conditioning on evaluation scores of sub-optimal configurations, we conditioned on the $\{0.25, 0.5, 0.75\}$-quantile of all evaluation scores obtained until iteration $t$. As expected, for higher quantiles (i.e. better evaluation scores), IBO-HPC finds better configurations.

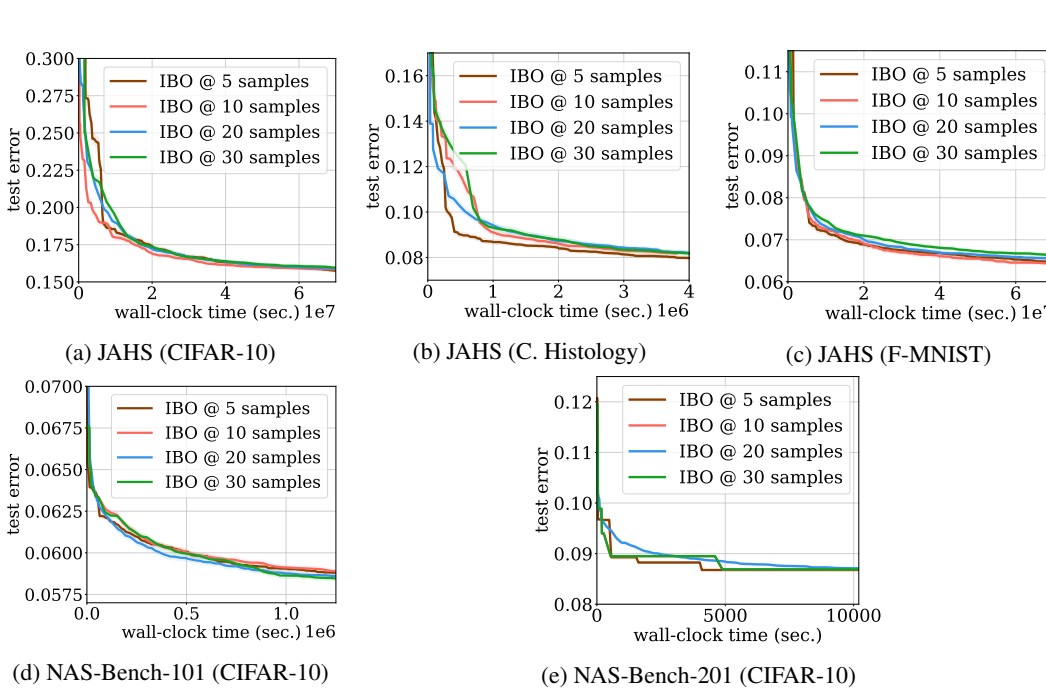

(a) JAHS (CIFAR-10)     (b) JAHS (C. Histology)     (c) JAHS (F-MNIST)

(d) NAS-Bench-101 (CIFAR-10)     (e) NAS-Bench-201 (CIFAR-10)

Figure 12: **$L$ has no significant effect on IBO-HPC's performance.** We found that fixing the surrogate model for $L = \{5, 10, 20, 30\}$ iterations does not lead to significant differences in the performance and convergence speed of IBO-HPC. Only in earlier iterations was a significant variation in convergence speed found on JAHS CIFAR-10 and JAHS CO. However, these variations vanish with the progress of optimization.

## D.4    COMPUTATIONAL COST

We now provide details on the computational costs of IBO-HPC. Therefore, we analyzed the composition of the overall runtime of an optimization run and measured the time needed to train configurations suggested by IBO-HPC versus the time spent on actually performing optimization (including fitting the surrogate PC and sampling new configurations). Fig. 13a shows that the computation time spent on learning the PC and sampling new configurations is negligible compared to the time spent on training the suggested configurations. Additionally, 13b shows that IBO-HPC is faster than SMAC in 4/5 cases in terms of runtime. Here, we considered the time spent in updating the surrogate and suggesting new configurations. Note that this does not include training costs. Interestingly, with the increasing size of the search space, the efficiency advantage of IBO-HPC is increasing. We suspect that the intensify-mechanism in SMAC, which includes a local search, is the reason for the higher computational costs of SMAC.

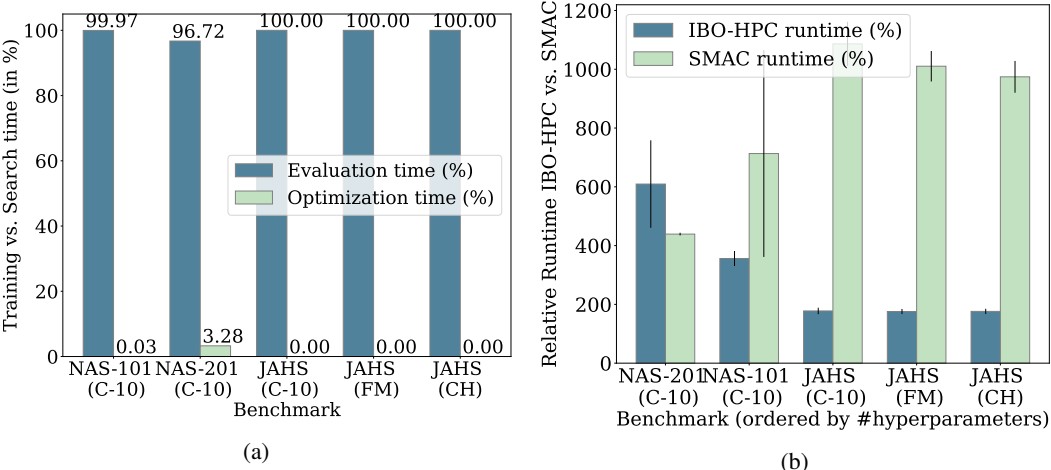

(a)

(b)

Figure 13: **IBO-HPC is a cost-efficient HPO method.** (a) Learning a surrogate and suggesting new configurations is negligible in terms of computational costs compared to training the suggested configurations. We computed the time spent on training configurations (blue) vs. time spent learning a PC and suggesting new configurations (orange). In all experiments, the training of configurations caused the large majority of computational costs, often even approaching 100%. (b) IBO-HPC is more efficient than the prominent HPO algorithm SMAC in 4/5 cases (averaged over 20 runs). Also, with the increasing number of hyperparameters, the gap between IBO-HPC and SMAC in terms of computational efficiency is larger. We report runtimes normalized between [0, 1] per benchmark s.t. the highest obtained runtime for a given benchmark is 1.

## D.5    EXPLORATION-EXPLOITATION TRADE-OFF OF IBO-HPC

An effective mechanism to trade off exploration versus exploitation is crucial for high-performing hyperparameter optimization algorithms. Below we show that IBO-HPC's sampling policy effectively achieves this trade-off. In early iterations, IBO-HPC explores the search space (high sample variance), while in later iterations, it exploits the knowledge collected (low sample variance).

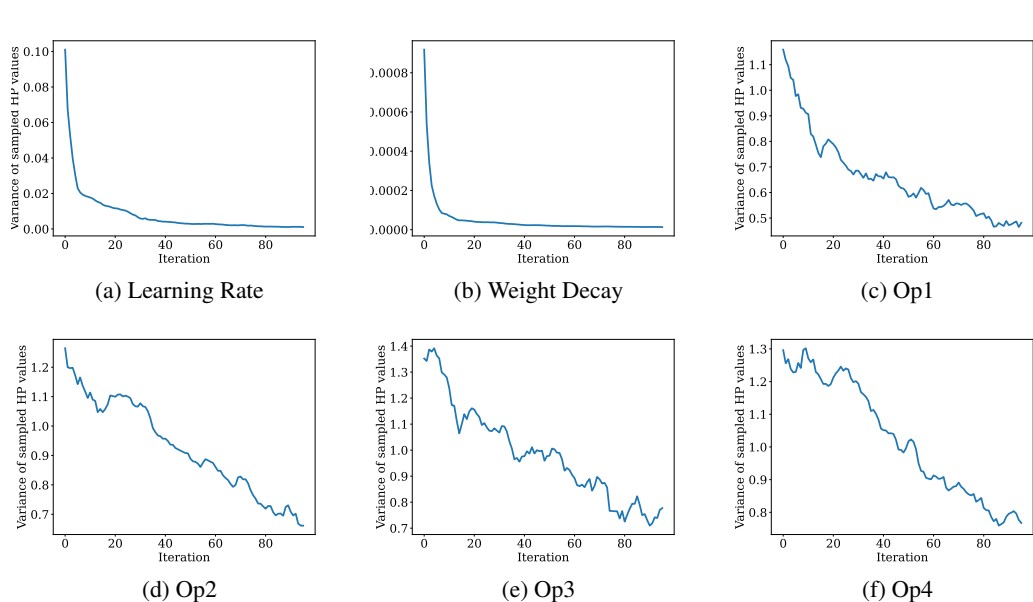

Figure 14: **IBO-HPC effectively trades off exploration and exploitation.** IBO-HPC's sampling policy naturally and effectively trades off exploration (high sampling variance in early iterations) versus exploitation (low sampling variance in later iterations). We show the sampling variance of 6 hyperparameters of the JAHS benchmark (CIFAR10) for each iteration, averaged over 20 runs of IBO-HPC.

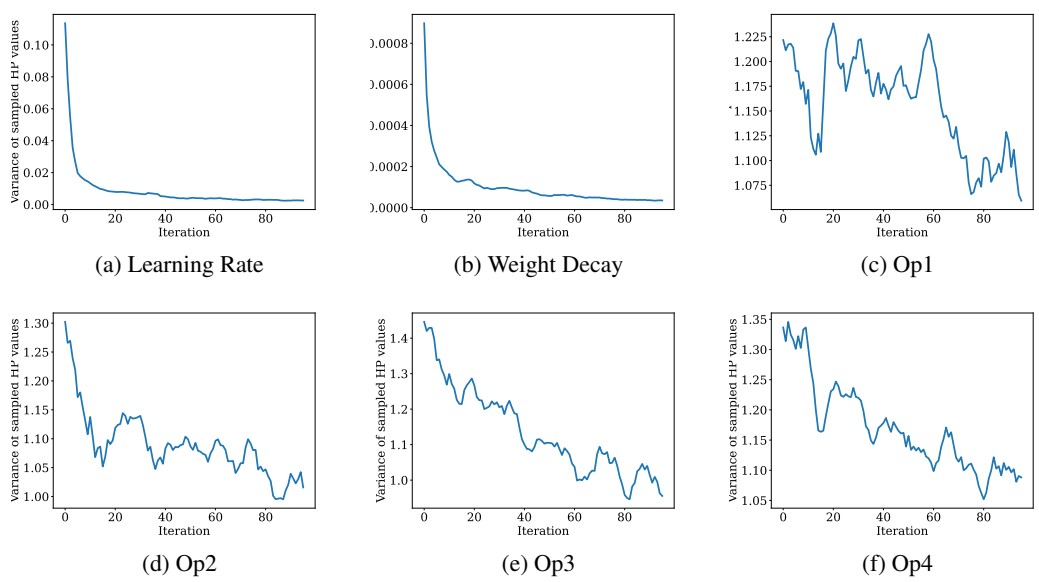

Figure 15: **IBO-HPC effectively trades off exploration and exploitation.** IBO-HPC's sampling policy naturally and effectively trades off exploration (high sampling variance in early iterations) versus exploitation (low sampling variance in later iterations). We show the sampling variance of 6 hyperparameters of the JAHS benchmark (Colorectal Histology) for each iteration, averaged over 20 runs of IBO-HPC.

### D.6 HYPERPARAMETERS OF IBO-HPC

IBO-HPC comes with a few hyperparameters itself, which have to be set. For our experiments, we set the number of iterations the surrogate is kept fixed $L = 20$, the decay value $\gamma = 0.9$. We let all methods optimize for 2000 iterations for fair comparison. Our surrogate models, i.e., PCs and the associated learning algorithm, have some hyperparameters as well. The structure learning algorithm splits use the RDC independence test and K-means clustering. The threshold to detect independencies is set to $0.3$, and the minimum number of instances per leaf is adapted dynamically based on the number of configurations tested during an optimization run.

### D.7 HARDWARE

We ran all our experiments on DGX-A100 machines and used 10 CPUs for each run, thus parallelizing some sub-routines (e.g. learning of PCs). We did not use any GPUs as we queried the benchmarks employed to provide the performance of configurations. The JAHS benchmark requires a relatively large RAM ($> 16GB$) to run smoothly as it loads large ensemble models.

## E WORKING EXAMPLE

In the following we consider a more detailed example of our proposed method from a user perspective. We assume that we only optimize 3 hyperparameters here, W, N and R which correspond to the hyperparameters W, N and RESOLUTION in the JAHS benchmark.

The optimization starts where each of the hyperparameters gets optimized by our method. At some point, the user interacts with the optimization process and sets W and N to a fixed value (blue in Fig. 16). From then on, the model only optimizes the remaining hyperparameter R (green in Fig. 16), using conditional sampling from the resulting conditional distribution that the HPC represents after the interaction.

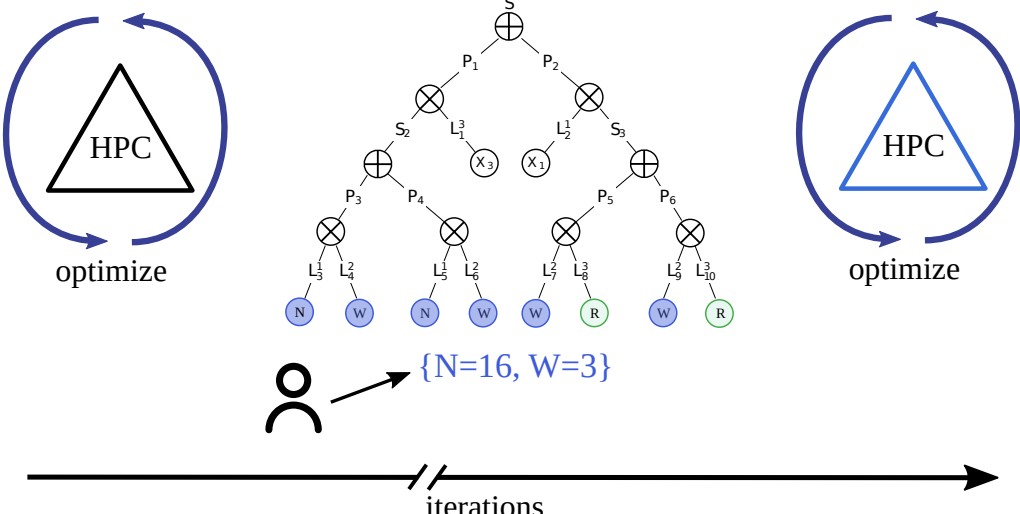

Figure 16: **Example of IBO-HPC.** A user specifies certain aspects of the hyperparameter search space during optimization. Afterward, IBO-HPC takes user knowledge into account when sampling new configuration candidates.

