# OpenReview forum: "Hyperparameter Optimization via Interacting with Probabilistic Circuits"
_ICLR.cc/2025/Conference — Submitted to ICLR 2025_

### Official Review · Reviewer_nTfS · 2024-11-01

**Soundness:** 3
**Presentation:** 2
**Contribution:** 2
**Rating:** 5
**Confidence:** 3

**Summary:**

The paper presents an algorithms for performing Bayesian optimization for hyperparameter tuning that is amenable to interaction with users. Users can indeed input their belief of promising hyperparameters at any iteration (compared to previously proposed approaches that only allow user to input at the beginning of the process) in the form of (possibly point mass) distributions. The method relies on previously proposed probability circuits and a sampling mechanism recalling Thompson sampling.
The authors report various experiments showing how the proposed method IBO-HPC performs both without user inputs and with beneficial and detrimental user inputs, comparing to various baselines.

**Strengths:**

- The motivation of the work is well presented and clear
- the application of probability circuits to HPO is novel to the best of my knowledge
- PCs the integration of user knowledge seems natural and justified

**Weaknesses:**

- some unclear notation and not properly introduced variables (see questions)
- to me, the overall novelty value of this work is not immediately clear. On the one hand, it does look like to me that the work is a direct application of the general framework of BO with a surrogate model given by PC. The integration of user knowledge in the form of prior distribution seems fairly straightforward, but not necessarily unique to the proposed framework (although this work focuses on this aspect). The user knowledge decay seems a reasonable heuristics. Perhaps I am missing some novelty aspect, I'd appreciate the authors comments on this.
- It is unclear to me why other HPO methods "cannot take user feedback" at any time, especially if this is given in the form of "fixing some hyperparameter value" (aka point mass distribution). Even thinking about random search, at any time (i.e. after any number of draws/trials), one can replace any previously given distribution with an updated guess by the user and continue the process. This sounds to me a reasonable simple baseline that is not included in the paper
- similarly (although I am not familiar with the methods) I would assume that both BOPrO and muBO could be modified in a straightforward way to allow for anytime user belief update by warm-restarting (i.e. using all previously evaluated configurations as starting points). Is there any fundamental limitation that prevents this integration?

For me these last aspects pared by the low level of novelty I see at the moment make me lean toward rejection.

**Questions:**

- notation:
  - what's the meaning of p(f|D) in 3.1? when I read this notation I think of a distribution over functions, but rather since f here is given, probably the authors mean a distribution over function values (given hyperparameters)?
  - what is $\mathcal{S}$ and $\mathcal{K}$ (are they just generic collections)?
  - what's the measure of the integral in definition 3?
- if possible, I would like to see comparison with at least simple baselines such as RS with the same user belief integration used by IBO-HPC.

---

> ### Author Response · Authors · 2024-11-18
> **Rebuttal for nTfS**
>
> We thank the reviewer for the insightful feedback. Furthermore, we appreciate that our framework leveraging probabilistic circuits (PCs) in Bayesian optimization (BO)  is received as novel and well-justified.
>
> In the following, we would like to address the remaining concerns.
>
> All changes made in our manuscript during the rebuttal are marked as **blue** for reviewer nTfS and **green** for all reviewers.
>
> > W1.to me, the overall novelty value of this work is not immediately clear. On the one hand, it does look like to me that the work is a direct application of the general framework of BO with a surrogate model given by PC. The integration of user knowledge in the form of prior distribution seems fairly straightforward, but not necessarily unique to the proposed framework (although this work focuses on this aspect).
>
> While the integration of prior distributions in BO is not unique to our work (see our baselines [Hvarfner2023, Souza2021] ), IBO-HPC - to the best of our knowledge - is the first work leveraging PCs as a surrogate model, enabling the integration of user knowledge elegantly (in the form of values or distributions) at any time during optimization via conditional sampling. This means that user knowledge can simply be infused by conditioning the surrogate on the provided knowledge/feedback. Furthermore, we propose a novel selection policy for suggesting new configurations to be tested by conditioning the PC on the best obtained evaluation score, followed by sampling from the conditioned PC. This operation uses the tractability property and efficient conditional sampling from PCs. In contrast to many existing BO methods that require an inner-loop optimization of an acquisition function to suggest new configurations, our selection policy **does not** require an additional inner-loop optimization since we rely on tractable conditional sampling.
>
> We highlighted these aspects further in our revision (Sec. 1 & 3).
>
> > W2 It is unclear to me why other HPO methods "cannot take user feedback" at any time, especially if this is given in the form of "fixing some hyperparameter value" (aka point mass distribution).
>
> We agree that it is straightforward for model-free optimization methods like random search to incorporate user knowledge by redefining the distribution of the optimizer samples from a user prior. However, we consider model-based optimization (BO), where integrating user knowledge in optimization is not as straightforward since the information encoded in the surrogate model **and** the information encoded in the user knowledge must be incorporated when selecting new configurations.
>
> We acknowledge that an additional random search baseline with user priors is helpful for comparison. We added the baseline in our revision (see Appendix D3 and Fig. 2). It can be seen that IBO-HPC consistently outperforms RS with user priors on all tasks.
>
> > W3 Is there any fundamental limitation that prevents this integration?
>
> Warm-starting PiBO and BOPrO with previously evaluated configurations to incorporate user priors “during” optimization is an exciting idea. However, note that this strategy has two drawbacks: (1) The surrogate model must be retrained from scratch every time the user adjusts the provided knowledge, while this is not necessary with PCs given that we can provide different evidence without changing the model (PCs can answer to a wider range of probabilistic queries). If the popular Gaussian Process (GP) surrogate model is used in the BO loop, “fitting” the GP has a runtime complexity of $\mathcal{O}(n^3)$, which can get costly if the number of data instances $n$ grows and/or the user adjusts feedback frequently. (2) Applying the suggested strategy to PiBO and BOPrO would not solve the issue that neither method accurately reflects the user prior, as shown in Fig. 1 in our paper.
>
> > Q1 what's the meaning of p(f|D) in 3.1? when I read this notation I think of a distribution over functions, but rather since f here is given, probably the authors mean a distribution over function values (given hyperparameters)?
>
> In Sec. 3.1 $p(f | \mathcal{D})$ refers to a distribution over functions given a dataset D since this section introduces BO in general. For example, $p$ could be approximated by a Gaussian Process. This distribution is not to be confused with $p(\mathbf{\mathcal{H}} | F=f^*)$ in Sec. 3.2 that forms a distribution over hyperparameter configurations given $f^*$, which is an actual function value (the best value obtained so far).

---

> ### Author Response · Authors · 2024-11-18
> **Rebuttal for nTfS**
>
> > Q2 what is S and K (are they just generic collections)?
>
> As correctly anticipated by the reviewer, $\mathcal{S}$ and $\mathcal{K}$ are generic collections. $\mathcal{S}$ is the collection of surrogates, and $\mathcal{K}$ is the collection of possible user knowledge bases. We do not specify them further to ensure a rather general definition of interactive selection policies that do not rely on specific forms of surrogate models and/or user knowledge.
>
> > Q3 what's the measure of the integral in definition 3?
>
> Since we assume the selection policy to be a distribution over configurations, the integral in Definition 3 is defined over the probability measure $p(\mathbf{\Theta}, s_t, \mathcal{K})$.
>
> **References**
>
> [Hvarfner2023] PiBo: Augmenting acquisition functions with user beliefs for bayesian optimization. ICLR 2022.
>
> [Souza2021] Bayesian optimization with a prior for the optimum. ECML. 2021.

---

> ### Author Response · Authors · 2024-11-21
> **Discussion period ends soon**
>
> Dear Reviewer,
>
> We hope to have resolved all your concerns. If you have any further comments, we will be happy to address them before the rebuttal period ends. If there are none, then we would appreciate it if you could reconsider your rating.
>
> Regards,
>
> Authors

---

> ### Author Response · Authors · 2024-11-25
> **End of Discussion Period**
>
> Dear Reviewer,
>
> We hope to have resolved all your concerns. If you have any further comments, we will be happy to address them before the discussion period ends. If there are none, then we would appreciate it if you could reconsider your rating.
>
> Regards,
>
> Authors

---

> > ### Author Response · Authors · 2024-12-01
> > **End of Discussion Phase**
> >
> > Dear Reviewer,
> >
> > We hope we have resolved all your concerns. If you have any additional comments, we would be glad to address them before the discussion phase ends. Otherwise, we would appreciate it if you could reconsider your rating.
> >
> > Best regards,
> >
> > The Authors

---

> ### Comment · Reviewer_nTfS · 2024-12-01
>
> I thank the authors for their reply and appreciate the additional experiments with random search.
> I increased the score for soundness to 3 do to the additional experiments. However, I am still overall unconvinced by the contribution of this work as I explained in my review and I also believe that the work would benefit from some more polishing, including in the notation and presentation of results.
>
> For instance, regarding my question about $p(f|\mathcal{D}_n)$, the function $f$ is introduced in line 174 as the underlying black box (which is unknown, but given). Now, when you have a probability distribution over functions whose domain *includes* $f$, it certainly makes sense to compute $p(f|\mathcal{D}_n)$; In other words, following the usual conventions, $p(f|\mathcal{D}_n)\in\mathbb{R}^+$ indicates the mass of the distribution on $f$ if $p$ is discrete or, I suppose, the value of the density function at $f$ if $p$ is continuous. I do not believe that any of the two is what the author meant in the passage. I would also encourage the author in framing $\mathcal{S}$ and $\mathcal{K}$ to provide meaning to these notations (or if not possible, perhaps remove them).
> Now, these are certainly not terminal flaws of the paper. However, passing over it again, I still find it unnecessarily difficult to navigate it, and I am unconvinced that the changes made with the revision actually made paper clearer.
>
> I believe that focusing the discussion on pinning down the advantages of PCs and the proposed methods, convincingly and analytically, if possible,  making the case for its usage would greatly strengthen the work.

---

> ### Author Response · Authors · 2024-12-02
>
> We thank the reviewer for participating in the discussion and increasing the soundness score due to our additional experiments!
>
> > However, I am still overall unconvinced by the contribution of this work
>
> To make our contributions even clearer, we rephrased the last paragraph of the Introduction, which now reads:
>
> "We make the following contributions: (1) We introduce a novel BO method named IBO-HPC that does not require an additional inner-loop optimization of an acquisition function and enables a direct, flexible, and targeted incorporation of user knowledge into the selection policy s.t. the user knowledge is precisely reflected in the selection policy as specified by the user. Both are achieved through tractable conditional sampling of PCs. (2) We formally define a notion of interactive policy in BO, and show that IBO-HPC conforms to this notion and is guaranteed to reflect user knowledge as provided in the optimization process. (3) We provide a theoretical convergence analysis of IBO-HPC, showing that it converges proportionally to the expected improvement. (4) We provide an extensive empirical evaluation of IBO-HPC showing that it is competitive with strong HPO and NAS baselines without user interaction and outperforms them when leveraging user knowledge."
>
> **N.B.**: We cannot apply these and the following changes to the uploaded PDF given the (past) deadline for applying changes to it. Thus, we will apply these edits to the camera-ready.
>
> > I believe that focusing the discussion on pinning down the advantages of PCs and the proposed methods, convincingly and analytically, if possible, making the case for its usage would greatly strengthen the work.
>
> Multiple advantages come with IBO-HPC and the use of PCs in the BO framework. Let us elaborate on this in a bit more detail.
>
> IBO-HPC has appealing properties regarding flexibly infusing user knowledge into the optimization process. As discussed in the Introduction (see Fig. 1) and Appendix A of the current revision available on OpenReview, existing approaches that allow users to specify a prior distribution over the search space fail to reflect the provided user prior effectively. Consider the real-world benchmark JAHS in Appendix A. We provided the same user prior to PiBO, BOPrO, and IBO-HPC. We found that BOPrO and PiBO sample configurations of the hyperparameter $R$ from a biased distribution that does not match with the given user prior over $R$. We attribute this behavior to the non-trivial interaction between the user prior and the acquisition function used by PiBO and BOPrO to select the next configuration to be evaluated.
>
> In contrast, IBO-HPC accurately reflects the given user prior (see Fig. 5(b) in Appendix A). We achieve this by leveraging PCs' tractable conditioning and sampling mechanisms [1--5]. Instead of weighting an acquisition function with prior, we **directly condition** the surrogate model (the PC) with the provided user knowledge. Thus, users can infuse their knowledge in a much more targeted and direct way into the optimization process because when we sample from the conditional PC to generate new configurations to be tested, the conditioning ensures that the provided user knowledge persists in the new samples as provided by the user.  This process is described in detail in lines 266-285 in our manuscript.
>
> **References**
>
> [1] Poon et al., Sum-Product Networks: A New Deep Architecture. UAI 2011.
>
> [2] Peharz et al., On Theoretical Properties of Sum-Product Networks. AISTATS 2015.
>
> [3] Peharz et al., On the Latent Variable Interpretation in Sum-Product Networks. IEEE TPAMI 2017
>
> [4] Molina et al., Mixed Sum-Product Networks: A Deep Architecture for Hybrid Domains. AAAI 2018.
>
> [5] Choi et al., Probabilistic Circuits: A Unifying Framework for Tractable Probabilistic Models. UCLA 2020.

---

> ### Author Response · Authors · 2024-12-02
>
> Besides the accurate reflection of user knowledge in the selection policy, the tractable conditioning and sampling procedures of PCs allow us to obtain new promising configurations without an additional inner-loop optimization of an acquisition function. This is achieved by conditioning the PC on the best evaluation score obtained so far, denoted $\mathbf{f}^*$ (along with optional user knowledge). Since the surrogate PC $s(\mathcal{H}, F)$ is a joint distribution over configurations and evaluations that maximizes the likelihood of the observations in $\mathcal{D}$, conditioning on $\mathbf{f}^*$ yields a conditional distribution over the search space $\mathcal{H}$ s.t. configurations that perform similar to $\mathbf{f}^*$ are likely to be sampled. Thus, it will likely sample a promising new configuration candidate, and we do not require inner-loop optimization. Instead, we can perform sampling from the conditional PC (which is linear in the size of $s$ and thus efficient).
>
> To highlight these aspects, we added the following paragraph below Eq. 2:
>
> "Note that the conditioning is the driving mechanism for both accurate incorporation of user knowledge and the inner-loop optimization-free selection of candidate configurations. With conditioning $s$ on user knowledge, we ensure that the provided user knowledge is precisely reflected in the next candidates; conditioning $s$ on $f^*$ ensures that only promising configurations are highly likely to be selected."
>
> We empirically demonstrate the more accurate incorporation of user knowledge and overall effectiveness of the optimization-free selection policy via conditional sampling in Fig. 2 and Fig. 6, where IBO-HPC **consistently outperforms** PiBO and BOPrO in **6/7 cases** when beneficial user knowledge is provided.
>
> Further, the tractable conditioning and sampling of PCs allows IBO-HPC to handle multiple user interactions straightforwardly and naturally. If users recognize that the provided knowledge is sub-optimal, for example, they can easily provide their revised knowledge by conditioning the PC on a different set of hyperparameters and values/distributions. The direct conditioning of the surrogate ensures that the provided user knowledge is reflected accurately when new configuration candidates are selected. In Fig. 3, we empirically show that IBO-HPC **successfully handled multiple, even contradicting, user interactions well**, and consistently outperforms PiBO and BOPrO on 4/5 tasks even under these challenging conditions.
>
> We rephrased lines 469-472 to discuss and better point out the advantages of PCs regarding incorporating user knowledge. Thus, we replaced
>
> "These results show that IBO-HPC’s policy reflects user beliefs accurately and leverages provided knowledge effectively by sampling configurations that are promising according to both, user belief and the surrogate HPC."
>
> by
>
> "The results demonstrate that the conditional sampling-based selection policy accurately represents given user knowledge, which allows IBO-HPC to converge to better solutions than our baselines. Also, it shows that the selection policy effectively balances provided user knowledge and knowledge encoded in the surrogate since sampled configurations are promising (i.e., high conditional likelihood) according to both user belief and the surrogate HPC."

---

> ### Author Response · Authors · 2024-12-02
>
> In Sec. 4.1 and Fig. 2, we empirically show that IBO-HPC is on par or outperforms existing BO algorithms when no user knowledge is provided. Further, in Fig. 4(a), we show that IBO-HPC achieves considerable runtime improvements over existing strong baselines such as SMAC. This demonstrates that IBO-HPC is faster than SMAC due to its more efficient selection policy and still achieves competitive results.
>
> To clarify the advantages of IBO-HPC and using PCs, we rephrased lines 515-517. Hence, we replaced
>
> "IBO-HPC is considerably faster than SMAC in 4/5 cases, especially in larger search spaces. We attribute this to the efficiency of PCs and of our selection policy (i.e., conditional sampling)."
>
> by
>
> "Paired with the results from **(Q1)**, this clearly demonstrates that our selection policy leveraging the conditional sampling of PCs not only achieves competitive or even better results than BO methods relying on a selection policy requiring an additional inner-loop optimization; it also shows that our selection policy achieves a remarkable improvement in efficiency due to the efficient conditional sampling mechanism of PC."
>
> To further verify our conclusion about the effectiveness of our selection policy, we empirically show that IBO-HPC effectively trades off exploration and exploitation during optimization (see Fig. 14 and 15 in Appendix D.3). The plots show the sampling variance of IBO-HPC in each iteration averaged over 20 runs on the JAHS benchmark. The sampling variance (roughly) monotonically decreases with increasing iterations (in almost all cases), demonstrating that IBO-HPC focuses on exploration in early iterations while it focuses on exploitation of the collected knowledge in later iterations.
>
> > On $p(f | \mathcal{D}_n)$ and Notation
>
> Thank you for pointing this out again. We agree that the way it is currently written can be confusing. We rephrased the sentence about $p(f | \mathcal{D})$ accordingly. We replaced
>
> "Given a set $\mathcal{D}_n$ of observations that correspond to the configurations with associated evaluations $(\theta_j, f(\theta_j)) j = \{1, ..., n\}$, the surrogate $s$ aims to induce a distribution over functions $p(f|\mathcal{D}_n)$."
>
> with
>
> "Given a set $\mathcal{D}_n$ of observations that correspond to the configurations with associated evaluations $(\theta_j, f(\theta_j)) j = \{1 \ldots n}$, the probabilistic surrogate $s$ aims to approximate $f$ as closely as possible given observations $\mathcal{D}_n$."
>
> For further clarity, we now denote policies as $g$ instead of calling policies $p$ in Def. 2 and 3 to avoid naming collisions with the joint distribution $p(\mathcal{H}, F)$. We also followed the suggestion of the reviewer to remove $\mathcal{S}$ since it added unnecessary overhead to our notation. For $\mathcal{K}$, we added a brief discussion in the Appendix (see below).
>
> Lastly, to clarify the meaning of $p$ and $s$, we rephrased a sentence in line 263. Therefore, we replaced
>
> "After evaluating each sampled θ we obtain a set D of pairs (θ, fθ(x)) and fit a HPC s estimating the joint distribution p(H, F), where H is the set of hyperparameters and F is a random variable representing the evaluation score (Line 7)."
>
> by
>
> "After evaluating each sampled $\theta$ we obtain a set $\mathcal{D}$ of pairs $(\theta, f_{\theta}(\mathbf{x}))$. We fit an HPC $s(\mathcal{H}, F)$ that models the observations $\mathcal{D}$ as a joint distribution over hyperparameters $\mathcal{H}$ and evaluation score $F$ by maximizing the likelihood of $\mathcal{D}$ (Line 7). Both hyperparameters $\mathcal{H}$ and evaluation score $F$ are treated as random variables and assumed to follow a ground truth distribution $p(\mathcal{H}, F)$ that is approximated by $s$."

---

> ### Author Response · Authors · 2024-12-02
>
> > I would also encourage the author in framing $\mathcal{S}$ and $\mathcal{K}$ to provide meaning to these notations (or if not possible, perhaps remove them).
>
> Following the suggestion of the reviewer, we removed $\mathcal{S}$ since it added unnecessary overhead to our notation. Since $\mathcal{K}$ is an integral part of Def. 1 and 2, we kept $\mathcal{K}$ and added a brief discussion on $\mathcal{K}$ in the Appendix. In addition to our brief specification of  $\mathcal{K}$ in the main paper (see line 270), we added a discussion on possible instantiations of $\mathcal{K}$ in the Appendix:
>
> "In general, $\mathcal{K}$ refers to a set of possible specifications a user can provide to guide the optimization process. In our case, all elements in  $\mathcal{K}$ are assumed to be either in the form of user priors $q(\hat{\mathcal{H}})$ or assignments of hyperparameters to a certain value, i.e., $\hat{\mathcal{H}} = \hat{\mathbf{h}}$. Here, $\hat{\mathcal{H}}$ refers to a subset of hyperparameters that define the search space. However, other forms of user knowledge are possible. For example, users could also specify beliefs about possible correlations between hyperparameters or between hyperparameters and the evaluation score. We decided to **not** restrict $\mathcal{K}$ to generalize the notion of interactive and feedback-adhering interactive policies to a broad set of types of user knowledge.
>
> We hope we clarified the reviewer's remaining concerns with our response. Considering our changes, we kindly ask the reviewer to reconsider their score.

---

> > ### Author Response · Authors · 2024-12-02
> > **End of Discussion Phase**
> >
> > Dear Reviewer,
> >
> > We hope we have resolved all your concerns. If you have any additional comments, we would be glad to address them before the discussion phase ends. Otherwise, we would appreciate it if you could reconsider your rating.
> >
> > Best regards,
> >
> > The Authors

---

> ### Comment · Reviewer_nTfS · 2024-12-03
>
> Dear authors,
>
> thanks for your reply. I appreciate your responsiveness and I believe your proposed changes may be beneficial for a revised submission (which anyway should be revaluated with new eyes). There are a lot of changes, some of which (as I mentioned) I am not sure they actually improve the whole presentation.
>
> Please, also stop asking me to revise my score. I would for sure revise my score if I deemed it justified (and I indeed raised my soundness score), but continuous requests are not helpful.
>
> Thanks

---

> > ### Author Response · Authors · 2024-12-04
> >
> > We thank the reviewer again for the feedback and appreciate their participation in the discussion. We also appreciate their soundness score reassessment.
> >
> > We would like to clarify why we asked for a reconsideration of the score. Following the reviewer’s feedback on the presentation of our work, we have made appropriate changes to improve it as suggested. We acknowledge that, e.g., the simplification of notation and the discussion on the advantages of using PC in the BO loop improved our paper. Thus, we made the changes as described in our latest response. We asked for reconsideration because the reviewer received the remaining concerns as being “certainly not terminal flaws of the paper”. Furthermore, since reviewers are burdened and might forget to adjust their score, we kindly remind them to reconsider it. Moreover, given that our latest proposed changes have been received as “may be beneficial for a revised submission”, it is still unclear which specific point/concern is left unresolved and on which changes the reviewer is unsure about their improvement.

---

### Official Review · Reviewer_uXRu · 2024-11-03

**Soundness:** 3
**Presentation:** 3
**Contribution:** 3
**Rating:** 6
**Confidence:** 2

**Summary:**

The paper leverages probabilistic circuits to enable user interaction in Bayesian optimization during hyperparameter optimization.

**Strengths:**

The paper is well-written and the methodology seems to be original.

**Weaknesses:**

Definition 3 would be clearer if it included an explanation of why the second condition is necessary and how it ensures the policy accurately represents the user's knowledge.

**Questions:**

Please provide more reasoning for the second condition on Definition 3.

---

> ### Author Response · Authors · 2024-11-18
> **Rebuttal for uXRu**
>
> We thank the reviewer for the review and appreciate that our work is perceived as original and clearly written. Please note that we marked the edits in the manuscript made by following the feedback provided by other reviews in color (green, blue, and pink).
>
> > Q1 Please provide more reasoning for the second condition on Definition 3.
>
> Let us clarify the two conditions in Definition 3: An interactive selection policy should be efficacious (i.e.,$p(\mathbf{\Theta}, s_t, \mathcal{K}) \neq p(\mathbf{\Theta}, s_t, \emptyset)$ ) to ensure that the given user knowledge has a guaranteed effect on the sampling of new configurations to be tested. If the user knowledge is provided as a distribution for a set of hyperparameters H at iteration t, the policy that samples new configurations to be tested should be governed by the provided prior $q$ at iteration $t+1$, i.e., $\int_{\mathcal{H} \setminus \hat{\mathcal{H}}} p(\mathbf{\Theta}, s_t, \mathcal{K}) = q(\hat{\mathcal{H}})$. This ensures that IBO-HPC adheres to the provided user knowledge at iteration $t+1$.

---

> ### Author Response · Authors · 2024-11-21
> **Discussion period is ending soon**
>
> Dear Reviewer,
>
> We hope to have resolved all your concerns. If you have any further comments, we will be happy to address them before the rebuttal period ends.
>
> Regards,
>
> Authors

---

> ### Author Response · Authors · 2024-12-01
> **End of Discussion Phase**
>
> Dear Reviewer,
>
> We hope we have resolved all your concerns. If you have any additional comments, we would be glad to address them before the discussion phase ends.
>
> Best regards,
>
> The Authors

---

### Official Review · Reviewer_cr4S · 2024-11-03

**Soundness:** 2
**Presentation:** 2
**Contribution:** 2
**Rating:** 3
**Confidence:** 5

**Summary:**

The paper introduces a hyperparameter optimization approach by incorporating user feedback directly into the optimization process through a Bayesian framework using probabilistic circuits (PCs). The proposed method, Interactive Bayesian Optimization via Hyperparameter Probabilistic Circuits (IBO-HPC), aims to overcome limitations in existing HPO by allowing dynamic incorporation of expert knowledge at any point during the optimization. The method leverages PCs for tractable inference and sampling, enabling users to provide valuable insights interactively and generate configurations adhering to their feedback.

**Strengths:**

- The paper addresses the important topic of interactive hyperparameter optimization.
- It presents a novel approach by using probabilistic circuits as a surrogate model.

**Weaknesses:**

1. In comparison to prior methods that incorporate static priors, this paper introduces flexibility in the timing of knowledge integration. However, the study of human knowledge remains restricted to optimal solutions or their distributions. It would be beneficial if the introduction provided some application scenarios to underscore the importance of addressing this limitation. Additionally, the primary distinction of this method from prior work lies in its approach to knowledge-based sampling, yet the proposed sampling technique does not seem to offer substantial theoretical advancements, such as analyzing the impact of the sampling method and prior integration on the convergence bounds of the algorithm.

2. Lines 277-283: The sampling method first draws a set of candidate points based on a prior distribution, then further samples from these points. The paper does not discuss how this approach ensures accuracy in approximating Equation 2. Additionally, it is unclear why the two product terms in the equation are approximated through separate sampling steps. Could you provide a theoretical justification evidence for the accuracy of their approximation method and give an explanation for why the two-step sampling process was chosen over alternatives.

3. The paper does not clearly explain how sampling is conducted in Algorithm 1, line 13, or how the value of 𝑠 is calculated in line 14.

4. The method for constructing the PC model using data and how to perform sampling with this model are not well-explained. What are the advantages of the PC model compared to other models? Is there any drawback of it?

5. Existing prior-based methods can be applied to the application scenario in this work. Could you provide a more explicit comparison of the method to existing prior chased method (for example, BOPrO and πB) highlight the key differences and advantages. The paper does not discuss related work in interactive hyperparameter optimization. Provide a discussion of related work in interactive hyperparameter optimization would help contextualize the contribution.

6. The rationale behind the time and quality of knowledge used in the experimental setup is unclear. What real-world applications might benefit from this approach?

7. The test problems presented are limited, making it difficult to verify the method’s applicability across a diverse range of problems. Furthermore, comparisons with more recent prior-based HPO methods, such as Priorband [1] and PFNs4BO [2], as well as HPO methods like DPL [3] and Hyper-tune [4], are lacking. For the problems tested, you could refer to the test cases used in these methods.
[1] Mallik N, Bergman E, Hvarfner C, et al. Priorband: Practical hyperparameter optimization in the age of deep learning[J]. Advances in Neural Information Processing Systems, 2024, 36.
[2] Müller S, Feurer M, Hollmann N, et al. Pfns4bo: In-context learning for bayesian optimization[C]//International Conference on Machine Learning. PMLR, 2023: 25444-25470.
[3] Kadra A, Janowski M, Wistuba M, et al. Scaling laws for hyperparameter optimization[J]. Advances in Neural Information Processing Systems, 2024, 36.
[4] Li Y, Shen Y, Jiang H, et al. Hyper-tune: towards efficient hyper-parameter tuning at scale[J]. Proceedings of the VLDB Endowment, 2022, 15(6): 1256-1265.
8. The paper could be improved with clearer explanations. For example：there are many equations $p(\boldsymbol{\Theta}, st, K) ̸= p(\boldsymbol{\Theta}, st, \phi)$, $\int_{\mathcal{H} \backslash \hat{\mathcal{H}}} p\left(\boldsymbol{\Theta}, s_t, \mathcal{K}\right)=q(\hat{\mathcal{H}})$. After defining these equations, it is essential to explain why satisfying these conditions supports the corresponding conclusions.

**Questions:**

1. The example in Figure 1 is confusing: "Users can guide the HPO algorithm with their knowledge (here, N = 16 and W = 3), thus considerably increasing convergence speed and the quality of the final solution by focusing the optimization on remaining hyperparameters." In what cases would users accurately know the precise value of certain hyperparameters during optimization? Why wouldn’t they fix these parameter values at the start of optimization? How does this work relate to preference-based optimization algorithms?

2. "Moreover, these approaches might not reflect user knowledge precisely as defined in the prior due to the non-linear integration of the priors in the acquisition function. For example, in Fig. 1 (Right), the configurations selected by both BOPrO (Souza et al., 2021) and πBO (Hvarfner et al., 2022) during the first 20 iterations of optimization remarkably deviate from the given user prior." How did the authors define "remarkably deviate"? If I understand correctly, these algorithms aim to move towards the region defined by the prior, rather than sampling exactly according to the prior distribution, since the user-provided prior might not be accurate. In this context, the example in Fig. 1 (Right) does not seem to support the authors’ claim.

3. What is the difference between "user knowledge" and "feedback" in Definitions 2 and 3?

4. There are several symbols in Definition 3, such as "an interactive policy $p$," "$p(\Theta, st, K)$," and "call $p$ feedback." I find these symbols confusing and difficult to interpret.

---

> ### Author Response · Authors · 2024-11-18
> **Rebuttal for cr4S**
>
> We thank the reviewer for the insightful and extensive feedback. We appreciate that the flexible incorporation of user feedback during optimization by leveraging probabilistic circuits (PCs) is perceived as important and novel. In the following, we address the raised concerns.
>
> All changes made in our manuscript during the rebuttal are marked as **pink** for reviewer cr4S and **green** for all reviewers.
>
> > W1, W6  However, the study of human knowledge remains restricted to optimal solutions or their distributions. [...] The rationale behind the time and quality of knowledge used in the experimental setup is unclear. What real-world applications might benefit from this approach?
>
> We believe there is a misunderstanding of our experimental setup. Let us provide some more explanation on our setup:
>
> In our experiments, we do not only consider optimal solutions as given user knowledge. Instead, **we also consider worst-case scenarios** in which users provide strongly harmful knowledge, i.e., the user specifies a point mass for a large subset of hyperparameters corresponding to poorly performing configurations (see Fig. 3 pink line and Sec. 4.2). Also, we provided IBO-HPC with an alternation of contradicting feedback (i.e., alternating between useful and harmful user knowledge; see Fig. 3 brown line).  With this, we aimed to analyze the behavior of IBO-HPC under extreme and challenging conditions. Our results show that IBO-HPC handles these extreme cases well due to our effective selection policy and the recovery mechanism.
>
> Overall, the experimental evaluation aimed to evaluate IBO-HPC on a broad set of tasks (NAS and HPO) and diverse user behavior (only helpful knowledge, alternating useful and harmful knowledge, only harmful knowledge)  that might occur in the real world. For example, a user might provide a distribution over certain hyperparameters early in the optimization that turn out to be a suboptimal guess. Then, it is beneficial if the user can withdraw the provided knowledge or replace it with a refined distribution while keeping the knowledge about the objective accumulated in the surrogate for the next iterations (i.e., no restart from scratch). Given the flexible use case and our strong empirical evidence showing that IBO-HPC excels in the challenging scenarios above, we believe that next to the real-world state-of-the-art benchmarks we employed, IBO-HPC is successfully applicable also to a broader set of real-world problems.
>
> > W1 [...] analyzing the impact of the sampling method and prior integration on the convergence bounds of the algorithm.
>
> We agree that a theoretical analysis of the convergence properties of IBO-HPC with and without user knowledge is beneficial to better understand our method's behavior and possible limitations. Therefore, we added a rigorous convergence analysis in our paper, showing that IBO-HPC (1) is a global optimizer, (2) has a convergence rate depending on the Expected Improvement guaranteeing convergence to local optima under standard conditions, and (3) benefits from helpful user knowledge in terms of convergence speed. Refer to Appendix B.2 and B.3 for details.
>
> Our theoretical results align with our strong empirical evidence that IBO-HPC converges to good solutions quickly, especially if helpful user knowledge is provided.

---

> ### Author Response · Authors · 2024-11-18
> **Rebuttal cr4S**
>
> > W2 Could you provide a theoretical justification evidence for the accuracy of their approximation method (Eq. 2) and give an explanation for why the two-step sampling process was chosen over alternatives.
>
> We thank the reviewer for raising this question which gives us the chance to better clarify our approach. Let us start by noting that the accuracy of our method depends on two components: (1) The accuracy of the learned PC s and (2) the accuracy/unbiased-ness of our sampling method.
>
> Since we use LearnSPN to infer the structure and weights of the PC given data $\mathcal{D}_i$  in the $i$-th iteration, the PC s will be an SPN that locally maximizes log-likelihood, as stated in Proposition 1 in [Gens2013]. This means that there is no other SPN in the space of the learnable SPNs via LearnSPN that achieves a better log-likelihood given the data. Hence, as long as the ground truth distribution $p$ (or a good approximation of it) is representable by an SPN, we can recover $p$ with arbitrarily small error with iterations $T \rightarrow \infty$. Now on (2): We sample from two distributions, the prior $q$ and the conditional distribution  $s(\mathbf{\mathcal{H}} | f^*, \mathcal{H}_i = h_i)$ over configuration-evaluation pairs. Here, $s$ refers to the HPC representing a joint distribution over hyperparameters and evaluation scores, while $\mathcal{H}_i = h_i$ refers to some hyperparameter(s) set to $h_i \sim q$.
> Assuming $q$ is a tractable distribution (e.g., a parametric one such as an isotropic Gaussian), sampling is immediate and not biased (i.e., performed via simple transformations such as the Box-Muller transform). Note that the assumption on $q$ being a tractable (and relatively simple) distribution can be made safely since providing highly complex distributions as user knowledge is hard to do for most users.
> Computing the conditional $s(\mathbf{\mathcal{H}} | f^*, \mathcal{H}_i = h_i)$ is tractable and accurate since $s$ is a PC [Choi2020; Sec. 4.1]. Note that conditioning a PC yields a valid PC again (specifically in the form of an SPN when obtained using LearnSPN). Then, PC sampling is performed top-down by sampling from the simple categorical variables represented by the sum nodes and then from the selected univariate leaves. Thus, the process is tractable (linear in the circuit size) and not biased by further operations or assumptions [Choi2020]. Thus, we conclude that the approximation in Eq. 2 is accurate in the limit $N, T \rightarrow \infty$. For more details, refer to Appendix B.4.
>
> We chose this two-step sampling as it can be implemented easily and works well in practice as demonstrated in our empirical results.
>
> > W3 The paper does not clearly explain how sampling is conducted in Algorithm 1, line 13, or how the value of 𝑠 is calculated in line 14.
>
> The sampling procedure in Algorithm 1 (line 13) is explained in Appendix C and follows standard conditional sampling in PCs. Since PCs are a rooted directed acyclic graph (DAG), sampling can be implemented as a recursive top-down operation: At each sum node, a child is sampled according to the normalized weights associated with the sum node, and at each product node, all children are selected to be sampled from. This way, the sampling procedure traverses the PC top-down, reaching a subset of leaf nodes. Since all leaf nodes are tractable distributions (e.g., Gaussians), one can simply sample from all selected leaves to obtain a new sample. Conditional sampling can be achieved by first conditioning the PC on some evidence (see Appendix C for details).
>
> Regarding line 14 of Algorithm 1: s is not a computed value; it refers to PC learned in line 6 using LearnSPN. The step in line 14 selects the configuration most likely to achieve $f^*$, i.e., the highest likelihood of yielding a good evaluation score.

---

> ### Author Response · Authors · 2024-11-18
> **Rebuttal for cr4S**
>
> > W4 The method for constructing the PC model using data and how to perform sampling with this model are not well-explained. What are the advantages of the PC model compared to other models? Is there any drawback of it?
>
> Thank you for pointing this out; we agree that explaining the learning of PCs improves the clarity of our paper. To learn the PC from data, we use a variant of LearnSPN [Poon2011, Gens2013] designed for hybrid domains [Molina2018]. This is a well-known and widely employed greedy recursive algorithm to both learn the PC structure and estimate its parameters. Intuitively, LearnSPN alternates between clustering instances in the dataset and finding independencies among features. To create a sum node, a clustering algorithm splits data instances into subsets, with weights set by the proportion of instances in each subset. For a product node, independence tests split the random variables in the data slice into sub-groups. We added a brief description of LearnSPN in Appendix C (also sampling is explained in Appendix C, see response to W3).
>
> In contrast to other models used as surrogates in BO, PCs exhibit multiple advantages and unique properties since they represent joint distributions over a set of random variables instead of modeling a mere conditional distribution  [Poon2011, Choi2020]. Furthermore, PCs can perform *exact tractable* marginalization and conditioning w.r.t. arbitrary subsets of random variables as well as tractable sampling. In the context of (interactive) BO, we model a distribution over the joint space of hyperparameters **and** evaluation scores instead of modeling a distribution over evaluation scores **given** a hyperparameter configuration. This modeling aspect of PCs paired with the tractable conditioning and sampling properties allows us to define a novel selection policy that (1) does not require an inner loop optimization of an acquisition function to identify promising configurations to be evaluated next and (2) can incorporate user knowledge naturally and dynamically via conditioning the surrogate model. Regarding (1), at iteration t, we select new configurations by conditioning the surrogate PC with the best score obtained so far and sample from the resulting conditional distribution over the hyperparameter search space. This makes our selection policy fully data-driven.
>
> One drawback of PCs is that they can be less sample efficient than e.g., Gaussian Processes (GPs) because PCs impose fewer assumptions on the data space than GPs do (e.g., no Gaussianity assumption, no kernel required). However, we found in our experiments IBO-HPC converges at similar rates to widely used BO methods relying on GPs or random forests as surrogates. Also, PCs scale better than GPs since inference time is linear in the circuit size, while for GPs, inference time is cubic in the number of data points.
>
> > W5.1 [...] comparison of the method to existing prior chased method (for example, BOPrO and πB) highlight the key differences and advantages.
>
> The key difference between existing prior-guided HPO methods like BOPrO and PiBO is that IBO-HPC leverages PCs approximating the full joint distribution over the joint search- and evaluation space. Instead, BOPrO and PiBO use traditional regressors such as random forests (RFs) to approximate the objective function. PCs offer exact inference tractable marginalization and (conditional) sampling. Thus users can provide feedback at any point during optimization by simply conditioning the PC on their knowledge instead of shaping the acquisition function by a pre-defined prior. Fig. 1 in our paper shows that this reflects user knowledge much more accurately than existing methods like BOPrO and PiBO. Also, our empirical evaluation shows that IBO-HPC outperforms BOPrO and PiBO, given the same user knowledge, indicating that IBO-HPC incorporates user feedback more effectively.
>
> Another key difference is that IBO-HPC’s selection policy is fully data-driven, relying on conditional sampling. This circumvents the necessity of performing an inner loop optimization of a (user prior weighted) acquisition function to select new configurations.
>
> We made the key differences and advantages of IBO-HPC clearer in the revision (Sec. 1, 2, 3).

---

> ### Author Response · Authors · 2024-11-18
> **Rebuttal for cr4S**
>
> > W5.2 The paper does not discuss related work in interactive hyperparameter optimization. Provide a discussion of related work in interactive hyperparameter optimization would help contextualize the contribution.
>
> We briefly discuss PiBO and BOPrO as two of our main baselines in our related work section (lines 141-150). We also added Priorband in our related work as we use it as an additional baseline, as suggested by the reviewer.
>
> > W7 The test problems presented are limited, making it difficult to verify the method’s applicability across a diverse range of problems. Furthermore, comparisons with more recent prior-based HPO methods, such as Priorband [1] and PFNs4BO [2], as well as HPO methods like DPL [3] and Hyper-tune [4], are lacking. For the problems tested, you could refer to the test cases used in these methods.
>
> Our experimental protocol follows standard practices [Lindauer2020, Bischl2023], and we evaluated IBO-HPC against 5 strong baselines on 5 different tasks across 3 real-world popular benchmarks (NAS-101, NAS-201, JAHS), thus, covering a diverse set of tasks and search spaces. We further compared IBO-HPC to our baselines on the HPO-B benchmark to further highlight IBO-HPC’s capabilities, including PriorBand as an additional baseline (see Appendix D.3). Also, we evaluated PriorBand on JAHS, NAS101 and NAS201 (see Fig. 2). However, PriorBand is a multi-fidelity HPO method. Thus, compatibility with IBO-HPC and other baselines is limited. Our results show that IBO-HPC consistently outperforms PriorBand on all tasks.
>
> Also, we are evaluating PFNs4BO on our selected tasks and try to add the results before the end of the discussion phase. Due to time constraints set by the discussion phase, we did not evaluate PriorBand on cases with harmful user feedback. However, since PriorBand is based on PiBO, we expect similar behavior. The results will be included in the camera-ready version.
>
> After carefully considering the suggestion of including DPL and Hyper-Tune as baselines, we are unsure whether these are appropriate baselines and which insights the reviewer expects to obtain from them. The reason is that DPL and Hyper-Tune are multi-fidelity methods with no explicit mechanism to incorporate user knowledge. Thus, the benefit of comparing to DPL is unclear to us.
>
> Hence, we kindly ask the reviewer to provide more details on why comparing IBO-HPC to the mentioned methods would be beneficial.
>
> > W8 The paper could be improved with clearer explanations.
>
> Thank you for pointing this out. We added more detailed explanations of our equations in our revision.
>
> > Q1.1/W1  In what cases would users accurately know the precise value of certain hyperparameters during optimization? Why wouldn’t they fix these parameter values at the start of optimization? It would be beneficial if the introduction provided some application scenarios to underscore the importance of addressing this limitation.
>
> This is an important point to clarify for the applicability of our method. We highlight that IBO-HPC allows users to specify distributions over values besides fixing precise values as user knowledge. In practice, however, users sometimes cannot specify a concrete prior distribution over hyperparameter values. For example, it is not straightforward to specify a joint distribution over the depth of a neural network and the learning rate. However, a user might know that deeper networks benefit from higher learning rates since the gradient signal is stronger in deeper layers. Thus, users might have a set of configurations they would like to test without starting the HPO run from scratch every time. Also, user knowledge might be **unknown before** optimization and users might develop an intuition **during optimization** on which hyperparameter’s values could work well when seeing actual evaluations. Since our method allows users to adjust the provided feedback at any iteration, users can quickly explore the effect of fixing some hyperparameters to specific values during optimization without the need to restart the optimization. We give a working example in Appendix E, highlighting how IBO-HPC works in such scenarios. To note also that when users set a subset of hyperparameters to a specific value and IBO-HPC samples new configurations to be tested, all the information about the objective function encoded in the surrogate model (the PC) is retained and (re)used. This can significantly decrease the time required for HPO, as shown in Fig. 4(b).
>
> We modified our  motivating example in our Introduction accordingly.

---

> ### Author Response · Authors · 2024-11-18
> **Rebuttal for cr4S**
>
> > Q1.2 How does this work relate to preference-based optimization algorithms?
>
> Thank you for this interesting question. In preference-based optimization such as [Giovanelli2024], user knowledge is incorporated to identify an objective function that is not easily computable. For example, preference-based optimization can be used to identify the Pareto-front in multi-objective optimization based on expert knowledge. Thus, preference-based optimization typically focuses on shaping the objective function/the evaluation space leveraging human preferences. In contrast, our work considers how human knowledge can be incorporated to guide a Bayesian optimization algorithm through the search space more effectively.
>
> > Q2.1 How did the authors define "remarkably deviate"?
>
> Since Fig. 1 is a motivational example, we do not provide a formal definition of “remarkable deviate”. However, by inspecting the learned density estimate used to plot Fig. 1, we find the following (approximate) deviations between the prior and the values actually sampled for R by different methods:
>
> - user prior: mean=1, std=0.5
>
> - PiBO samples: mean=0.5, std=1.1
>
> - BOPrO samples: mean=1.05, std=0.24
>
> - IBO-HPC samples: mean=1.09, std=0.49
>
>
> This shows that both, PiBO and BOPrO are significantly off w.r.t. the mean or the standard deviation, hence supporting our claim that PiBO and BOPrO do not accurately reflect the user prior in early iterations.
>
> > Q2.2 If I understand correctly, these algorithms aim to move towards the region defined by the prior, rather than sampling exactly according to the prior distribution, since the user-provided prior might not be accurate. In this context, the example in Fig. 1 (Right) does not seem to support the authors’ claim.
>
> This is an interesting point raised! However, we believe there is a slight misunderstanding about incorporating user priors in BO. Note that algorithms allowing user knowledge/priors incorporation do not aim to move towards the prior given. Instead, the provided prior is used as a starting point for the subsequent iterations since the user believes in finding good configuration candidates in the region specified by the prior. Hence, within the first iterations of optimization, PiBO and BOPrO should sample configurations closer to the user prior. In later iterations, the algorithms might diverge from the given prior (or ignore it with a recovery mechanism in case of harmful feedback).
>
> > Q3 What is the difference between "user knowledge" and "feedback" in Definitions 2 and 3?
>
> We use these terms interchangeably.
>
> **References**
>
> [Poon2011] Poon et al., Sum-Product Networks: A New Deep Architecture. UAI 2011.
>
> [Gens2013] Gens et al., Learning the Structure of Sum-Product Networks. ICML 2013.
>
> [Lindauer2020] Lindauer et al., Best Practices for Scientific Research on Neural Architecture Search. arXiv preprint 2020.
>
> [Choi2020] YooJung Choi et al., Probabilistic Circuits: A Unifying Framework for Tractable Probabilistic Models. Tech. Report UCLA 2020.
>
> [Bischl2023] Bischl et al., Hyperparameter optimization: Foundations, algorithms, best practices, and open challenges. WIREs Data Mining and Knowledge Discovery 2023.
>
> [Giovanelli2024] Giovanelli et al., Interactive Hyperparameter Optimization in Multi-Objective Problems via Preference Learning. AAAI 2024.
>
> [Molina2018] Molina et al., Mixed Sum-Product Networks: A deep architecture for hybrid domains. AAAI 2018.

---

> > ### Comment · Reviewer_cr4S · 2024-11-26
> >
> > Thank you very much for the detailed answer. After carefully reading it, I believe that the overall idea has not been explored yet, but I still have some questions.
> >
> > **About W4:** How does PC balance exploration and exploitation? Does it consider the uncertainty of the model?
> >
> > **About W5.1:** The primary difference between the method in the paper and piBO appears to be in data modeling. For incorporating priors, both methods add prior distributions to the sampling process. However, since PC is an existing method, its way of integrating priors does not seem different. So, what is the innovation in utilizing priors in this paper?
> >
> > **About W7:** The tested problems are five tabular problems, which are insufficient to fully demonstrate the effectiveness of the algorithm in HPO scenarios. Since the proposed method is specifically designed for HPO, which often involves extremely expensive problems, it is crucial to compare it with state-of-the-art HPO methods, even if they are multi-fidelity methods. If these methods can outperform the proposed method without using priors, what is the significance of integrating the proposed prior-based approach with existing methods?
> >
> > **About Q2.2:** The example provided by the authors shows the distribution of sampled points in the first 20 iterations. Why is it necessary for the sampled points in the first 20 iterations to match the prior distribution? In paper [1], Figure 1 demonstrates that sampling starts to deviate from the prior as early as the 8th iteration.
> >
> >  [1]Hvarfner C, Stoll D, Souza A, et al. πBO: Augmenting Acquisition Functions with User Beliefs for Bayesian Optimization[C]//Tenth International Conference of Learning Representations, ICLR 2022. 2022.

---

> > > ### Author Response · Authors · 2024-11-28
> > >
> > > Thank you for participating in the discussion and for carefully reading our rebuttal. We are glad that we could resolve most of your concerns and appreciate that our work is perceived as novel!
> > >
> > > > **About W4:** How does PC balance exploration and exploitation? Does it consider the uncertainty of the model?
> > >
> > > We start the optimization by testing a small number of randomly chosen configurations. Since these configurations are chosen from a uniform distribution over the search space, they are widely spread across the configuration space. Then, we learn a PC employing LearnSPN, which maximizes the likelihood of the given observations. For learning, we require a minimum number $n$ of samples to fall in each PC leaf [11, 12]. Given that in early iterations, the number of available observations is only slightly bigger than $n$ (we choose $n$ accordingly), LearnSPN can only learn shallow PCs (which are not far off of being mere naive factorizations of the joint). In such shallow PCs, each leaf has to cover nearly all available samples because $n$ is close to the number of observations in early iterations. The effect of this is that most (if not all) leaf distributions exhibit relatively large variance in early iterations to cover all observations that are widely spread over the search space. Thus, the probability/density spreads over a large part of the search space to maximize the likelihood of the observations. For example, think of fitting a multivariate Gaussian over 5 evenly spread points of a bound subset of $\mathcal{X} \subset \mathbb{R}$ s.t. the likelihood is maximized. Then, the density will be spread along large parts of $\mathcal{X}$. Since all PC leaves have high variance, the conditional distribution over the search space given the best obtained evaluation score, i.e., $s(\mathcal{H} | \mathbf{y}^*)$, will also exhibit high variance in its leaves (those leaves stay untouched during inference and sampling). However, at the same time, the conditional $s(\mathcal{H} | \mathbf{y}^*)$ will already assign lower likelihoods to poorly performing configurations in early iterations because conditioning on $\mathbf{y}^*$ can be thought of as “selecting” promising search space regions. Here, $\mathbf{y}^*$ is the highest obtained evaluation score and $s$ is the learned PC. Consequently, our sampling-based selection policy will focus the exploration on the more promising regions by assigning higher likelihoods to those regions. Then, it will explore more within these promising areas of the search space due to the high variance of the conditional distribution. The (top-down) PC sampling [13, 14 15, 16] is able to leverage the combination of a relatively high likelihood of promising regions and high variance at leaves to generate new promising configurations (the variance of the corresponding leaf distributions when conditioning to get promising regions directly impacts the nature and the diversity of the resulting samples). Although we do not estimate the model uncertainty like in a pure Bayesian framework (we could adapt and extend the work in [17] to pursue this direction in future work), the high variance at leaf distributions can be interpreted as the uncertainty of the model about the location of high-performing configurations.
> > >
> > > During the progress of the optimization, we collect new observations, mostly moving toward more promising regions of the search space. Thus, in later iterations, the number of observations collected in the promising areas of the search space gets higher. Since LearnSPN constructs PCs by alternating between clustering (i.e., splitting a dataset along the sample-axis) and independence testing (i.e., splitting a dataset along the feature-axis) and the minimum number of samples per leaf $n$ is fixed, LearnSPN will yield deeper PCs to maximize the likelihood of the observations. Thus, each leaf of the PC has to model only a small subset of observations in late(r) iterations. Most of these observations will lie in promising regions of the search space due to our selection policy that samples from $s(\mathcal{H} | \mathbf{y}^*)$. Thus, the variance of the leaves (especially of those of the conditional $s(\mathcal{H} | \mathbf{y}^*)$) will decrease with the increase of the optimization steps, making the model more certain about where high-performing configurations are located. Consequently, in later optimization iterations, our selection policy exploits the knowledge encoded in the PC, leading to a smooth transition from exploration to exploitation as the number of available observations increases.
> > >
> > > We also empirically validated the behavior described above and included plots in Appendix D.5, which show that the variance of the sampled configurations is higher in early iterations and decreases in later iterations. Combined with our strong results in Fig. 2 and 3, these results indicate that our sampling policy trades off exploration and exploitation well.

---

> ### Author Response · Authors · 2024-11-21
> **Discussion period is ending soon**
>
> Dear Reviewer,
>
> We hope to have resolved all your concerns. If you have any further comments, we will be happy to address them before the rebuttal period ends. If there are none, then we would appreciate it if you could reconsider your rating.
>
> Regards,
>
> Authors

---

> ### Author Response · Authors · 2024-11-25
> **End of Discussion Period**
>
> Dear Reviewer,
>
> We hope to have resolved all your concerns. If you have any further comments, we will be happy to address them before the discussion period ends. If there are none, then we would appreciate it if you could reconsider your rating.
>
> Regards,
>
> Authors

---

> ### Author Response · Authors · 2024-11-28
>
> >So, what is the innovation in utilizing priors in this paper?
>
> Both IBO-HPC and PiBO incorporate user priors during optimization. However, both methods differ in how priors are incorporated. First, in contrast to PiBO, our method does **not** use the user prior to weight/shape an acquisition function such as Expected Improvement (EI). Instead, IBO-HPC uses the priors to guide our novel selection policy according to the user's beliefs. Our selection policy exploits the tractable (exact) conditioning and sampling of PCs. It samples configurations from the PC conditioned on the best obtained evaluation score **and** the given user knowledge/prior (if provided). By conditioning on the best obtained evaluation score, we steer the sampling towards promising configurations, and by conditioning on user beliefs, we steer sampling towards regions the user believes to be promising. This allows for a much more targeted, natural, and direct integration of user knowledge because users do not have to consider an acquisition function's behavior (which is non-trivial and often highly unintuitive) when defining a prior (see Fig. 1 and App. A). Second, users can still provide their feedback **also** by simply fixing the values of an arbitrary subset of hyperparameters, thus, without requiring the users to specify a distribution for them. Third, thanks to the tractable and exact inference, users can change their feedback (add/remove arbitrary partial evidence) any time during the optimization without altering the encoded distribution. The PC will still encode a valid distribution on the remaining hyperparameters (conditioned on the user’s input) [13, 14, 15, 16]. Such distribution models complex dependencies between the (remaining) hyperparameters and evaluation scores, thus, it goes beyond the modeling capabilities of simple independent distributions. Fourth, next to the more targeted integration of user knowledge, our novel selection policy is optimization-free. That is, we do not require an additional inner-loop optimization of an acquisition function, making our algorithm more efficient (e.g., see Fig. 4a).

---

> ### Author Response · Authors · 2024-11-28
>
> > **About W7:** The tested problems are five tabular problems, which are insufficient to fully demonstrate the effectiveness of the algorithm in HPO scenarios.
>
> We would like to elaborate more on the choice of benchmarks and tasks to evaluate IBO-HPC. Our experimental setup follows the widely acknowledged practice and used benchmarks representing **real-world HPO/NAS tasks** (see Sec. 3 in [2]). Benchmarks enable researchers to provide statistically robust and reproducible empirical evaluations of HPO methods on relevant practical tasks without having considerable (and often impractical) computational costs. For example, JAHS provides full evaluation statistics over **270K** configurations from a search space composed of neural architectures and other hyperparameters, such as learning rate [3]. The configurations are evaluated on real-world image classification tasks on widely used datasets including CIFAR-10, Fashion-MNIST, and Colorectal Histology. We remark again that the tasks considered in our evaluation are widely adopted and cover image data and tabular data. Thus, we evaluate the performance of IBO-HPC and our baselines methods in real settings that go **beyond mere tabular problems**.
>
> Thanks to benchmarks like JAHS, one does not have to train thousands of neural network configurations but can focus on the optimization/search strategy. In fact, the selected benchmarks allow us to simply "look up" the corresponding evaluation score (in our case, valid/test accuracy) for a given configuration. Thus, the benchmarks replace a full and **costly training** of a neural network with a **cheap** look-up, enabling us to repeat our experiments with 500 different seeds to ensure robustness to stochastic fluctuations. Moreover, benchmarks significantly contribute to the comparability of HPO/NAS methods because, by quoting [3], they enable "democratized NAS research, enabling small academic labs to work on NAS and allowing scientific comparisons between methods without confounding factors". Thus, the essence of such benchmarks is to provide a common framework to test and compare optimization and search methods on real-world scenarios without requiring enormous computational resources (estimated to be **years of computation** on expensive devices [2, 10]).
>
> Note that many HPO methods, including PriorBand and PFNs4BO, among many others, follow this common practice and base their evaluation on this kind of benchmarks. The following list provides a brief overview:
>
> - PiBO [1] employs 4 benchmarks, including an XGBoost and OpenML benchmark (similar to our HPO-B setup, see App. F2 in [1])
> - Priorband [4] uses a subset of tasks from 3 benchmarks, including JAHS (see Sec. 6.1 in [4])
> - PFNs4BO [5] involves 3 benchmarks, including XGBoost tasks from HPO-B (see Sec. 7.1-7.3 in [5])
> - FT-PFN [6] uses 3 benchmarks, including NAS-benchmarks (PD1 and Taskset, see Sec. 6 and App. B in [6])
> - HPO with Conformal Quantile Regression [7] uses 4 benchmarks, including NAS-201 and NAS-301 (that is similar to NAS-201, see Sec. 6 in [7])
> - DPL [8] employs 3 benchmarks, including 2 NAS benchmarks (PD1 and Taskset, see Sec. 5.1 in [8]).
>
> We employ 4 benchmarks (NAS-101, NAS-201, JAHS, and HPO-B) covering 7 tasks across 5 diverse search spaces, including discrete, continuous, and mixed spaces. Thus, our empirical setup is adequate and comprehensive enough to validate the effectiveness of IBO-HPC on real-world cases.

---

> ### Author Response · Authors · 2024-11-28
>
> > **About W7**: [...] Since the proposed method is specifically designed for HPO, which often involves extremely expensive problems, it is crucial to compare it with state-of-the-art HPO methods, even if they are multi-fidelity methods. If these methods can outperform the proposed method without using priors, what is the significance of integrating the proposed prior-based approach with existing methods?
>
> We agree that comparing multi-fidelity methods provides further insights regarding the significance of providing user knowledge/priors. However, dynamically incorporating user knowledge during optimization (like the method we introduce; see Fig. 2 and 3) goes beyond fully automatic and (often) heuristic-based multi-fidelity optimization. In fact, being able to incorporate available user knowledge effectively could considerably speed up the optimization and make these systems truly interactive. Furthermore, we would like to remark that, in our experiments, we **included PriorBand as a multi-fidelity baseline** that **additionally** allows the incorporation of user priors (see Fig. 2 and Sec. 4). IBO-HPC **outperforms** PriorBand in 6/7 tasks where both methods were provided with the same beneficial user knowledge (PriorBand from iteration 1 on, IBO-HPC from iteration 5/15 on).
>
> To provide a further reference with a multi-fidelity approach, we will also add Hyper-tune as an additional multi-fidelity baseline in our experiments. Given the deadline for revising the manuscripts (November 27 AoE), we cannot include Hyper-tune in our revised version.
>
> > **About Q2.2:** The example provided by the authors shows the distribution of sampled points in the first 20 iterations. Why is it necessary for the sampled points in the first 20 iterations to match the prior distribution? In paper [1], Figure 1 demonstrates that sampling starts to deviate from the prior as early as the 8th iteration.
>
> Let us explain why we chose 20 as the horizon under consideration. PiBO and BOPrO employ the same exponential decay mechanism in which the decay parameter $\beta$ determines the speed at which the user prior is weighted down over time. Higher $\beta$ values weigh the prior down in later iterations since the decay exponent is computed as $\frac{\beta}{t}$ where $t$ is the current iteration (see [1], Alg. 1 for PiBO and see [9] Eq. 4 for BOPrO). In Fig. 1, we consider JAHS as a real-world example and define a Gaussian prior with $\mu=1$ and $\sigma=0.3$ over the hyperparameter $R$. We set the number of optimization iterations $T=100$ for PiBO, BOPrO, and IBO-HPC. Following the recommendation in [1], we set the decay parameter $\beta=\frac{T}{10} = 10$ for PiBO and BOPrO. This means that the prior is weighted down the first time after the 10th iteration in PiO and BOPrO. For a fair comparison, we determined the number of iterations $t$ at which the weighted prior has about the same density value at its mode in PiBO, BOPrO, and IBO-HPC. This happens to be the case at $t=20$: At the 20th iteration, the decay factor $\beta$ for both PiBO and BOPrO is $\frac{10}{20} = 0.5$. Since an exponential decay is applied in both methods, the density value of the mode of our prior is then $1.26^{0.5} \approx 1.12$. For IBO-HPC, we chose the decay $\gamma=0.995$. Hence, after 20 iterations, we get $1.26 \cdot \gamma^{20} \approx 1.14$ for the mode of the prior. Thus, we weigh down the prior by approximately the same amount in PiBO, BOPro, and IBO-HPC, ensuring a fair comparison. We discuss this point in more detail in App. A of our revision.
>
> Note that the sampled configurations do **not** need to **strictly** follow the user prior. However, the sampled configurations should not deviate too much from the user prior at early iterations because the selection policy might miss high-performing configurations in the early stages of optimization (plausibly assuming that the norm is that user feedback is beneficial and helpful, especially at early iterations). This could slow down the optimization procedure and even make the algorithm converge to suboptimal solutions. Our empirical results in Fig. 2 and 3 show that IBO-HPC consistently outperforms PiBO and BOPrO when (beneficial) user knowledge is provided. This indicates that the accurate reflection of user knowledge/priors is crucial to efficiently find high-performing configurations.

---

> ### Author Response · Authors · 2024-11-28
>
> **Note on Fig. 2 and Fig. 6 (App. D.3)**
>
> After the deadline for uploading a manuscript revision passed, we noticed two minor issues in two of our plots. In Fig. 6, the line of Priorband is **dotted pink**, but it is represented as **dotted black** in the legend. In Fig. 2(d) we accidentally uploaded the wrong plot file. Hence, the line for Priorband is missing there. However, we provide the final test error in the table below and find that IBO-HPC outperforms Priorband on NAS-101. We will fix both issues in the camera-ready version.
>
> |            | IBO-HPC (no user prior)       | IBO-HPC (prior given at iteration 5) | IBO-HPC (prior given at iteration 10) | Priorband                   |
> | ---------- | ----------------------------- | ------------------------------------ | ------------------------------------- | --------------------------- |
> | test error | $0.0562 \pm 1.2\times10^{-4}$ | $0.0559 \pm 1.3\times10^{-4}$        | $0.0558 \pm 1.2\times10^{-4}$           | $0.0564 \pm 1.4\times10^{-4}$ |
>
>
> *We hope we clarified all the points and kindly ask the reviewer to reassess and raise their score.*
>
>
>
> **References**
>
> [1] Hvarfner C, Stoll D, Souza A, et al., πBO: Augmenting Acquisition Functions with User Beliefs for Bayesian Optimization, ICLR 2022.
>
> [2] Lindauer et al., Best Practices for Scientific Research on Neural Architecture Search. JMLR 2020.
>
> [3] Bansal et al., JAHS-Bench-201: A Foundation For Research On Joint Architecture And Hyperparameter Search. NeurIPS 2022.
>
> [4] Mallik et al., Priorband: A Practical hyperparameter optimization in the age of deep learning. NeurIPS 2024.
>
> [5] Müller et al., Pfns4bo: In-context learning for bayesian optimization. ICML 2023.
>
> [6] Rakotoarison et al., In-Context Freeze-Thaw Bayesian Optimization for Hyperparameter Optimization. ICML 2024.
>
> [7] Salinas et al., Optimizing Hyperparameters with Conformal Quantile Regression. ICML 2023.
>
> [8] Kadra et al., Scaling Laws for Hyperparameter Optimization. NeurIPS 2023.
>
> [9]  Souza et al., Bayesian Optimization with  a Prior for the Optimum. PKDD 2021.
>
> [10] Ying et al., NAS-Bench-101: Towards Reproducible Neural Architecture Search. ICML 2019.
>
> [11] Gens et al., Learning the Structure of Sum-Product Networks, ICML 2013.
>
> [12] Molina et al., Mixed Sum-Product Networks: A Deep Architecture for Hybrid Domains. AAAI 2018.
>
> [13] Poon et al., Sum-Product Networks: A New Deep Architecture. UAI 2011.
>
> [14] Peharz et al., On Theoretical Properties of Sum-Product Networks. AISTATS 2015.
>
> [15] Peharz et al., On the Latent Variable Interpretation in Sum-Product Networks. IEEE TPAMI 2017
>
> [16] Choi et al., Probabilistic Circuits: A Unifying Framework for Tractable Probabilistic Models. UCLA 2020.
>
> [17] Ventola et al., Probabilistic Circuits That Know What They Don't Know. UAI 2023.

---

> > ### Author Response · Authors · 2024-12-01
> > **End of Discussion Phase**
> >
> > Dear Reviewer,
> >
> > Thank you again for the fruitful discussion!
> >
> > We hope we have resolved all your concerns. If you have any additional comments, we would be glad to address them before the discussion phase ends. Otherwise, we would appreciate it if you could reconsider your rating.
> >
> > Best regards,
> >
> > The Authors

---

> > > ### Author Response · Authors · 2024-12-02
> > > **End of Discussion Phase**
> > >
> > > Dear Reviewer,
> > >
> > > We hope we have resolved all your concerns. If you have any additional comments, we would be glad to address them before the discussion phase ends. Otherwise, we would appreciate it if you could reconsider your rating.
> > >
> > > Best regards,
> > >
> > > The Authors

---

> > > > ### Comment · Reviewer_cr4S · 2024-12-03
> > > >
> > > > Thanks for the response. However, I have several concerns regarding the experimental setup and its presentation:
> > > >
> > > > 1. **Completeness of Experiments**:
> > > >     - The number of test problems included in the experiments seems insufficient to comprehensively evaluate the proposed method.
> > > >     - The presentation of results is limited. For instance, Figure 2 uses trajectory plots with a large number of algorithms, where similar line colors make it challenging to distinguish between results clearly.
> > > > 2. **Lack of Statistical Analysis**:
> > > >     - The study does not provide statistical significance tests or aggregated performance metrics to rigorously compare the proposed method with baseline algorithms. This limits the ability to draw strong conclusions about the effectiveness of the approach.
> > > > 3. **Rationale for Intervention Design**:
> > > >     - The timing and frequency of interventions in the experiments appear to be based on intuition, with no clear justification provided. The rationale behind this design choice and its significance remains unclear.
> > > > 4. **Parameter Sensitivity**:
> > > >     - The algorithm seems to exhibit sensitivity to its parameters. There is insufficient guidance on how these parameters should be set for different problems, particularly in cases where the quality of prior knowledge (good or bad) is uncertain.

---

> > > > > ### Author Response · Authors · 2024-12-04
> > > > >
> > > > > >  The number of test problems included in the experiments seems insufficient to comprehensively evaluate the proposed method.
> > > > >
> > > > > We do **not** agree that the problems included in our setup are insufficient for a comprehensive evaluation of IBO-HPC. As we have highlighted in our last response, the selection of tasks and benchmarks follows standard practices in BO and HPO research. We have already pointed to several peer-reviewed works published at top-ranked conferences (ICLR, ICML, NeurIPS) that use the same or similar benchmarks and tasks as we do (not only in the type of task/benchmark but also in the number of tasks). See, for example, PiBO [1], Priorband [4],  PFNs4BO [5], FT-PFN [6], HPO with Conformal Quantile Regression [7], and DPL [8]. For more details, please refer to our last response above.
> > > > >
> > > > > > The presentation of results is limited. For instance, Figure 2 uses trajectory plots with a large number of algorithms, where similar line colors make it challenging to distinguish between results clearly.
> > > > >
> > > > > We disagree on this point because we follow standard practice from the BO and HPO literature to represent our results [1, 2, 3, 4, 5, 6, 7]. We tried to make the plots as clear as possible using different line styles for IBO-HPC (solid), other methods that allow user priors such as PiBO (dotted), standard BO/HPO, and multi-fidelity baselines (dashed), and slightly modified baselines that also allow infusing user knowledge after modifications (dash-dotted).
> > > > >
> > > > > To address the request of two reviewers (**including yours**) to show more baselines, we added three more baselines (Priorband, random search with early user interaction, and random search with late user interaction). Since we want to provide the reader with an easy-to-see and immediate comparison of the results of different methods, we decided to show the trajectories of all baselines and IBO-HPC in one plot (Fig. 2) in the main paper. In Fig. 7 (App. D.3), we show our results with fewer and most competitive baselines.
> > > > >
> > > > > To make the plots even clearer, we will make minor adaptations in coloring for the camera-ready version.

---

> > > > > > ### Author Response · Authors · 2024-12-04
> > > > > >
> > > > > > > The study does not provide statistical significance tests or aggregated performance metrics to rigorously compare the proposed method with baseline algorithms. This limits the ability to draw strong conclusions about the effectiveness of the approach.
> > > > > >
> > > > > > As reported in Sec. 4, **we repeated our experiments with 500 different seeds and plotted the mean and standard error in all our plots**. This is in line (often even more than commonly done) with standard practices in BO and HPO research [1, 2, 3, 4, 5, 6, 7]. Also, we mitigated stochastic fluctuations and obtained a low standard error thanks to the high number of repetitions. Thus, it is often barely visible in our plots. Since there are no overlaps of standard error regions in our plots (e.g., see Fig. 2), and given the 500 repetitions of all experiments, we conclude that our results are statistically significant. Moreover, to better show the level of sensitivity of the different methods to random seeds, **we already provided the cumulative distribution function (CDF) of the test error** obtained with 500 seeds in Fig. 9 in the Appendix.
> > > > > >
> > > > > > As requested by the reviewer, we also run a statistical test by applying the one-sided Wilcoxon test on pairs of runs conducted with the same seed (the same logs used in Fig. 2 and 3). The statistical tests **confirm our findings** in Sec. 4.1 and 4.2. Please see the following.
> > > > > >
> > > > > > **IBO-HPC outperforms existing methods when user knowledge is given**
> > > > > >
> > > > > > The table below shows the p-values from the one-sided Wilcoxon test on runs where user knowledge is given. Therefore, we compare IBO-HPC, PiBO, BOPrO, and Priorband because those are the baselines allowing user priors. The results in the table below demonstrate that IBO-HPC **significantly outperforms** PiBO and BOPrO on all tasks shown in Fig. 2. Also, IBO-HPC is statistically significantly better than Priorband on **4/5** tasks. The only exception is NAS-201, where Priorband slightly outperforms IBO-HPC (see Fig. 2(e)).
> > > > > >
> > > > > >
> > > > > > |                                   | JAHS (CIFAR10)        | JAHS (C. Histology)   | JAHS (Fashion-MNIST)  | NAS201               | NAS101               |
> > > > > > | --------------------------------- | --------------------- | --------------------- | --------------------- | -------------------- | -------------------- |
> > > > > > | IBO-HPC (int. @ 5) vs. PiBO       | $4.0 \times 10^{-11}$ | $8.8 \times 10^{-15}$ | $6.4 \times 10^{-6}$  | $1.9 \times 10^{-6}$ | $1.3 \times 10^{-4}$ |
> > > > > > | BO-HPC (int. @ 5) vs. BOPrO       | $1.7 \times 10^{-15}$ | $1.6 \times 10^{-8}$  | $1.2 \times 10^{-10}$ | $1.9 \times 10^{-6}$ | $2.8 \times 10^{-4}$ |
> > > > > > | BO-HPC (int. @ 5) vs. Priorband   | $8.8 \times 10^{-16}$ | $1.2 \times 10^{-13}$ | $1.2 \times 10^{-13}$ | $0.14$               | $2.4 \times 10^{-4}$ |
> > > > > > | IBO-HPC (int. @ 10) vs. PiBO      | $1.5 \times 10^{-15}$ | $5.3 \times 10^{-15}$ | $5.8 \times 10^{-6}$  | $1.9 \times 10^{-6}$ | $1.3 \times 10^{-4}$ |
> > > > > > | IBO-HPC (int. @ 10) vs. BOPrO     | $2.3 \times 10^{-13}$ | $1.8 \times 10^{-8}$  | $2.9 \times 10^{-10}$ | $1.9 \times 10^{-6}$ | $1.3 \times 10^{-4}$ |
> > > > > > | IBO-HPC (int. @ 10) vs. Priorband | $2.4 \times 10^{-15}$ | $2.5 \times 10^{-13}$ | $4.1 \times 10^{-6}$  | $0.1$                | $2.9 \times 10^{-4}$ |
> > > > > >
> > > > > >
> > > > > > **IBO-HPC is on par with existing BO approaches when no user knowledge is given**
> > > > > >
> > > > > > The table below shows the p-values from the one-sided Wilcoxon test on runs where **no** user knowledge is provided. Therefore, we compare IBO-HPC with BO using random forests (RF) as a surrogate, BO using Tree-Parzen Estimator (TPE) as a surrogate, and SMAC. We find that no baseline consistently outperforms IBO-HPC on all tasks. Notably, IBO-HPC significantly outperforms all BO baselines on NAS-201 and NAS-101. On JAHS, the results are mixed. While IBO-HPC outperforms BO w/ RF on 2/3 of tasks significantly, BO /w TPE, and SMAC outperforms IBO-HPC on JAHS. Since there is no clear winner in vanilla BO tasks, we conclude that IBO-HPC is on par with existing approaches when no user knowledge is available.
> > > > > >
> > > > > > |                       | JAHS (CIFAR10)       | JAHS (C. Histology)  | JAHS (Fashion-MNIST) | NAS201               | NAS101               |
> > > > > > | --------------------- | -------------------- | -------------------- | -------------------- | -------------------- | -------------------- |
> > > > > > | IBO-HPC vs. BO /w RF  | $5.3 \times 10^{-9}$ | $9.8 \times 10^{-6}$ | $0.08$               | $1.4 \times 10^{-4}$ | $8.8 \times 10^{-4}$ |
> > > > > > | IBO-HPC vs. BO /w TPE | $0.19$               | $0.93$               | $0.28$               | $7.8 \times 10^{-4}$ | $9.5 \times 10^{-7}$ |
> > > > > > | IBO-HPC vs. SMAC      | $0.9$                | $0.96$               | $0.52$               | $7.7 \times 10^{-3}$ | $2.8 \times 10^{-4}$ |
> > > > > >
> > > > > > We will include the results of the Wilcoxon test in the camera-ready version of our paper.

---

> > > > > > > ### Author Response · Authors · 2024-12-04
> > > > > > >
> > > > > > > > The timing and frequency of interventions in the experiments appear to be based on intuition, with no clear justification provided. The rationale behind this design choice and its significance remains unclear.
> > > > > > >
> > > > > > > The user interactions were **not selected based on intuition**, and we **provide a clear justification** for selecting the user interactions in our experiments in our paper (see lines 400-415 and App. D). Our goal was to evaluate IBO-HPC on a wide variety of user knowledge, including highly beneficial knowledge, knowledge with uncertainty (i.e., distributions provided by the user), highly harmful knowledge, and an alternating sequence of beneficial and harmful knowledge. With this, our evaluation covers **diverse** user interactions that can occur in practical use, ranging from extreme cases (e.g., the harmful interactions where the user fixes the majority of hyperparameters to the worst possible values) to moderate and plausible cases (e.g., the user specifies a distribution over a small set of hyperparameters s.t. promising values receive a higher probability).
> > > > > > >
> > > > > > > Note that our setup **is very similar** to PiBO [1] and BOPrO [8]. Here, the authors defined strong, weak, and moderate priors over the optimum or poorly performing configurations. Similarly, we define user beneficial and harmful user knowledge (either as fixed values/point mass or with uncertainty, reflected by a user-defined distribution over values of certain hyperparameters). Moreover, we simulated the interventions more towards the beginning of the search since, at that stage, their impact is higher. In fact, at a very late stage of the optimization, beneficial feedback could have low or no impact (it is more likely that a good solution has been already found, as shown in the plots), and harmful feedback could be simply ignored (finding a better solution is even more unlikely, instead, at an early stage, the recovery mechanism could be much more decisive for the search).

---

> > > > > > > > ### Author Response · Authors · 2024-12-04
> > > > > > > >
> > > > > > > > > The algorithm seems to exhibit sensitivity to its parameters. There is insufficient guidance on how these parameters should be set for different problems, particularly in cases where the quality of prior knowledge (good or bad) is uncertain.
> > > > > > > >
> > > > > > > > We disagree that we do not provide guidance on the selection of parameters of IBO-HPC. Two parameters mainly control our algorithm:
> > > > > > > > - the decay rate $\gamma$ that ensures recovery from harmful user knowledge
> > > > > > > > - $L$, which controls for how many iterations the surrogate PC is fixed before retrained on new observations to ensure sufficient exploration of the search space
> > > > > > > >
> > > > > > > > We have provided an ablation study in App. D3 (see Fig. 10-12) analyzing the effect of both parameters on the behavior of IBO-HPC. We find that lower $\gamma$ leads to faster recovery as expected. Also, lower $\gamma$ can lead to better final performance if harmful user knowledge is provided. However, the sensitivity to the choice of $\gamma$ is relatively low; hence, setting $\gamma$ between 0.3 and 0.9 is usually a good choice.
> > > > > > > >
> > > > > > > > Similarly, $L$ has no severe effect on the final performance of IBO-HPC. We varied $L$ between 5 and 30 and found that retraining the PC more often (lower $L$) can lead to faster convergence. However, given sufficiently many iterations, the final performance is similar for different values of $L$. Thus, IBO-HPC does not exhibit high sensitivity w.r.t. $L$, and a choice between 5 and 10 works well in practice.
> > > > > > > >
> > > > > > > > Furthermore, we analyzed the behavior of IBO-HPC when $f^*$ used to compute the conditional distribution $s(\mathcal{H} | f^*)$ over the configuration space is not set to the best evaluation score (see Fig. 11 in App. D.3). Since our selection policy samples from $s(\mathcal{H} | f^*)$, setting $f^*$ to other values than the best evaluation score obtained in the observations (we tested setting $f^*$ the .75, .5 and .25 quantile) degrades the performance of IBO-HPC in accordance with the distance between $f^*$ and the best observed evaluation score. This demonstrates that the conditioning mechanism and our selection policy guide the optimization process towards high-performing configurations.
> > > > > > > >
> > > > > > > > In the last reply, we also discussed the exploration-exploitation mechanism of IBO-HPC. We mentioned the parameter $n$ which determines the minimum number of samples represented by each leaf in the learned PC. $n$ is set to 10 in all of our experiments. This is a standard value used in LearnSPN [10] and ensures that IBO-HPC explores the search space in early iterations (because each leaf must cover almost all samples in early iterations). In later iterations (i.e., more observations are available), $n=10$ ensures that LearnSPN yields an accurate model that leverages the information about $f$ present in the observations. This allows IBO-HPC to exploit the available observations and it enables the optimization to converge to high-performing configurations.
> > > > > > > >
> > > > > > > > To summarize, in our paper, we have already provided a thorough ablation study of $\gamma$ and $L$ (Fig. 10-12 in App. D.3), i.e., the two parameters that mostly influence the behavior of IBO-HPC. This study provides useful and helpful insights that can be used to set them properly. Since we use a standard value for $n$ ($n=10$), practitioners and researchers can follow standard practices from the PC literature (N.B., in several PC/SPN papers $n$ is denoted as $\mu$ but we avoid confusion with what is commonly used to represent an expected value).
> > > > > > > >
> > > > > > > > **References**
> > > > > > > >
> > > > > > > > [1] Hvarfner C, Stoll D, Souza A, et al., πBO: Augmenting Acquisition Functions with User Beliefs for Bayesian Optimization, ICLR 2022.
> > > > > > > >
> > > > > > > > [2] Lindauer et al., Best Practices for Scientific Research on Neural Architecture Search. JMLR 2020.
> > > > > > > >
> > > > > > > > [3] Mallik et al., Priorband: A Practical hyperparameter optimization in the age of deep learning. NeurIPS 2024.
> > > > > > > >
> > > > > > > > [4] Müller et al., Pfns4bo: In-context learning for bayesian optimization. ICML 2023.
> > > > > > > >
> > > > > > > > [5] Rakotoarison et al., In-Context Freeze-Thaw Bayesian Optimization for Hyperparameter Optimization. ICML 2024.
> > > > > > > >
> > > > > > > > [6] Salinas et al., Optimizing Hyperparameters with Conformal Quantile Regression. ICML 2023.
> > > > > > > >
> > > > > > > > [7] Kadra et al., Scaling Laws for Hyperparameter Optimization. NeurIPS 2023.
> > > > > > > >
> > > > > > > > [8]  Souza et al., Bayesian Optimization with  a Prior for the Optimum. PKDD 2021.
> > > > > > > >
> > > > > > > > [9] Ying et al., NAS-Bench-101: Towards Reproducible Neural Architecture Search. ICML 2019.
> > > > > > > >
> > > > > > > > [10] Gens et al., Learning the Structure of Sum-Product Networks, ICML 2013.

---

### Official Review · Reviewer_Knfb · 2024-11-04

**Soundness:** 4
**Presentation:** 4
**Contribution:** 4
**Rating:** 8
**Confidence:** 3

**Summary:**

This paper introduces a Bayesian Optimization (BIO) framework for interactive Hyperparameter Optimization (HPO).
This framework is termed Bayesian Optimization via Hyperparameter Probabilistic Circuits (IBO-HPC).
By interactive, they allow human experts to provide feedback during the HPO process through tractable Probabilistic Circuits (PCs).
Specifically, they use PCs, a tractable joint distribution over hyperparameters and evaluation scores, to allow exact conditional inference and sample.
They back up their method with extensive numerical experiments.

**Strengths:**

This paper presents a refreshing and  interesting idea with solid execution.
**Clarity.** The writing of this paper is good. This paper is easy to follow. I especially appreciate the line-by-line walkthrough of algo1 in page 5.

**Originality.**  This paper utilizes PCs to perform exact Bayesian inference, especially conditional inference to include human experts’ feedback. While it makes sense as it sounds, I haven’t seen other solid executions of similar ideas.


**Significance.** I find IBO-HPC significant. Beyond its empirical effectiveness, unlike many human-in-the-loop HPO methods, it does not add computational burden.

**Solid Experiments.** Numerical validations are solid and extensive, making this paper more convincing.

**Weaknesses:**

NA

**Questions:**

How does IBO-HPC handle multiple human experts’ feedback? For example, consider a distributed or federated setting.

Can IBO-HPC handle online or life-long learning models with human experts monitoring in real time? If yes, can you sketch the response time of the model?

---

> ### Author Response · Authors · 2024-11-18
> **Rebuttal for Knfb**
>
> We thank the reviewer for the positive review and appreciate that the paper is perceived as significant and original with a solid evaluation. Please note that we marked the edits in the manuscript made by following the feedback provided by other reviews in color (green, blue, and pink).
>
> Let us answer the remaining questions below.
>
> > Q1 How does IBO-HPC handle multiple human experts’ feedback? For example, consider a distributed or federated setting.
>
> This is an interesting use case of our method! Currently, there is no explicit mechanism to handle feedback from multiple users. However, a straightforward way to represent different users is to define a distribution over the knowledge provided by different users. For example, having $n$ users, each with some knowledge $\mathcal{K}_i$, one could define a distribution $q( \mathcal{K}) = b(n)*qi( \mathcal{K}_i | n=i)$ where $b$ is a discrete distribution over all users. In this way, first, the feedback of a user $i$ is selected, then, the selected user’s knowledge (here, a distribution) is used to guide the optimization. Note that we could also represent the level of expertise of each user in $b$ by assigning higher probabilities to users with a high level of expertise and lower probabilities to users with a low level of expertise.
>
> > Q2 Can IBO-HPC handle online or life-long learning models with human experts monitoring in real time? If yes, can you sketch the response time of the model?
>
> We assume that this question aims to use IBO-HPC in a lifelong learning setting to optimize the hyperparameters of a model that is continuously updated based on new observations. IBO-HPC can be used in these settings as well. Let us briefly describe how: Given a task t that should be solved by some model $m$ in a lifelong learning setting, the hyperparameters $\mathcal{H}$ of $m$ can be optimized to obtain a high-performing model. Leveraging IBO-HPC for this yields a surrogate PC s representing the joint distribution over hyperparameters and evaluations for task t, i.e., it approximates some distribution $p_t$ . On a newly arriving task $t+1$ (e.g., a new class in image classification), s can be used as a surrogate to warm-start the HPO procedure for task $t+1$ if the hyperparameter search space $\mathcal{H}$ remains (roughly) the same. Running IBO-HPC to optimize m on $t+1$ then corresponds to just running IBO-HPC on $t$. Hence, users can interact with the optimization as described in our paper on the new task. Since the distribution over configurations and evaluations on $t+1$ might change, i.e, $p_t$ != $p_{t+1}$, the PC s would be updated automatically when performing HPO on $t+1$ matching $p_{t+1}$ better.
>
> This use case is closely related to Hyperparameter Transfer Learning (HTL). We already started investigating how PCs can be used in HTL settings, even when the search spaces change across tasks (i.e., $\mathcal{H}$ is not fixed) by leveraging tractable marginalization of unused hyperparameters.

---

### Author Response · Authors · 2024-12-04
**Summary of Changes**

We thank all the reviewers again for their time and feedback. Given the end of the discussion phase, we would like to briefly summarize all the points we addressed that have been mentioned in the reviews and the discussion phase.

We would like to remark that the clarifications and the additional empirical evidence provided during the rebuttal have only confirmed and further supported our previous claims based on the initial submitted results. Thus, the integrity, consistency, and validity of our method have not been compromised and, instead, further consolidated.

**Reviewer cr4S**

1. We clarified the rationale behind the time and quality of user knowledge provided to IBO-HPC and to all baselines allowing the integration of user priors (see [our rebuttal](https://openreview.net/forum?id=uC003NHlEi&noteId=btEfwOFOfm)).
2. We added a rigorous convergence analysis of IBO-HPC (see App. B.2)
3. We provided a theoretical justification showing the accuracy of our approximation in Eq. 2 (see App. B.4)
4. To improve clarity, we added more explanations to the equations employed to formally introduce our method (see Sec. 3 in our revision).
5. We added a brief discussion of LearnSPN, which is employed to learn the surrogate model during optimization (see App. C)
6. We better clarified the differences between PiBO, BOPrO, and IBO-HPC, focusing on how user knowledge is incorporated into the generation and selection process of configurations. We made this clearer (see [our rebuttal](https://openreview.net/forum?id=uC003NHlEi&noteId=vEokiw8MlF) and Sec. 1, 2, 3 in our revision).
7. As requested by the reviewer, we added Priorband as an additional multi-fidelity baseline to our evaluation (see Sec. 4 in our revision).
8. We discussed our experimental setup in detail and showed that we follow standard practices of the HPO/NAS literature by indicating several published and peer-reviewed works (see our answer to cr4S above). This discussion includes the qualitative choice and the number of selected benchmarks and tasks on which we evaluate our method IBO-HPC.
9. Besides the choice of benchmarks, we also discussed the choice of user knowledge employed in our experiments. Our paper provides a detailed description and justification of how user knowledge is specified. Moreover, we provided references to peer-reviewed published works that follow a similar specification of user knowledge (see our answer to cr4S above).
10. We provided a detailed explanation of how IBO-HPC balances exploration and exploitation. Next to the theoretical viewpoint, we also empirically verified our explanation by showing that the sampling variance decreases over time, indicating that IBO-HPC tends to explore in early iterations and to exploit in later iterations (see App. D.5; Fig. 14 and 15).
11. We added the one-sided Wilcoxon tests that further confirms our results and shows the statistical significance of our findings, i.e., IBO-HPC outperforms PiBO, BOPrO, and Priorband on a set of diverse tasks (see our [answer to cr4S](https://openreview.net/forum?id=uC003NHlEi&noteId=oQJa1jmjwO)).
12. We discussed and empirically showed the sensitivity of the parameters of IBO-HPC, referring to our ablation analysis in App. D.3 (Fig. 10-12). The empirical results demonstrate that IBO-HPC has low sensitivity with respect to the choice of these parameters.

**Reviewer nTFs**

1. As requested by the reviewer, we provided two additional baselines in our experiments (random search with early user interaction and random search with late user interaction, see Sec. 4 in our revision).
2. We pointed out the advantages of IBO-HPC's mechanism to incorporate user knowledge compared to other methods that allow the incorporation of user priors (see Introduction, Fig. 1 and App. A in our manuscript).
3. We have better highlighted the contributions we make and will incorporate appropriate changes in the camera-ready version of the paper (these changes were done at the time no revised PDF could be uploaded anymore; see our [response](https://openreview.net/forum?id=uC003NHlEi&noteId=UmJGTaJV34)).
4. We polished our notation for the camera-ready version to make it easier for the reader to follow (these changes were done at the time no revision could be uploaded anymore; see our [response](https://openreview.net/forum?id=uC003NHlEi&noteId=NE3s5JEWXj)).
5. In Sec. 3 and Sec. 4, we further highlighted the advantages of the novel idea of using PCs in BO to better justify our choice of PCs as a surrogate model in interactive BO (these changes were done at the time no revision could be uploaded anymore; see our response [response](https://openreview.net/forum?id=uC003NHlEi&noteId=DlFF2FVe9p)).

---

### Meta-Review · Area_Chair_hzXG · 2024-12-20

**Metareview:**

This paper introduces a novel iteractive Bayesian optimization framework using probabilistic circuits (PCs) as the surrogate model. It allows users to provide external knowledge and intervene the optimization process. It's evaluated on both HPO and neural architecture search benchmarks.

Pros:
- Novel integration of PCs in the interactive Bayesian optimization framework
- Flexible integration of user feedbacks.
- Tractable conditional inference that enables sampling from the exact conditional hyperparameter distribution

Cons:
- Unclear notation and presentation
- Significance compared to other methods could allow incorporating user knowledge
- Insufficient evaluation
- Unclear rational in the user intervention design

Reviewers have divergent opinions on the quality of the presentation, the design of user intervention and empirical evaluation. Authors feedback addressed some concerns (see the additinoal comments section), but do not change the negative reviewers' assessment on the main concerns. The reviewers with positive ratings provided fairly short and abstract reviews, and did not engage in the following discussion. There is no sufficient evidence to support acceptance of this paper.

**Additional Comments On Reviewer Discussion:**

The authors provided detailed response to the initial reviews and follow up questions. They addressed the questions about the lack of statistical analysis of results, lack of parameter sensitivity analysis and added additional baselines.

They also provided revision to notations and presentation. However, that does not alleviate the reviewer's concern on clarity. They suggest a fresh round of reviewing consideration the amount of changes made to the submission.

The authors' rebuttal does not address the concerns on the design of user intervention shared by both negative reviewers.

---

### Decision · Program_Chairs · 2025-01-22

Reject